# Provably Efficient Multi-Objective Bandit Algorithms under Preference-Centric Customization

## Abstract

Existing multi-objective multi-armed bandit (MO-MAB) approaches mainly focus on achieving Pareto optimality. However, a Pareto optimal arm that receives a high score from one user may lead to a low score from another, since in real-world scenarios, users often have diverse preferences across different objectives. Instead, these preferences should inform *customized learning*, a factor usually neglected in prior research. To address this need, we study a *preference-aware* MO-MAB framework in the presence of explicit user preferences, where each user's overall-reward is modeled as the inner product of user preference and arm reward. This new framework shifts the focus from merely achieving Pareto optimality to further optimizing within the Pareto front under preference-centric customization. To the best of our knowledge, this is the first theoretical exploration of customized MO-MAB optimization based on explicit user preferences. This framework introduces new and unique challenges for algorithm design for customized optimization. To address these challenges, we incorporate *preference estimation* and *preference-aware optimization* as key mechanisms for preference adaptation, and develop new analytical techniques to rigorously account for the impact of preference estimation errors on overall performance. Under this framework, we consider three preference structures inspired by practical applications, with tailored algorithms that are proven to achieve near-optimal regret, and show good numerical performance.

## 1 Introduction

Multi-objective multi-armed bandit (MO-MAB) problem is an important extension of the multi-armed bandits (MAB) (Drugan & Nowe, 2013). In MO-MAB problems each arm is associated with a $D$-dimensional reward vector. In this environment, objectives could conflict, leading to arms that are optimal in one dimension, but suboptimal in others. A natural solution is utilizing Pareto ordering to compare arms based on their rewards (Drugan & Nowe, 2013). Specifically, for any arm $i \in [K]$, if its expected reward $\boldsymbol{\mu}_i$ is non-dominated by that of any other arms, arm $i$ is deemed to be Pareto optimal. The set containing all Pareto optimal arms is denoted as Pareto front $\mathcal{O}^*$. Formally, $\mathcal{O}^* = \{i \mid \boldsymbol{\mu}_j \nsucc \boldsymbol{\mu}_i, \forall j \in [K] \setminus i\}$, where $\boldsymbol{u} \succ \boldsymbol{v}$ holds if and only if $\boldsymbol{u}(d) > \boldsymbol{v}(d), \forall d \in [D]$. The performance is then evaluated by Pareto regret, which measures the cumulative minimum distance between the learner's obtained rewards and rewards of arms within $\mathcal{O}^*$ (Drugan & Nowe, 2013). However, simply obtaining a solution that has good Pareto regret does not take into account the fact that individual users would like to pick the choice that matches their specific needs. As the example depicted in Fig. 1, given multiple Pareto optimal restaurants, one user may give a higher preference to quality, while another user may give a higher preference to affordability. This means that *user preferences* need to be accounted for in the MO-MAB problem set up in order to choose the right solution on the Pareto front $\mathcal{O}^*$. This is the focus of this paper.

Although numerous MO-MAB studies have been conducted, **most of them achieve Pareto optimality via an arm selection policy that is uniform across all users**, which we refer to as a *global policy*. Specifically, one representative line of research focuses on efficiently estimating the entire Pareto front $\mathcal{O}^*$, and the action in each round is *randomly* chosen on the estimated Pareto front (Drugan & Nowe, 2013; Turgay et al., 2018; Lu et al., 2019; Drugan, 2018; Balef & Maghsudi, 2023). Another line of research transforms the $D$-dimensional reward into a scalar using a scalarization function, which targets a specific Pareto optimal arm solution without the costly estimation of entire Pareto front Drugan & Nowe (2013); Busa-Fekete et al. (2017); Mehrotra et al. (2020); Xu & Klabjan (2023).

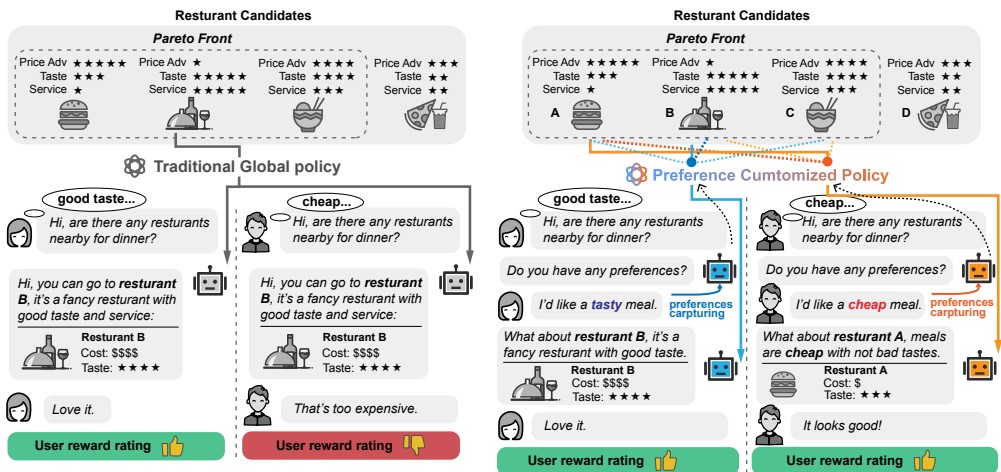

Figure 1: A scenario of users interacting with a conversational recommender for restaurant recommendation. (a) Recommender achieves Pareto optimality but receives low rating from user. (b) Recommendations with high users' ratings when the recommender captures users' preferences and aligns optimization with preferences.

These studies construct the scalarization function in a user-agnostic manner, causing the target arm solution to remain the same across different users.

*However, simply achieving Pareto optimality using a global policy may not yield favorable outcomes, since, as mentioned earlier, users often have diverse preferences across different objectives.* Consider the following scenario depicted in Fig. 1(a), where two users with distinct preferences interact with a conversational recommender to find a nearby restaurant for dinner. The upper section lists restaurant options, each associated with multi-dimensional rewards (e.g., price, taste, service), while the lower section shows the dialogues and users' reward ratings for the recommendations. Clearly, restaurants A, B, and C are Pareto optimal, as none of their rewards are dominated by others. Previous research using a global policy would either randomly recommend a restaurant from A, B, or C, or select one based on a fixed global criterion to achieve Pareto optimality. However, while recommending a restaurant like B might lead to positive feedback from user-1, it is likely to result in a low reward rating from user-2, who prefers an economical meal, since restaurant B is expensive. In contrast, Fig. 1(b) illustrates that when the system accurately captures user preferences (e.g., user-1 prefers a tasty meal, while user-2 prefers a cheap meal), it can select options more likely to receive positive reward ratings from both users. *Therefore, we argue that optimizing MO-MAB should be customized based on the user preferences rather than solely aiming for Pareto optimality with a global policy.*

While interactive user modeling and customized optimization cross multiple objectives presents promising experimental results in some areas including recommendation (Xie et al., 2021), ranking (Wanigasekara et al., 2019), and more (Reymond et al., 2024), there are no theoretical studies on MO-MAB customization under explicit user preferences. Particularly, two open problems remain: *(1) how to develop provably efficient algorithms for customized optimization under different preference structure (e.g., unknown preference, non-stationary preference, corrupted preference)? (2) how does the additional user preferences impact the overall performance?*

To fill this gap, we introduce a formulation of MO-MAB problem, where each user is associated with a $D$-dimensional *preference vector*, referred to as a preference for short, with each element representing the user's preference for the corresponding objective. Formally, in each round $t$, user incurs a stochastic preference $c_t \in \mathbb{R}^D$. The player selects an arm $a_t$ and observes a stochastic reward $r_{a_t,t} \in \mathbb{R}^D$. We define the scalar *overall-reward* as the inner product of arm reward $r_{a_t,t}$ and user preference $c_t$. The learner's goal is to maximize the overall-reward accrued over a given time horizon. For performance evaluation, we define the regret metric as the cumulative expected gap related to the overall-reward. We term this problem as **Preference-Aware** MO-MAB (PAMO-MAB).

Our contributions are summarized as follows.

- **New theoretical results.** *To the best of our knowledge, this is the first work that explicitly showcases the fundamental impact of user preferences in the regret optimization of MO-MAB problems.* Motivated by real applications, we consider the PAMO-MAB problem under three practical preference structures: known (possibly dynamic) preferences, unknown (possibly dynamic) preferences with

feedback, and hidden preferences, with tailored algorithms that are proven to achieve near-optimal regret in each case. The expressions of our results are in an explicit form that capture a clear dependency on various preference setups.

- **New preference-aware algorithm design.** We derive a lower bound to highlight the fundamental reason why existing algorithms based on the global policies are no longer feasible for the PAMO-MAB problem. Hence, we propose tailored algorithms for PAMO-MAB under different preference structures. In contrast to other MO-MAB methods, our algorithms involve two novel designs: (D1) *Preference estimation mechanism* and (D2) *Preference-aware optimization*, which allows us to effectively capture the user preferences and optimize the overall outcome under the estimated preferences for customization. Note that the designs of (D1) and (D2) are not trivial generalizations of existing MO-MAB methods because the preference structure and the reward structure are different. In addition to reward estimation, the preference estimation also introduces uncertainty, which further affects the arm selection and reward estimation, making it necessary to carefully design the estimation approach and new objective term for optimization.

- **New analytical ideas.** Our regret analysis involves novel ideas for solving the new difficulties due to the design of (D1) and (D2). (a) The regret is influenced by the joint estimation error of both preference and reward, which significantly increases the difficulty of regret analysis. To address this, we introduce a tunable parameter $\epsilon$ to decompose the suboptimal actions into two disjoint sets based on whether the corresponding preference estimation is sufficiently accurate or not. This enables the regret that is caused by reward estimation error to be independently analyzed on such two sets. (b) When the preference estimation is accurate under parameter $\epsilon$, the error can be analyzed based on the reward estimation. Moreover, when the preference estimation is not sufficiently accurate, since the idea (a) does not explicitly decouple the effect of the joint error in preference and reward estimations, the effect of the set of suboptimal actions is still unclear. To address this, we transfer this set to a uniform imprecise estimation set, such that a tractable formulation can be constructed based on the distance bound.

## 2 RELATED WORK

**Multi-Objective Multi-Armed Bandits.** MO-MAB extends scalar rewards in the standard MAB problem to multi-dimensional vectors. The Pareto-UCB work (Drugan & Nowe, 2013) introduced the MO-MAB framework and Pareto regret as a metric, achieving $O(\log T)$ Pareto regret using the UCB technique. Other techniques, including Knowledge Gradient (Yahyaa et al., 2014) and Thompson Sampling (Yahyaa & Manderick, 2015), have subsequently been adapted for MO-MAB. Additionally, researchers have extended the contextual setup to MO-MAB, where the action reward for each objective is modeled as a function of the input context and action (Turgay et al., 2018; Lu et al., 2019). These studies aim to efficiently approximate the entire Pareto front $\mathcal{O}^*$, and employ a *random arm selection policy* on the estimated Pareto front to achieve Pareto optimality. However, computing the full Pareto front is computationally expensive, leading to another line of work where multi-dimensional rewards are scalarized. This approach converts the multi-dimensional reward into a scalar value through a scalarization function, targeting a specific Pareto optimal solution without approximating the entire Pareto front. The scalarization function can either be randomly initialized (chosen) (Drugan & Nowe, 2013; Xu & Klabjan, 2023), or optimized based on a fixed metric, such as the Generalized Gini Index score (Busa-Fekete et al., 2017; Mehrotra et al., 2020). Nonetheless, existing studies primarily achieve Pareto optimality through a *global policy* for arm selection across all users. As discussed in Section 1, merely achieving Pareto optimality with a global policy may not yield favorable outcomes, as users have diverse preferences on different objectives. Therefore, customized MO-MAB optimization under user preferences is essential, which is the goal of our work.

**Preference-based MO-MAB optimization.** Recent studies have explored MO-MAB optimization using lexicographic order (Ehrgott, 2005) to reflect user preferences. In lexicographic order, objectives are prioritized hierarchically, where the first objective takes absolute precedence over the second, and so on. Hüyük & Tekin (2021) first introduced lexicographic order to MO-MAB, and Cheng et al. (2024) extended it to mixed Pareto-lexicographic environments. However, lexicographic order may not adequately capture a user's overall satisfaction in real-world applications, where preferences often involve trade-offs rather than strict prioritization. For example, a user may prefer a \$10 meal with good taste over a \$9.5 meal with poor taste, even though cost is a priority. Our work proposes a more general framework that incorporates a weighted order based on the user's explicit preference space. Notably, the lexicographic order becomes a special case of our proposed PAMO-MAB framework.

## 3 PROBLEM FORMULATION

We consider MO-MAB with $K$ arms and $D$ objectives. At each round $t \in [T]$, the learner chooses an arm $a_t$ to play and observes a stochastic $D$-dimensional *reward vector* $\boldsymbol{r}_{a_t,t} \in \mathcal{R} \subseteq \mathbb{R}^D$ for action $a_t$, which we refer to as *reward*. For the reward, we make the following standard assumption:

**Assumption 3.1** (Bounded stochastic reward). *For $i \in [K], t \in [T], d \in [D]$, each reward entry $\boldsymbol{r}_{i,t}(d)$ is independently drawn from a **fixed** but **unknown** distribution $\mathcal{F}^r_{i,d}$ with mean $\boldsymbol{\mu}_i(d)$ and variance $\sigma^2_{r,i,d}$, satisfying $\boldsymbol{r}_{i,t}(d) \in [0, 1]$, and $\sigma^2_{r,i,d} \in [\sigma^2_{r\downarrow}, \sigma^2_{r\uparrow}]$, where $\sigma^2_{r\downarrow}, \sigma^2_{r\uparrow} \in \mathbb{R}^+$.*

**User preferences.** At each round $t$, we consider the user to be associated with a stochastic $D$-dimensional *preference vector* $\boldsymbol{c}_t \in \mathcal{C} \subseteq \mathbb{R}^D$, indicating the user preferences across the $D$ objectives. We refer to this vector as *preference* for short. Specifically, we make the following assumptions:

**Assumption 3.2** (Bounded stochastic preference). *For $t \in [T], d \in [D]$, each preference entry $\boldsymbol{c}_t(d)$ is independently drawn from a **possibly dynamic** distribution $\mathcal{F}^c_{t,d}$ (**either known or unknown**) with mean $\overline{\boldsymbol{c}}_t(d)$ and variance $\sigma^2_{c,t,d}$, satisfying $\boldsymbol{c}_t(d) \geq 0$, $\|\boldsymbol{c}_t\|_1 \leq \delta$, $\sigma^2_{c,t,d} \in [0, \sigma^2_c]$.*

**Assumption 3.3** (Independence). *For $t \in [T]$, $i \in [K]$, $d_1, d_2 \in [D]$, $\boldsymbol{r}_{i,t}(d_1)$, $\boldsymbol{c}_t(d_2)$ are independent.*

Assumption 3.3 is common in real applications since $\boldsymbol{c}_t$ and $\boldsymbol{r}_t$ are inherently determined by independent factors: user characteristics and arm properties. For example, an individual user's preferences do not influence a restaurant's location, environment, pricing level, etc., and vice versa.

**Preference-aware reward.** We define an *overall-reward* as the inner product of arm's reward and user's preference, which is as a scalar and models the user reward rating under their preferences. Specifically, we refer to the inner product mapping $\Phi : \mathcal{C} \times \mathcal{R} \to \mathbb{R}$ as the *aggregation function*. In each round $t$, the overall-reward $g_{a_t,t}$ for the chosen arm $a_t$ is defined as:

$$g_{a_t,t} = \Phi(\boldsymbol{c}_t, \boldsymbol{r}_{a_t,t}) = \sum_{d \in [D]} \boldsymbol{c}_t(d) \cdot \boldsymbol{r}_{a_t,t}(d) = \boldsymbol{c}_t^T \boldsymbol{r}_{a_t,t}. \tag{1}$$

To evaluate the learner's performance, we define regret relative to a *possibly dynamic* oracle as the difference in expected overall-reward, i.e., the difference between the expected cumulative overall-reward by selecting the arm with the highest expected overall-reward at each time $t$ and the expected overall-reward under the learner's policy:

$$R(T) = \sum_{t=1}^{T} \left( \mathbb{E}[\Phi(\boldsymbol{c}_t, \boldsymbol{r}_{a_t^*,t})] - \mathbb{E}[\Phi(\boldsymbol{c}_t, \boldsymbol{r}_{a_t,t})] \right) = \sum_{t=1}^{T} \overline{\boldsymbol{c}}_t^T (\boldsymbol{\mu}_{a_t^*} - \boldsymbol{\mu}_{a_t}) \tag{2}$$

where $a_t^* = \arg\max_{i \in [K]} \mathbb{E}[\Phi(\boldsymbol{c}_t, \boldsymbol{r}_{i,t})]$ refers to the best arm at round $t$. The goal is to minimize the cumulative regret $R(T)$. We term this problem as **Preference-Aware** MO-MAB (PAMO-MAB).

**Remark 3.1.** *Despite the linear model of overall reward, PAMO-MAB differs fundamentally from linear (contextual) bandits (Abbasi-Yadkori et al., 2011; Chu et al., 2011) for the following reasons:*
- *In linear bandits, the input features are observable before making decisions, whereas in PAMO-MAB, both the random reward and preference can be unknown and must be estimated.*
- *In linear bandits, the feedback is a scalar reward, whereas in PAMO-MAB, the feedback can take on various forms: a $D$-dimensional reward, a $D$-dimensional reward with a $D$-dimensional preference, or a $D$-dimensional reward with an overall-reward, depending on the interaction protocols.*

## 4 A LOWER BOUND

In the following, we develop a lower bound (Proposition 1) on the defined regret for PAMO-MAB. Such a lower bound will quantify how difficult it is to control regret without preference-adaptive policies under PAMO-MAB. Firstly, we present a definition characterizing a class of MO-MAB algorithms of which the sequential decision-making is independent of the preference information.

**Definition 1** (Preference-Free Algorithm). *Let $\mathbf{c}^t = \{\boldsymbol{c}_1, \boldsymbol{c}_2, ..., \boldsymbol{c}_t\} \in \mathbb{R}^{D \times t}$ and $\overline{\mathbf{c}}^t = \{\overline{\boldsymbol{c}}_1, \overline{\boldsymbol{c}}_2, ..., \overline{\boldsymbol{c}}_t\} \in \mathbb{R}^{D \times t}$ be the preference sequence and the sequence of corresponding mean vectors up to $t$ episodes. Let $\pi_t^{\mathcal{A}}$ be the policy of algorithm $\mathcal{A}$ at time $t$ for selecting arm $a_t$ in a PAMO-MAB problem. Then $\mathcal{A}$ is defined as a preference-free algorithm if its policy $\pi_t^{\mathcal{A}}$ is independent of $\mathbf{c}^t$ and $\overline{\mathbf{c}}^t$, i.e., $\mathbb{P}_{\pi_t^{\mathcal{A}}}(a_t = i | \mathbf{c}^t, \overline{\mathbf{c}}^t) = \mathbb{P}_{\pi_t^{\mathcal{A}}}(a_t = i)$ for all arms $i \in [K]$ and all episodes $t \in (0, T]$.*

To our knowledge, most existing algorithms in theoretical MO-MAB studies (Drugan & Nowe, 2013; Busa-Fekete et al., 2017; Xu & Klabjan, 2023; Hüyük & Tekin, 2021; Cheng et al., 2024) fall within the class of preference-free algorithms, which employ a global policy for arm selection, while neglecting users' preferences—an essential feature commonly observed in practical applications.

**Proposition 1.** *Assume an MO-MAB environment contains multiple objective-conflicting arms, i.e., $|\mathcal{O}^*| \geq 2$, where $\mathcal{O}^*$ is the Pareto Optimal front. Then, for any preference-free algorithm, there exists a subset of preference such that the regret $R(T) = \Omega(T)$.*

Proposition 1 shows that for the PAMO-MAB problem with $|\mathcal{O}^*| \geq 2$, sub-linear regret is no longer achievable for preference-free algorithms. The reason is that for any arm $i \in \mathcal{O}^*$ that is optimal in one preference subset $\mathcal{C}^+$, there exists another preference subset $\mathcal{C}^-$ where arm $i$ becomes suboptimal. However, preference-free algorithms cannot adapt their policies to different sets of preferences, and thus fail to consistently perform optimally across the entire preference space $\mathcal{C}$. Please see Appendix B for the detailed proof of Proposition 1. We therefore ask the following question: ***Can we design preference-adaptive algorithms that achieve sub-linear regret for PAMO-MAB?*** The answer is **yes.** In the following, we conduct a comprehensive analysis of PAMO-MAB under three structures, considering both *prior-known* and *unknown* preference environments. We demonstrate that through preference adaptation, the algorithms can achieve sub-linear regret.

## 5 THE CASE WHEN THE PREFERENCE IS KNOWN

We begin with the simpler case where the learner knows the user's expected preferences before arm selection, as a warm-up for understanding the structure of the problem. Formally, at each round $t$, the learner obtains $\overline{c}_t \in \mathbb{R}^D$ from user's input and selects an arm $a_t \in [K]$, then observes $r_{a_t,t} \in \mathbb{R}^D$. This setup is inspired by numerous real-world applications. In personalized recommender, systems are typically informed of user preferences (e.g., quality, price, style) before recommendation. Many online systems now enable users to express their preferences before decision-making through interactive techniques such as conversations,

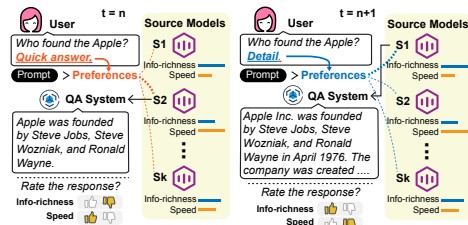

Figure 2: User expressing her expected preferences to QA system by customizing input prompts before source language model selection.

prompt design, keyword search, and more. An example is shown in Fig. 2, where the user personalizes the prompt input, allowing for the adaptive selection of the source model in a QA system.

To this end, we propose a novel Preference-UCB (PRUCB) algorithm, presented in Algorithm 1. At a high level, Algorithm 1 is an extension of the UCB approach (Auer et al., 2002) for PAMO-MAB. As discussed in Section 4, it is crucial for the learner to adapt to user preferences; otherwise, sub-linear regret is unattainable. To address this, we introduce two key designs in PRUCB as follows:

**Preference estimation.** Capturing user preferences is a fundamental step toward preference adaptation. In this case, since the expected preference is known in advance, we can trivially leverage this information as the preference estimation: $\hat{c}_t \leftarrow \overline{c}_t$. However, we still emphasize that this mechanism is crucial, as in the unknown preference scenarios explored in Section 6 and Section 7, preference estimation must be carefully designed.

**Preference-aware optimization.** To enable the policy to adapt to the estimated preference $\hat{c}_t$, and following the "optimism in the face of uncertainty" principle (Auer et al., 2002), the arm selection policy of PRUCB at each round $t$ is designed as:

$$a_t = \arg\max_{i \in [K]} \Phi(\hat{c}_t, \hat{r}_{i,t} + \sqrt{\frac{\log(t/\alpha)}{\max\{1, N_{i,t}\}}} e), \tag{3}$$

where $\Phi(\cdot, \cdot)$ is the aggregation function defined in Eq. 1, and $N_{i,t} = \sum_{j=1}^{t-1} \mathbb{1}_{\{a_j = i\}}$ is the number of

---

**Algorithm 1** Preference UCB (PRUCB)

1: **Parameters:** $\alpha$.
2: **Initialization:** $N_{i,1} \leftarrow 0$; $\hat{r}_{i,1} \leftarrow [0]^D, \forall i \in [K]$.
3: **for** $t = 1, \cdots, T$ **do**
4:     Obtain user expected preference $\overline{c}_t$, $\hat{c}_t \leftarrow \overline{c}_t$.
                     ▷ (Preference estimation)
5:     Draw $a_t$ by Eq. 3, observe reward $r_{a_t,t}$.
                ▷ (Preference-aware optimization)
6:     Update $N_{i,t+1}$ and $\hat{r}_{i,t+1}, \forall i \in [K]$ by Eq.4.
                   ▷ (Reward estimation)
7: **end for**

---

pulls of arm $i$ within the first $t-1$ rounds. $\hat{r}_{i,t}$ is reward estimation of arm $i$, with a bonus vector $\sqrt{\log(t/\alpha)/N_{i,t}} e$ to strikes a balance between exploration and exploitation, where $\alpha \in (0, 1]$ is an algorithm hyper-parameter. For $t \in [2, T]$ and $i \in [K]$, $N_{i,t}$ and $\hat{r}_{i,t}$ are updated as follows:

$$N_{i,t} = N_{i,t-1} + \mathbb{1}_{\{a_{t-1}=i\}}, \quad \hat{r}_{i,t} = \frac{\hat{r}_{i,t-1}N_{i,t-1} + r_{a_{t-1},t-1} \cdot \mathbb{1}_{\{a_{t-1}=i\}}}{N_{i,t}}, \tag{4}$$

with $N_{i,1} \leftarrow 0, \hat{r}_{i,1} \leftarrow [0]^D, \forall i \in [K]$. In a nutshell, PRUCB models the user preference and arm rewards simultaneously by updating $\hat{c}_t$ and $\hat{r}_t$, then leverages this knowledge to formulate the upper confidence bound (UCB) of the overall-reward through the aggregation function $\Phi$. In this way, PRUCB elegantly transforms the problem into maximizing the UCB of the estimated overall-reward under the estimates of preference $\hat{c}_t$ and reward $\hat{r}_t$, achieving preference-awareness. Building upon these two major components, we summarize the main PRUCB algorithm in Algorithm 1. The regret is characterized in Theorem 2 below.

**Theorem 2.** *Assuming $\boldsymbol{c}_t \in \mathbb{R}^D$ follows (possibly dynamic) distribution with expectation vector $\overline{\boldsymbol{c}}_t$ known before decision making, then for any $\alpha \in (0, 1]$, the regret of PRUCB is upper-bounded as*

$$R(T) \le \sum_{i=1}^{K} \left( \frac{4\delta^2 \eta_i^\uparrow \log(\frac{T}{\alpha})}{\eta_i^{\downarrow 2}} + \frac{D\pi^2 \alpha^2 \eta_i^\uparrow}{3} \right) = O(\delta \log T)^1$$

*where $\eta_i^\uparrow = \max_{t \in \mathcal{T}_i} \{\overline{\boldsymbol{c}}_t^T \Delta_{i,t}\}$, $\eta_i^\downarrow = \min_{t \in \mathcal{T}_i} \{\overline{\boldsymbol{c}}_t^T \Delta_{i,t}\}$, $\mathcal{T}_i = \{t \in [T] \mid a_t^* \ne i\}$ is the set of episodes when arm $i$ is suboptimal, $\Delta_{i,t} = \mu_{a_t^*} - \mu_i \in \mathbb{R}^D, \forall t \in [T]$.*

The proof of Theorem 2 is provided in Appendix C.1. Particularly, Theorem 2 demonstrates the benefit of introduced preference estimation and preference-aware optimization mechanisms, achieving the near-optimal regret (on the order of $O(\log T)$) for PAMO-MAB problem

## 6 THE CASE WHEN THE PREFERENCE IS UNKNOWN

In this section, we explore a more challenging scenario where, at each round $t$, the user preference $\boldsymbol{c}_t$ is unknown and only revealed after action $a_t$ is taken, along with the reward $\boldsymbol{r}_{a_t}$. This protocol is common in practical applications. Fig. 3 illustrates an example where a user on a streaming platform (e.g., TikTok) refreshes for a new video list, and the system selects a source model for recommending new videos. If the recommender selects a source model with good empirical recommendation performance (e.g., click-through rate) but low efficiency, the user may refresh again or close the app during content loading. This behav-

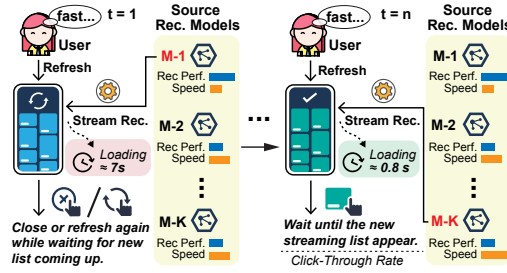

Figure 3: A scenario of user indicating her instantaneous preferences after arm pulling.

ior suggests that the user might have a stronger preference for efficiency over content quality. Such preference information can only be obtained after taking the action (i.e., selecting the source model). We begin with the case where the preference $\boldsymbol{c}_t$ follows a fixed distribution, and then extend the analysis to a more complex yet more practical scenario where the preference distribution is dynamic.

### 6.1 STATIONARY PREFERENCE

For the unknown preference case, the inaccessibility of the true preference expectation $\overline{\boldsymbol{c}}$ raises two fundamental questions for algorithm design: *1) how to estimate the unknown preferences via feedback? 2) how to handle the uncertainty of preference estimation in decision-making?* To this end, we advance PRUCB into PRUCB-SPM and elaborate on the key designs involved as follows.

**Preference estimation.** Due to the unknown expected preference, directly using $\overline{\boldsymbol{c}}$ as the modeled $\hat{\boldsymbol{c}}$ is no longer feasible. To resolve this issue, we leverage the empirical average of preference feedback as the preference estimate. For $t \in [2, T]$, PRUCB-SPM updates preference estimate as

$$\hat{\boldsymbol{c}}_t = \frac{(t-2)\hat{\boldsymbol{c}}_{t-1} + \boldsymbol{c}_{t-1}}{t-1}. \quad (5)$$

**Preference-aware optimization.** Since the reward environment remains the same as in Section 5, for all $i \in [K]$, we follow Eq. 4

---

**Algorithm 2** Preference UCB with Stationary Preference estimation (PRUCB-SPM)

1: **Parameters:** $\alpha$.
2: $N_{i,1} \leftarrow 0, \hat{\boldsymbol{r}}_{i,1} \leftarrow [0]^D, \forall i \in [K]; \hat{\boldsymbol{c}}_1 \leftarrow [0]^D$.
3: **for** $t = 1, \cdots, T$ **do**
4:     Draw arm $a_t$ by Eq. 6, observe reward $\boldsymbol{r}_{a_t,t}$ and user preference $\boldsymbol{c}_t$. ▷ (Preference-aware optimization)
5:     Update $N_{i,t+1}$ and reward estimate $\hat{\boldsymbol{r}}_{i,t+1}, \forall i \in [K]$ by Eq. 4. ▷ (Reward estimation)
6:     Update preference estimate $\hat{\boldsymbol{c}}_{t+1}$ by Eq.5. ▷ (Preference estimation)
7: **end for**

---

for the updating of $N_{i,t}$ and reward estimation $\hat{\boldsymbol{r}}_{i,t}$. Based on the estimated $\hat{\boldsymbol{c}}_t$ and $\hat{\boldsymbol{r}}_t$, we can construct a preference-aware optimization measure, analogous to PRUCB. However, the unknown preference introduces two new challenges in the preference-aware optimization measure design:

- The updated preference estimate could deviate from the true expectation. An intuitive approach might involve constructing a confidence region $\Theta_t$ for $\hat{\boldsymbol{c}}_t$, similar to the reward estimation $\hat{\boldsymbol{r}}_t$. The solution would then be to choose the pair $(a_t, \hat{\boldsymbol{c}}_t') \in [K] \times \Theta_t$ that jointly maximizes the UCB of the overall-reward, i.e., $a_t = \arg\max_{i \in [K]} \max_{\hat{\boldsymbol{c}}_t' \in \Theta_t} \Phi(\hat{\boldsymbol{c}}_t', \hat{\boldsymbol{r}}_{i,t} + \sqrt{\log(t/\alpha)/N_{i,t}}\boldsymbol{e})$. However,

---

[1] We consider $\|\overline{\boldsymbol{c}}_t\|_1 = \Theta(\delta), \eta_i^\downarrow = \Theta(\eta_i^\uparrow)$, thus simplify $\delta^2 \eta_i^\uparrow / \eta_i^{\downarrow 2} \le C\delta^2/(\overline{\boldsymbol{c}}_t^T \Delta_{i,t}) = C\delta/((\overline{\boldsymbol{c}}_t/\delta)^T \Delta_{i,t}) = C\delta/(\overline{\boldsymbol{c}}_t'^T \Delta_i) = O(\delta)$, where $\overline{\boldsymbol{c}}_t' = \Theta(1)\boldsymbol{e}$ is the $\delta$-scale normed preferences, $C = \Theta(1)$ satisfies $\eta_i^\uparrow \le C\eta_i^\downarrow$.

in this case, *a confidence region for the preference estimate $\hat{c}_t$ is unnecessary*. The fundamental reason is that preference estimation does not involve *sequential action decision-making* component. Specifically, at each round $t$, the preference feedback $c_t$ is observed with certainty after arm pulling and is independent of the chosen action $a_t$. Thus, the empirical average suffices, as $\hat{c}_t$ will converge to the true mean $\overline{c}$ over time by law of large numbers, whereas additional exploration is unnecessary. In contrast, for reward estimation, the action $a_t$ determined by $\hat{r}_t$ will also influence the future estimate $\hat{r}_{t+1}$. In this context, adding a confidence term is necessary to avoid overconfidence in the estimates and encourage the exploration of different arms, improving future decision-making.

- Another concern is whether the confidence width $\sqrt{\log(t/\alpha)/N_{i,t}}$ for $\hat{r}_{i,t}$ in known preference case remains feasible in unknown case. Errors in preference estimation can propagate to reward estimation. Specifically, imprecise preference estimation can lead to inaccurate overall-reward UCB estimation, resulting in misguided exploitation. This, in turn, affects reward estimation, as it depends on the arms selected. Despite this, we show that *the confidence width of $\sqrt{\log(t/\alpha)/N_{i,t}}$ for the reward estimate suffices to control the regret*, as preference estimation benefits from higher learning efficiency due to higher sampling rate compared to reward estimation of each arm. Thus, the impact of imprecise $\hat{c}_t$ on the estimation of $\hat{r}_t$ becomes negligible as $t$ increases.

Building upon the analysis above, the arm selection policy of PRUCB-SPM is designed as:

$$a_t = \arg\max_{i \in [K]} \Phi(\hat{c}_t, \hat{r}_{i,t} + \sqrt{\log(t/\alpha)/\max\{1, N_{i,t}\}}e). \tag{6}$$

We characterize the regret upper-bound of PRUCB-SPM in Theorem 3. Note that in the stationary preference case, we omit the subscript of $t$ in $a_t^*, \Delta_{i,t}$ for simplicity, as they are independent of $t$.

**Theorem 3.** *Assume the preference follows unknown fixed distribution with the value being revealed after each arm pull. Let $\eta_i = \overline{c}^T \Delta_i$, $\Delta_i = \mu_{a^*} - \mu_i \in \mathbb{R}^D$, PRUCB-SPM has*

$$R(T) \leq \sum_{i \neq a^*} \Big( \underbrace{\frac{4(\delta + \frac{\delta}{\sqrt{D}})^2 \log(\frac{T}{\alpha})}{\eta_i} + \frac{D\pi^2 \alpha^2 \eta_i}{3}}_{R^r(T): \text{ Regret caused by } \textbf{reward estimation} \text{ error}} + \underbrace{\frac{4\sqrt{2}(D\delta \|\Delta_i\|_2)^{2.5}}{\eta_i^{1.5}} + \frac{D\pi^2 \eta_i}{3}}_{R^c(T): \text{ Regret caused by } \textbf{preference estimation} \text{ error}} \Big). \tag{7}$$

**Remark 6.1.** *Theorem 3 shows that, without known user preferences, PRUCB-SPM achieves a regret of $\mathcal{O}(\delta \log T)$, demonstrating near-optimal performance. Notably, the regret caused by additional preference estimation error is bounded by a constant related to objective dimension $D$ and $\ell_1$-norm bound $\delta$ of preference. Furthermore, the dominant regret term, caused by reward estimation error, degrades performance by only a factor of $(1 + 1/\sqrt{D})^2$ compared to the known-preference case. This implies that the impact of additional preference estimation error on the final regret is small.*

To prove Theorem 3, the main difficulty lies in decoupling and capturing the effects of the joint error from both reward estimation and preference estimation on the final regret. To address this, we introduce a tunable parameter $\epsilon_t$ to quantify the accuracy of preference estimation $\hat{c}_t$, and decompose suboptimal actions into two disjoint sets, accounting for two regret terms of $R^r(T)$ and $R^c(T)$ in Eq 7. The derivation of $R^r(T)$ relies on Proposition 8 in Appendix C.1, which characterizes the policy behavior under accurate preference estimation updates. The derivation of $R^c(T)$ relies on Lemma 10 in Appendix D.1.2 to transfer the original set with joint error to a preference estimation deviation event, making it more tractable. Please refer to Appendix D.1 for the full proof of Theorem 3.

**Corrupted Preference?** The potential limitation of the above result is that, in some applications, precise user preference feedback may not be obtainable. For example, in Figure 3, the system infers user preferences (efficiency vs. quality) from action logs rather than explicit user feedback, which can introduce *corruption* into the preference estimation. Therefore, we further explore the performance of PRUCB-SPM under corrupted preference feedback. Building on the assumptions in Theorem 3, we define the observed preference feedback as being manipulated by stochastic corruption: $\tilde{c}_t = c_t + z_t$, where $c_t$ is the true preference, $\tilde{c}_t$ is the observed (corrupted) feedback, and $z_t \in \mathbb{R}^D$ is the stochastic corruption component. For $d \in [D]$, $z_t(d)$ is independently drawn from a fixed distribution with mean $\overline{z}(d)$ and variance $\sigma_{z,d}^2 \leq \sigma_z^2$. We use $\|\overline{z}\|_2$ to denote the level of stochastic corruption.

The following Theorem 4 characterizes the regret and robustness of PRUCB-SPM (Algorithm 2) under stochastic preference corruptions. The proof is provided in Appendix D.2.

**Theorem 4.** *Inherit the assumptions in Theorem 3, but assume that the observed preference feedback is under stochastic corruption. Let $B_i = \frac{\eta_i}{1 + \frac{1}{D}} - \|\overline{z}\|_2 \|\Delta_i\|_2$, $\eta_i = \overline{c}^T \Delta_i$. Then PRUCB-SPM has*

①  *if $\exists i \neq a^*$, s.t., $B_i \leq 0$, then $R(T) = \Omega(T)$;* ②  *else if $B_i > 0, \forall i \neq a^*$, then*

$$R(T) \leq \sum_{i \neq a^*} \Big( \frac{4(D+1)^2 \delta^2 \log(\frac{T}{\alpha})}{\eta_i} + \frac{D\pi^2 \alpha^2 \eta_i}{3} + \frac{4D^2 \eta_i \|\Delta_i\|_2^2 (\sigma_c^2 + \sigma_z^2)}{B_i^2} + \frac{4D^{1.5} \eta_i \|\Delta_i\|_2 (\delta + \delta_z)}{3B_i} \Big).$$

Theorem 4 shows as long as the corruption level satisfies the attack tolerance threshold of $B_i > 0, \forall i \neq a^*$, PRUCB-SPM attains an $O(D^2 \delta \log T)$ regret, implying its robustness. Moreover, our analysis of adversarial corruption case also demonstrates the robustness of PRUCB-SPM against adversarial attack up to a corruption level of $o(T)$. See Appendix D.3 for the detailed analysis.

## 6.2 Non-Stationary Preference

In this section, we consider *abruptly changing environments*, a more practical scenario in real-world applications. Building on the assumptions of Theorem 3, we assume that the preference distribution $c_t$ remains fixed during periods but changes at unknown time instants called *breakpoints*. The number of breakpoints within $T$ is denoted by $\psi_T$. Unlike the stationary preference case, the challenge here is that the empirical estimate $\hat{c}_t$ by Eq. 5 becomes a biased estimator of the expected preference $\bar{c}_t$ due to the time-varying distribution. To address this, we propose PRUCB-APM (Algorithm 3).

Specifically, inspired by the sliding-window UCB (Garivier & Moulines, 2008), we consider averaging recent observations over a fixed horizon for user preference estimation, rather than averaging observations over all past rounds. Formally, at round $t \in [2, T]$, PRUCB-APM updates the preference estimate by computing a local empirical average using the last $\tau$ plays:

$$\hat{c}_t = \frac{1}{\min\{\tau, t-1\}} \sum_{\ell=\max\{1, t-\tau\}}^{t-1} c_\ell, \quad (8)$$

where $\tau$ is an algorithm parameter denoting the sliding-window length. The sliding-window estimator removes outdated samples and retains recent ones, enabling it to track the latest preference patterns. For reward estimation and preference-aware optimization, we follow the Eq. 4 and Eq. 6.

---

**Algorithm 3** Preference UCB with Abrupt Preference estimation (PRUCB-APM)

1: **Parameters:** $\alpha$. Sliding-window length $\tau$.
2: $N_{i,1} \leftarrow 0; \hat{r}_{i,1} \leftarrow [0]^D, \forall i \in [K]; \hat{c}_1 \leftarrow [0]^D$.
3: **for** $t = 1, \cdots, T$ **do**
4:     Draw arm $a_t$ by Eq. 6, observe $r_{a_t, t}$ and user's preference $c_t$.   ▷ (Preference-aware optimization)
5:     Update $N_{i,t+1}$, and reward estimate $\hat{r}_{i,t+1}$, $\forall i \in [K]$ by Eq. 4.   ▷ (Reward estimation)
6:     Update preference estimate $\hat{c}_{t+1}$ by Eq. 8.   ▷ (Preference estimation)
7: **end for**

---

In Theorem 5 below, we characterize the regret of PRUCB-APM, and show that it is controlled by $\tau$. Please refer to Appendix D.4 for the proof sketch and detailed proof steps of Theorem 5.

**Theorem 5.** *Inherit the assumptions in Theorem 3 but assume $c_t$ follows abruptly changing distribution. Let $\mathcal{T}_i = \{t \in [T] \mid a_t^* \neq i\}$, $\eta_i^\downarrow = \min_{t \in \mathcal{T}_i}\{\bar{c}_t^T \Delta_{i,t}\}$ and $\eta_i^\uparrow = \max_{t \in \mathcal{T}_i}\{c_t^T \Delta_{i,t}\}$. $\Delta_{i,t} = \mu_{a_t^*} - \mu_{i,t} \in \mathbb{R}^D$, $a_t^*$ is the dynamic oracle. $\|\Delta_i^\uparrow\|_2 = \max_{\{t,j\} \in [T] \times [K]/i} \|\mu_{i,t} - \mu_{j,t}\|_2$. Then for any $\tau > \max_{i \in [K]} (2D\delta \|\Delta_i^\uparrow\|_2 / \eta_i^\downarrow)^{\frac{5}{2}}$, any $\alpha \in (0, 1]$, PRUCB-APM follows*

$$R(T) \leq \sum_{i=1}^{K} \eta_i^\uparrow \Big( \frac{4(\delta + \frac{\delta}{\sqrt{D}})^2 \log(T/\alpha)}{(\eta_i^\downarrow)^2} + D\frac{\pi^2 \alpha^2}{3} + \psi_T \tau + \frac{2D(T-\tau)}{\tau^2} + \Big(\frac{2D\delta \|\Delta_i^\uparrow\|_2}{\eta_i^\downarrow}\Big)^{\frac{5}{2}} + \frac{D\pi^2}{3} \Big),$$

**Remark 6.2.** *If the horizon $T$ and the number of breakpoints $\psi_T$ are known in advance, the window size $\tau$ can be chosen to minimize $R(T)$. Specifically, taking $\tau = (4DT/\psi_T)^{\frac{1}{3}}$ yields $R(T) = O(\delta \log(T) + D^{\frac{1}{3}} \psi_T^{\frac{2}{3}} T^{\frac{1}{3}})$. Assuming that $\psi_T = O(T^\gamma)$ for some $\gamma \in [0, 1)$, then we have $R(T)$ is dominant with order of $\mathcal{O}(T^{(1+2\gamma)/3})$. In particular, if $\gamma = 0$, $R(T) = O(\delta \log(T) + D^{\frac{1}{3}} T^{\frac{1}{3}})$.*

**Remark 6.3.** *If there is no breakpoint, i.e., $\psi_T = 0$, the problem reduces to the stationary preference case. In this case, the optimal window length $\tau$ is obviously $T$ (as large as possible), and $\eta_i^\uparrow = \eta_i^\downarrow$. Plugging these back to Theorem 5 yields the regret that matches the result obtained in Theorem 3, indicating Theorem 5 is an effective generalization of Theorem 3.*

## 7 The Case with Hidden Preference

Finally, we consider another practical scenario where only feedback on the reward and overall reward is observable, while preference feedback is not provided. For instance, in hotel surveys, customers often provide ratings on specific objectives (e.g., price, location, environment, amenities) along with an overall rating (as depicted in Fig. 4). In such cases, user preferences can be inferred from the latent relationship between the overall rating and the individual objective ratings. Formally, in each round $t$, the learner

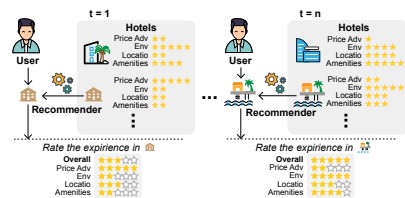

Figure 4: A scenario of user's preferences feedback is not provided.

selects an arm $a_t \in [K]$, and observes the reward vector $\boldsymbol{r}_{a_t} \in \mathbb{R}^D$, as well as the overall-reward score $g_{a_t,t} = \Phi(\boldsymbol{c}_t, \boldsymbol{r}_{a_t,t}) = \boldsymbol{c}_t^T \boldsymbol{r}_{a_t,t} \in \mathbb{R}$ corresponding to the selected action. The preference $\boldsymbol{c}_t \in \mathbb{R}^D$ is *stationary* and follows an *unknown distribution*.

Given this framework, we adhere to the original Assumption 3.1 on rewards. Note in many real-world applications, such as hotel rating systems, the overall rating shares the same scale as individual objective ratings. Thus, we introduce Assumption 7.1, where the bound on the overall reward is identical to that of the reward. This, in turn, leads to a revised Assumption 7.2 on preference.

**Assumption 7.1.** *For $t \in [T]$, $a_t \in [K]$, the overall-reward score satisfies $g_{a_t,t} \in [0,1]$.*

**Assumption 7.2.** *For $t \in [T], d \in [D]$, preference satisfies $\boldsymbol{c}_t(d) \in [0,1]$ and $\|\boldsymbol{c}_t\|_1 \leq 1$.*

To address this problem, we propose a novel PRUCB-HPM (see Algorithm 4). The fundamentally different preference structure with Section 6 introduces new challenges, which we discuss below.

**Preference estimation.** Due to the absence of preference feedback, we can only infer user preference knowledge through the latent relationship from rewards $\boldsymbol{r}_{a_t,t}$ and overall-rewards $g_{a_t,t}$. Recall that the overall-reward is the inner product of preference and reward, it becomes natural to estimate the latent preference by regression based on previous rewards and overall-rewards. While regression-based coefficient estimation has been widely used in linear (contextual) bandits works (Abbasi-Yadkori et al., 2011; Zhao et al., 2020; Hanna et al., 2024), designing preference estimation by regression in our case is non-

---

**Algorithm 4** Preference UCB with Hidden Preference estimation (PRUCB-HPM)

1: **Parameters:** $\alpha$, $\lambda$, $\beta_t$.
2: $\hat{\boldsymbol{r}}_{i,1} \leftarrow [0]^D$, $N_{i,1} \leftarrow 0, \forall i \in [K]$, $\hat{\boldsymbol{c}}_1 \leftarrow [\frac{1}{D}]^D$, $\Upsilon_1 \leftarrow \lambda \boldsymbol{I}$,
   $\Theta_1 \leftarrow \{\boldsymbol{c}' | (\boldsymbol{c}' - \hat{\boldsymbol{c}}_1)^T \Upsilon_1 (\boldsymbol{c}' - \hat{\boldsymbol{c}}_1) \leq \beta_1 \wedge \|\boldsymbol{c}'\|_1 \leq 1\}$.
3: **for** $t = 1, \cdots, T$ **do**
4:    Draw arm $a_t$ by Eq.11, observe reward $\boldsymbol{r}_{a_t,t}$ and overall-reward $g_{a_t,t}$. ▷ (Preference-aware optimization)
5:    Update $N_{i,t+1}$, and rewards estimation $\hat{\boldsymbol{r}}_{i,t+1}, \forall i \in [K]$ by Eq. 4. ▷ (Reward estimation)
6:    Update $\Upsilon_{t+1}$ and latent preference estimation $\hat{\boldsymbol{c}}_{t+1}$ by Eq.9. ▷ (Preference estimation)
7:    Update preference confidence ellipse $\Theta_{t+1}$ by Eq.10.
8: **end for**

---

trivial due to the fundamentally different setting. Specifically, in our scenario, the latent coefficient (preference) vector $\boldsymbol{c}_t$ is random in each round $t$, unlike the fixed coefficients in linear bandit literature. The regression model can be written as $g_{a_t,t} = (\overline{\boldsymbol{c}} + \boldsymbol{\zeta}_t)^T \boldsymbol{r}_{a_t,t} = \overline{\boldsymbol{c}}^T \boldsymbol{r}_{a_t,t} + \boldsymbol{\zeta}_t^T \boldsymbol{r}_{a_t,t}$, where $\boldsymbol{\zeta}_t = \boldsymbol{c}_t - \overline{\boldsymbol{c}} \in \mathbb{R}^D$ is an independent random noise term. Note we condition on all observed variables up to round $t$, so that $g_{a_t,t}$ and $\boldsymbol{r}_{a_t,t}$ are deterministic. This model implies that the noise term $\boldsymbol{\zeta}_t^T \boldsymbol{r}_{a_t,t}$ on output $g_{a_t,t}$ is no longer independent of the input $\boldsymbol{r}_{a_t,t}$. Intuitively, the standard regression models are not applicable here due to the violated assumption of noise of output being independent of the input, whereas the errors-in-variables methods (e.g., Deming regression) would be preferred.

However, we assert that *standard regression remains feasible for preference estimation in this problem*. Thanks to the fact that $\mathbb{E}[\boldsymbol{\zeta}_t] = \mathbb{E}[\boldsymbol{c}_t] - \mathbb{E}[\overline{\boldsymbol{c}}] = [0]^D$, we have $\mathbb{E}[g_{a_t,t}] = \mathbb{E}[\overline{\boldsymbol{c}}^T \boldsymbol{r}_{a_t,t}] + \mathbb{E}[\boldsymbol{\zeta}_t^T \boldsymbol{r}_{a_t,t}] = \overline{\boldsymbol{c}}^T \boldsymbol{r}_{a_t,t}$, implying the noise term $\boldsymbol{\zeta}_t^T \boldsymbol{r}_{a_t,t}$ vanishes in expectation, and the model behaves like a standard linear regression model in expectation. This suggests that, in expectation, the noise does not systematically bias the model. Hence, in PRUCB-HPM, we estimate the latent preference by solving a ridge regression problem: $\hat{\boldsymbol{c}}_t = \arg\min_{\boldsymbol{c}'} \sum_{\ell=1}^{t-1} (\boldsymbol{c}'^T \boldsymbol{r}_{a_t,\ell} - g_{a_t,\ell})^2 + \lambda \|\boldsymbol{c}'\|_2^2$, where $\lambda \geq 0$ is a regularization parameter of Algorithm 4 to reduce overfitting and handle the variance introduced by $\boldsymbol{\zeta}_t^T \boldsymbol{r}_{a_t,t}$. Above equation yields a close form solution as follows:

$$\hat{\boldsymbol{c}}_t = \Upsilon_t^{-1} \sum_{\ell=1}^{t-1} g_{a_\ell,\ell} \boldsymbol{r}_{a_\ell,\ell}, \quad \Upsilon_t = \Upsilon_{t-1} + \boldsymbol{r}_{a_{t-1},t-1} \boldsymbol{r}_{a_{t-1},t-1}^T, \text{and } \Upsilon_1 = \lambda \boldsymbol{I} \quad (9)$$

**Preference-aware optimization.** Next, we adopt the principle of "optimism in the face of uncertainty" for arm selection. It is important to note that in this case, *constructing a confidence set for the preference estimate $\hat{\boldsymbol{c}}_t$ is necessary*, as $\hat{\boldsymbol{c}}_t$ is now involved in the *sequential decision-making* process. More specifically, the selection of arm $a_t$ depends on $\hat{\boldsymbol{c}}_t$, while the future estimate $\hat{\boldsymbol{c}}_{t+1}$ is inferred from observations of $\{\boldsymbol{r}_{a_\ell,\ell}\}_{\ell=1}^t$ and $\{g_{a_\ell,\ell}\}_{\ell=1}^t$, which are dependent on actions $a_t$ in turn. Therefore, we define the confidence set for the preference estimation as a constrained ellipse:

$$\Theta_t = \{\boldsymbol{c}' \mid (\boldsymbol{c}' - \hat{\boldsymbol{c}}_t)^T \Upsilon_t (\boldsymbol{c}' - \hat{\boldsymbol{c}}_t) \leq \beta_t \wedge \|\boldsymbol{c}'\|_1 \leq 1\}, \quad (10)$$

where $\beta_t > 1$ is an algorithm parameter that increases with $t$. Inspired by prior linear bandit studies (Abbasi-Yadkori et al., 2011; He et al., 2022), we set $\beta_t = \tilde{O}(D)$ [2] in our problem and show that, for

---

[2]We use the notation $\tilde{O}$ to suppress dependence on logarithmic factors of T

| Preference | Known | | Unknown | | | Hidden |
|---|---|---|---|---|---|---|
| Stationary | ✓ | ✗ | ✓ | ✓ | ✗ | ✓ |
| Notification / Feedback | ✓ | ✓ | ✓ | ✓ | ✓ | ✗ |
| Corrupted | ✗ | ✗ | ✗ | ✓ | ✗ | ✗ |
| Algorithm | Algorithm 1 | | Algorithm 2 | | Algorithm 3 | Algorithm 4 |
| Regret | $O(\delta \log T)$ (Theorem 2) | $O(\delta \log T)$ (Theorem 3) | $O(D^2 \delta \log T)$ if $B_i > 0, \forall i \neq a^*$ (Theorem 4) | | $O(D^{\frac{1}{3}} \psi_T^{\frac{2}{3}} T^{\frac{1}{3}})$ (Theorem 5) | $\tilde{O}(D\sqrt{T})$ (Theorem 6) |

Table 1: Summery of our main analytical results of PAMO-MAB problem under different preference structures.

regression under stochastic coefficients (preferences), $\overline{c} \in \Theta_t$ holds with high probability (please see detailed analysis of Proposition 14 in Appendix E.1). The reward estimation $\hat{r}_{i,t}, \forall i \in [K]$ follows Eq. 4. At each round $t$, the learner selects the arm $a_t$ by solving the joint optimization problem as:

$$a_t = \arg\max_{i \in [K]} \max_{c' \in \Theta_t} \Phi(c', \hat{r}_{i,t} + \sqrt{\log(t/\alpha)/\max\{N_{i,t}, 1\}} e). \tag{11}$$

**Theorem 6.** *Let preference $c_t$ follows unknown stationary distribution, and only over-reward and reward feedback is provided. For any $\lambda > 0$, by setting $\sqrt{\beta_t} = \sqrt{\lambda} + \sqrt{D \log\left(1 + \frac{t-1}{\lambda}\right) + 4\log\left(\frac{\pi t}{\sqrt{2\vartheta}}\right)}$ and $\alpha = \sqrt{\frac{8\vartheta}{KD(D+3)\pi^2}}$, let $M = \lfloor \min\left\{t' \mid (t-1)\sigma_{r\downarrow}^2 + \lambda \geq 2D\sqrt{K(t-1)\log\frac{t}{\alpha}}, \forall t \geq t'\right\} \rfloor$, with probability greater than $1 - \vartheta$, PRUCB-HPM has,*

$$R(T) \leq \underbrace{\sqrt{\beta_T}\sqrt{\frac{2D}{\log(\frac{5}{4})}\log\left(1 + \frac{(1+\sigma_{r\uparrow}^2)(T-M)}{\lambda}\right)(T-M)}}_{R^c(T): \text{Regret by \textbf{preference estimation} error}} + \underbrace{4\sqrt{K\log\left(\frac{T}{\alpha}\right)(T-M)}}_{R^r(T): \text{Regret by \textbf{reward estimation} error}} + M$$

$$= \mathcal{O}\left(D\log(T)\sqrt{T} + \sqrt{D\log(T/\vartheta)T} + \sqrt{K\log(T/\vartheta)T}\right) = \tilde{O}(D\sqrt{T}).$$

Theorem 6 shows that, even without direct preference feedback, PRUCB-HPM achieves sub-linear regret through carefully designed mechanisms for preference adaptation. In particular, for $t \geq M$, where $M$ [3] is a constant independent of $T$, the regret asymptotically scales as $\tilde{O}(D\sqrt{T})$. Interestingly, the regret due to preference estimation error exceeds that due to reward estimation error, becoming the dominant regret term. This is expected, given the increased difficulty of estimating latent preferences through regression. The proof of Theorem 6 is provided in Appendix E.2.

## 8 NUMERICAL ANALYSIS

In this section, we report the performance of PRUCB and PRUCB-SPM in a stationary preference environment. The PAMO-MAB instance is set with $K$ arms and $D$ objectives. The preference means are random defined, and the regret is defined by Eq 2. Detailed experimental settings and more experimental results can be found in Appendix A.1.

Fig. 5 shows that our algorithms significantly outperform other competitors. Moreover, from the zoom-in window, we observe that PRUCB-SPM exhibits only a very slight performance degradation compared to PRUCB (under known preferences), indicating that the proposed PRUCB-SPM can effectively model user preference in stationary preference environments.

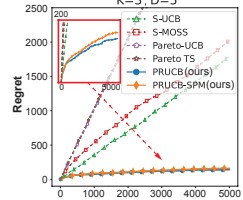

Figure 5: Regrets under stationary preference environment.

It is worth noting that other competitors are preference-free algorithms, all of which exhibit linear regret, aligning with our lower bound (Proposition 1). In other words, this demonstrates that approaches agnostic to user preferences cannot align their outputs with user preferences, even if they achieve Pareto optimality. For more experimental results under stationary, non-stationary and hidden preference environments, please refer to Appendix A.1, A.2 and A.3.

## 9 CONCLUSION

In this paper, we make the first effort to theoretically explore the explicit user preferences-aware MO-MAB, where the overall-reward is determined by both arm reward and user preference. Motivated by real-world applications, we provide a comprehensive analysis of this problem under three preference structures, with corresponding algorithms that achieve provably efficient with sub-linear regrets. The main analytical results in this paper are summarized in Table 1.

---

[3]Since $\sigma_{r\downarrow}^2 \in \mathbb{R}^+$, we have $\lim_{t \to \infty} 2D\sqrt{K(t-1)\log\frac{t}{\alpha}}/(\sigma_{r\downarrow}^2(t-1)) = \lim_{t \to \infty} C_1\sqrt{\frac{\log(t)-C_2}{t-1}} = 0$, because $\sqrt{\log(t)}$ grows very slowly compared to $\sqrt{t-1}$ as $t$ increases. Hence $M$ exists for sufficiently large $t'$.

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

# A EXPERIMENTS

In this section, we conduct numerical experiments to evaluate the effectiveness of our proposed algorithms under different user preference environments.

## A.1 EXPERIMENTS IN STATIONARY PREFERENCE ENVIRONMENT

### A.1.1 COMPARISON WITH BASELINES

In this section, we verify the capability of PRUCB and PRUCB-SPM to model user preference $c_t$ and optimize the overall reward in a stationary preference environment. We compare these two algorithms in terms of regret defined in Eq 2 with the following multi-objective bandits algorithms.

- S-UCB (Drugan & Nowe, 2013): the scalarized UCB algorithm, which scalarizes the multi-dimensional reward by assigning weights to each objective and then employs the single objective UCB algorithm Auer et al. (2002). Throughout the experiments, we assign each objective with equal weight.

- S-MOSS: the scalarized UCB algorithm, which follows the similar way with S-UCB by scalarizing the multi-dimensional reward into a single one, but uses MOSS (Audibert & Bubeck, 2009) policy for arm selection.

- Pareto-UCB (Drugan & Nowe, 2013): the Pareto-based algorithm, which compares different arms by the upper confidence bounds of their expected multi-dimensional reward by Pareto order and pulls an arm uniformly from the approximate Pareto front.

- Pareto-TS (Yahyaa & Manderick, 2015): the Pareto-based algorithm, which makes use of the Thompson sampling technique to estimate the expected reward for every arm and selects an arm uniformly at random from the estimated Pareto front.

**Experimental settings.** For evaluation, we use a synthetic dataset. Specifically, we consider the MO-MAB with $K$ arms, each arm $i \in [K]$ associated with a $D$-dimensional reward, where the reward of each objective $d$ follows a Bernoulli distribution with a randomized mean $\boldsymbol{\mu}_i(d) \in [0, 1]$. For user preference, we consider two settings including predefined preference and randomized preference. For predefined preference-aware structure, we define the mean preference $\overline{c}$ as $\overline{c}(d) = 2.0$ if $d = j; 0.5$ otherwise, where $j \in [D]$ is randomly selected. The practical implication of this structure is that it represents a common scenario in which the user exhibits a markedly higher preference for one particular objective while showing little interest in others. For randomized preference, the values of mean preference $\overline{c}$ are randomly defined within $[0, 5]$. For both setups, the instantaneous preference is generated under Gaussian distributions with corresponding means and variance of 0.5. To guarantee the non-negative preference, we clip the generated instantaneous preference within $[0, 2\overline{c}]$.

**Implementations.** For the implementations of the algorithms, we reveal the true expected preference for PRUCB before arm pulling in each episode, while for PRUCB-SPM, we use the estimated preference instead. Following the previous studies (Auer et al., 2002; Audibert et al., 2007), we set $\alpha = 1$. The time horizon is set to $T = 5000$ rounds, and we repeat 10 trials for each set of evaluation due to the randomness from both environment and algorithms.

**Results.** We report the averaged regret performance of the algorithms under stationary preference distributions in Fig. 6. It is evident that our algorithms significantly outperform other competitors in all experiments. This is expected since the competing algorithms are designed for Pareto-optimality identification and do not utilize the preference structure of users considered in this paper, which our algorithm explicitly exploits. Additionally, from the zoom-in window, we observe that PRUCB-SPM exhibits only a very slight performance degradation compared to PRUCB, which knows the preference expectation in advance. This indicates that the proposed PRUCB-SPM can effectively model user preference via empirical estimation in stationary preference environments.

### A.1.2 ROBUSTNESS TO STOCHASTIC ATTACKS

In this section, we explore the robustness of our proposed RUCB-SPM against stochastic corruptions on preference feedback.

**Experimental settings and implementations.** We consider the same preference-aware MO-MAB environment as Appendix A.1.1. Specifically, the stochastic reward and preference is generated in the same manner as in Appendix A.1.1. Additionally, we define a stochastic attacker which

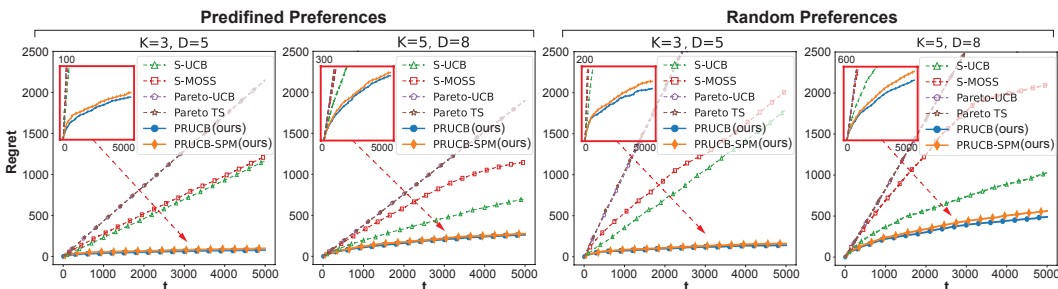

Figure 6: Regrets of different algorithms under stationary preference environment.

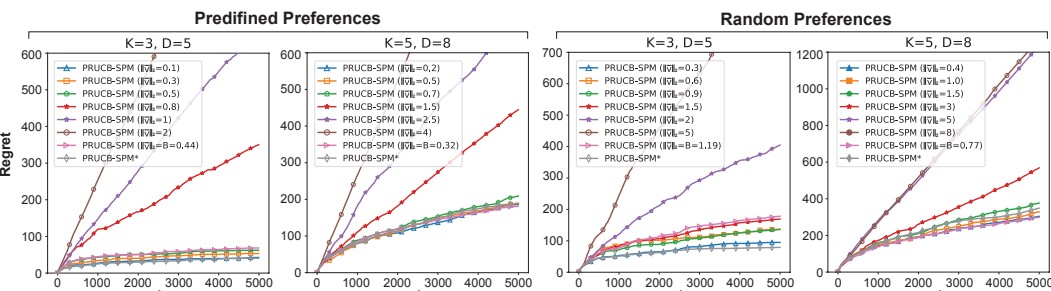

Figure 7: Regrets of RUCB-SPM against different level of stochastic preference corruptions.

manipulates the observed preference feedback with a corruption component $z_t$ at each episode $t$, i.e., $\tilde{c}_t = c_t + z_t$, where $c_t$ is the ground-truth while $\tilde{c}_t$ is the corrupted preference observed by learner, and the corruption component $z_t$ has the mean vector of $\overline{z}$. The mean corruption vector $\overline{z}$ is generated by uniformly selecting a value within $[-1, 1]$ for each objective $d \in [D]$, and then rescaling the vector to a fixed $L_2$-norm $\|\overline{z}\|_2$ to represent the level of corruption. In our experiment, we vary the level of $\|\overline{z}\|_2$ to investigate the robustness of our proposed RUCB-SPM against stochastic preference attacks. The parameter settings of RUCB-SPM follows the implementation in Appendix A.1.1. Similarly, we set time horizon $T = 5000$ rounds, and repeat 10 trials for each set of evaluation.

**Results.** We report the averaged regret of RUCB-SPM under different level of preference corruptions ($\|\overline{v}\|_2$) in Fig. 7. Specifically, $B = \min_{i \in [K] \setminus a^*} \left\{ \frac{\overline{c}^T \Delta_i}{(1+\frac{1}{D})\|\Delta_i\|_2} \right\}$ denotes the robustness threshold of RUCB-SPM derived in our theoretical analysis in Remark D.1. RUCB-SPM* denotes the algorithm under no attacks.

From the results, we can see that for the attack level under or even slightly higher than $B$, RUCB-SPM can achieve very close sub-linear regret with the original RUCB-SPM* without attacks, indicating the robustness of RUCB-SPM against stochastic preference corruptions. One interesting discovery is that higher objective dimensions present greater tolerance to corruption. Specifically, in the case with with $D = 5$, RRUCB-SPM is robust to a corruption level of approximately $1.2B$ (see the curve of $\|\overline{v}\|_2 = 0.5$ in the first column subplot, and the curve of $\|\overline{v}\|_2 = 1.5$ in the third column subplot). In contrast, for the case with $D = 8$, RUCB-SPM remains robust up to a corruption level of $2B$ (see the curve of $\|\overline{v}\|_2 = 0.7$ in the second column subplot, and the curve of $\|\overline{v}\|_2 = 1.5$ in the fourth column subplot). This might be due to the fact that, as the dimension of the preference space increases, it becomes more challenging to find an efficient attack combination across $D$ dimensions under the constraint $\|\overline{v}\|_2$ to achieve successful attack.

## A.2 EXPERIMENTS IN ABRUPTLY PREFERENCES CHANGING ENVIRONMENT

In this section, we verify the capability of PRUCB-APM to model user preference $c_t$ and optimize the overall reward in a preference abruptly changing environment.

### A.2.1 COMPARISON WITH BASELINES

**Experimental settings.** We consider the same MO-MAB baseline algorithms as in the stationary preference setting for comparison. The reward is generated in the same manner as in the stationary preference setting. Similarly, two preference settings are evaluated: predefined preference and

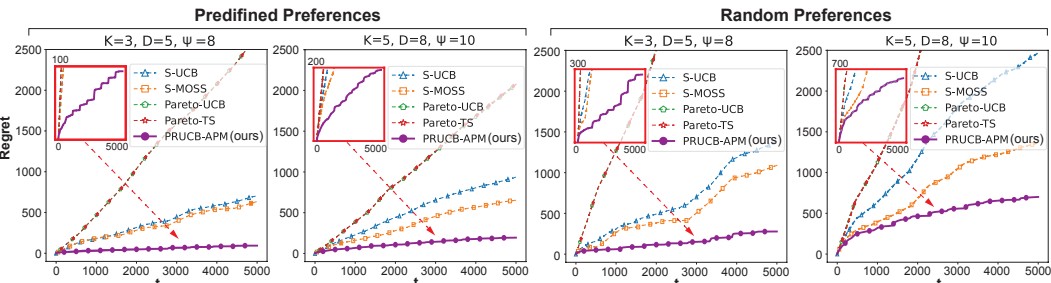

Figure 8: Regrets of different algorithms under abruptly changing preference distribution.

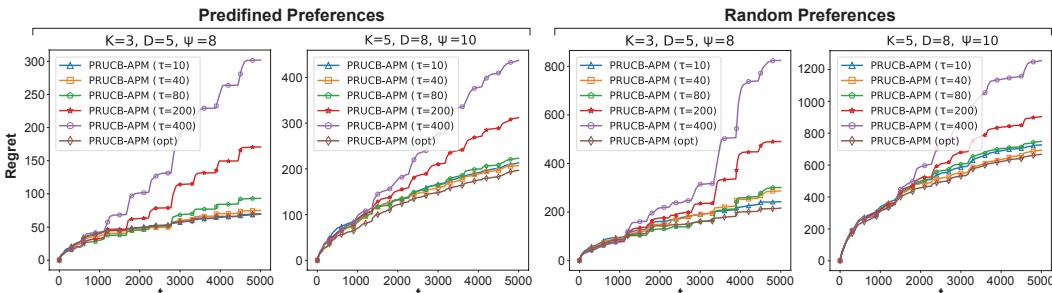

Figure 9: Regrets of RUCB-APM with different choices of sliding-window lengths $\tau$.

randomized preference. To simulate the abruptly changing preference environment, we define the number of breakpoints as $\psi$, and the changing episodes are isometrically sampled within $T$. At each changing episode $t_l$, we re-define the mean value of preference $\bar{c}_t$ for instantaneous preference generation in the following episodes until the next changing episode $t_{l+1}$. For predefined preference, we set the mean preference $\bar{c}_t$ as $\bar{c}(d) = 2.0$ if $d = j_{t_l}; 0.5$ otherwise, where $j_{t_l} \in [D]$ is randomly chosen at each changing episode $t_l$. For randomized preference, the mean vector of preference $\bar{c}$ is randomly re-defined within $[0, 5]$ at each changing episode.

**Implementation.** For the proposed PRUCB-APM, we set $\alpha = 1$ following previous studies (Auer et al., 2002; Audibert et al., 2007), and set the sliding-window length $\tau = 80$ while not the value as Remark 6.2 suggests since we assume $T$ and $\psi$ are not known to the learner. We perform 10 trials up to round $T = 5000$ for evaluation.

**Results.** The average regrets of the algorithms under abrupt environment with different settings of $K, D$ and $\psi$ are reported in Fig. 8. It is evident that our algorithm PRUCB-APM significantly outperform other competitors in all experiments. By the zoom-in window, we observe that PRUCB-APM can well estimate user preference $c_t$ with a fast convergence rate and utilize the preference information for optimizing the overall reward in a preference abruptly changing environment.

**Parameter analysis of PRUCB-APM on sliding-window lengths $\tau$.** We investigate the impact of sliding-window lengths $\tau$ in PRUCB-APM on the overall performance by varying $\tau$ from 10 to 400. The results are depicted in Fig. 9. PRUCB-APM (opt) refer to the choice of $\tau = \left(\frac{4DT}{\psi}\right)^{\frac{1}{3}}$ as suggested in Remark 6.2. It shows that for the choice of small $\tau$ (under 80), it present a close regret performance, indicating PRUCB-APM is not that sensitive to the choice of small sliding-window length. Specifically, for very small sliding-window length (i.e., $\tau = 10$), it presents slightly worse performance than that of the optimal $\tau$. However, for the large sliding-window length (above 200), it adapts to changes slowly.

## A.3 EXPERIMENTS IN HIDDEN PREFERENCES ENVIRONMENT

In this section, we evaluate the performance of PRUCB-HPM in modeling user preference $c_t$ and optimizing the overall reward when explicit user preference is not visible, but overall reward $g_{a_t,t}$ and reward $r_{a_t,t}$ are revealed after each episode.

**Experimental protocol.** Given that PRUCB-HPM models both the expected arms reward and user preference, we designed a new *user-switching protocol* for evaluation. Figure 10 illustrates this

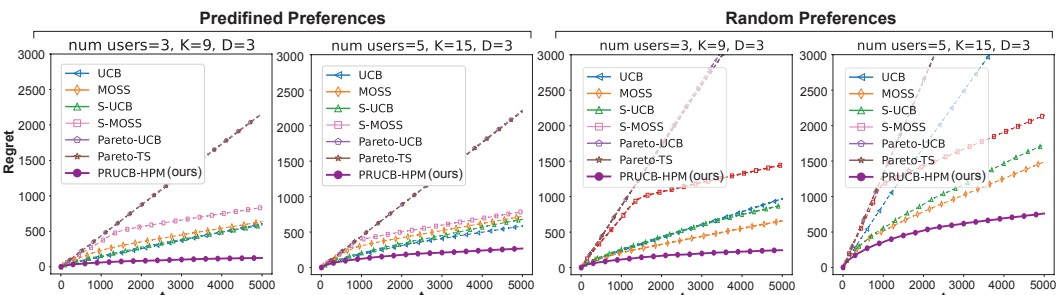

Figure 10: (a) Users switching protocol for experimental evaluation of hidden preference and multi-objective reward modelings. (b) One real-world example of the experimental protocol.

Figure 11: Regrets of different algorithms under hidden preference environment.

protocol with 3 users and 9 arms. Specifically, at each episode, one user is exposed to a block of arms (3 in our illustration). Only the arms within this block can be selected for this user. After one arm has been pulled, the system observes the reward $r_{a_t,t}$ and user's overall ratings $g_{a_t,t}$ corresponding to the pulled arm $a_t$. In the next episode, the arm block rotates to another user. The goal is to maximize the cumulative overall ratings from all users.

This protocol simulates real-world applications, such as recommender systems, where empirical multi-objective rewards (ratings) of arms (recommendation candidates) are obtained from a diverse set of users rather than a single fixed user. Additionally, users are not always exposed to a fixed set of arms (recommendation candidates). This user-switching protocol allows us to evaluate the algorithm's ability to model arm reward and user preference, thus enabling the customized optimization of users' overall ratings. In Figure 10(b), we present an intuitive example of the protocol in the context of real-world hotel recommendations. Specifically, the blocks represent different cities (e.g., NYC, LA, CHI), and the hotel candidates within these cities correspond to the arms within the blocks. At each time step, a customer travels to a city, stays in a hotel recommended by the system, and leaves feedback (both objective and overall ratings) after her or his stay. In the next episode, the customer travels to a different city and encounters a new set of hotel options. The hotel recommender system needs to learn the multi-objective rewards of all hotel candidates from various customers and model each customer's preference based on their multi-objective and overall feedback. This enables the system to customize optimal hotel recommendations tailored to individual user preference.

**Baselines.** For performance comparison, we choose the MO-MAB baselines used in stationary environment (Appendix A.1.1, including S-UCB (Drugan & Nowe, 2013), S-MOSS, Pareto-UCB (Drugan & Nowe, 2013) and Pareto-TS (Yahyaa & Manderick, 2015)). Additionally, note that the scale overall score is also provided, it is feasible to use standard MAB methods by leveraging historical overall rewards for optimization. Hence we also choose classic MAB algorithms including UCB (Auer et al., 2002) and MOSS (Audibert & Bubeck, 2009) for comparison.

**Experimental settings.** In our experiment, we set $N$ users and $3N$ arms in total, and each arm associates with $D$-dimensional reward. The generations of instantaneous reward $r_{i,t}$ of arms and user preference $c_t$ follow the same settings as stationary environment.

For user-switching protocol, we set $N$ blocks in total, with each block containing 3 fixed arms. At each episode, each user will be randomly assigned one block without replacement. The learner can only select the arm within assigned block for each user.

**Implementation.** Similarly, we set $\alpha = 1$ in PRUCB-HPM. For regularization coefficient, we set $\lambda = 1$. For confidence radius, we set $\sqrt{\beta_t} = 0.1\sqrt{D\log(t)}$. We perform 10 trials up to round T = 5000 for each set of evaluation.

**Results.** We report average performance of the algorithms in Fig. 11. As shown, our proposed PRUCB-HPM achieves superior results in terms of regret under all experimental settings compared to other competitors. This empirical evidence suggests that modeling user preference and leveraging this information for arm selection significantly enhances the performance of customized bandits optimization.

## B PROOF OF PROPOSITION 1

**Lemma 7** (Variant of Lemma 7 in Jun et al. (2018)). *Assume that a bandit algorithm enjoys a sub-linear regret bound, then $\mathbb{E}[N_{i,T}] = o(T), \forall i \neq a^*$.*

*Proof.* The sub-linear regret bound implies that for a sufficiently large $T$ there exists a constant $C > 0$ such that $\sum_{i=1}^{K} \mathbb{E}[N_{i,T}]\overline{c}_t^T(\mu_{a^*} - \mu_i) < CT$. Hence we have $\mathbb{E}[N_{i,T}]\overline{c}_t^T(\mu_{a^*} - \mu_i) \leq CT, \forall i \neq a^*$, implying $\mathbb{E}[N_{i,T}] < \frac{CT}{\overline{c}_t^T(\mu_{a^*} - \mu_i)}$. $\qquad\square$

**Definition 2** (Pareto order, Lu et al. (2019)). *Let $u, v \in \mathbb{R}^D$ be two vectors.*

- *$u$ dominates $v$, denoted as $u \succ v$, if and only if $\forall d \in [D], u(d) > v(d)$.*

- *$v$ is not dominated by $u$, denoted as by $u \not\succ v$, if and only if $u = v$ or $\exists d \in [D], v(d) > u(d)$.*

- *$u$ and $v$ are incomparable, denoted as $u \| v$, if and only if either vector is not dominated by the other, i.e., $u \not\succ v$ and $v \not\succ u$.*

*Proof of Proposition 1.* We first construct an arbitrary $K$-armed $D$-objective MO-MAB environment with conflicting reward objectives. Let each objective reward of each arm follow a distribution, i.e., $r_{i,t}(d) \sim \text{Dist}_{i,d}, \forall i \in [K], \forall d \in [D]$, with mean of $\mu_i(d)$. Define $\mathcal{P} := \{[\text{Dist}_{1,d}]^D, [\text{Dist}_{2,d}]^D, ..., [\text{Dist}_{K,d}]^D\}$ be the set of $K$-armed $D$-dimensional reward distributions.

We start with a simple case where the MO-MAB environment has two conflicting objective arms. Specifically, assume that $\exists u, v \in [K]$, s.t.,

$$\mu_u \neq \mu_v; \quad \mu_u \| \mu_v$$

and

$$\mu_u \succ \mu_i, \mu_v \succ \mu_i, \forall i \in [k] \setminus \{u, v\}.$$

Due to $\mu_u \neq \mu_v$, by taking the orthogonal complement of $\mu_u - \mu_v$, we can construct a subset $\mathcal{C}_{\varsigma^+} := \{c \in \mathbb{R}^D | c^T(\mu_u - \mu_v) = 0\}$. Next we consider two different constant preferences vector sets as the user's preferences, to construct two sets of preferences-aware MO-MAB scenarios.

**Scenarios $\mathcal{S}_{\varsigma^+}$.** For any $\varsigma^+ > 0$, we can construct a subset $\mathcal{C}_{\varsigma^+} := \{c \in \mathbb{R}^D | c^T(\mu_u - \mu_v) = \varsigma^+\}$. Specifically, the general form of $c_{\varsigma^+} \in \mathcal{C}_{\varsigma^+}$ can be written as $c_{\varsigma^+} = \frac{\varsigma^+}{\|\mu_u - \mu_v\|_2^2}(\mu_u - \mu_v) + c_0$, where $c_0$ is any vector such that $c_0 \in \mathcal{C}_0$. Then for the preferences-aware MO-MAB scenarios $\mathcal{S}_{\varsigma^+} := \{\mathcal{P} \times \mathcal{C}_{\varsigma^+}\}$ under the sets of arm reward distributions $\mathcal{P}$ and user preferences $\mathcal{C}_{\varsigma^+}$, it is obvious that arm $u$ is the optimal arm since $\mu_u \succ \mu_i, \forall i \in [K] \setminus \{u, v\}$ and $c_{\varsigma^+}^T \mu_u > c_{\varsigma^+}^T \mu_v, \forall c_{\varsigma^+} \in \mathcal{C}_{\varsigma^+}$.

**Scenarios $\mathcal{S}_{\varepsilon^-}$.** Similarly, for any $\varepsilon^- < 0$, we can construct a subset $\mathcal{C}_{\varepsilon^-} := \{c \in \mathbb{R}^D | c^T(\mu_u - \mu_v) = \varepsilon^-\}$, with the general form of $c_{\varepsilon^-} = \frac{\varepsilon^-}{\|\mu_u - \mu_v\|_2^2}(\mu_u - \mu_v) + c_0$, where $c_0$ is any vector such that $c_0 \in \mathcal{C}_0$. For scenarios $\mathcal{S}_{\varepsilon^-} := \{\mathcal{P} \times \mathcal{C}_{\varepsilon^-}\}$ with same arm rewards distributions $\mathcal{P}$ but modified user preferences $\mathcal{C}_{\varepsilon^-}$ sets, we have the arm $v$ to be the optimal.

We use $\mathbb{P}_{\varsigma^+}$ to denote the probability with respect to the scenarios $\mathcal{S}_{\varsigma^+}$, and use $\mathbb{P}_{\varepsilon^-}$ to denote the probability conditioned on $\mathcal{S}_{\varepsilon^-}$. Analogous expectations $\mathbb{E}_{\varsigma^+}[\cdot]$ and $\mathbb{E}_{\varepsilon^-}[\cdot]$ will also be used. Let $\mathbf{a}^{t-1} = \{A_1, ..., A_{t-1}\}$ and $\mathbf{r}^{t-1} = \{x_1, ..., x_{t-1}\}$ be the actual sequence of arms pulled and the sequence of received rewards up to episode $t-1$, and $\mathbf{H}^{t-1} = \{\langle A_1, x_1 \rangle, ..., \langle A_{t-1}, x_{t-1} \rangle\}$ be the

corresponding historical rewards sequence. For consistency, we define $\mathbf{a}^0$, $\mathbf{r}^0$ and $\mathbf{H}^0$ as the empty sets. Assume there exists a preferences-free algorithm $\mathcal{A}$ (i.e., Pareto-UCB (Drugan & Nowe, 2013)) that is possibly dependent on historical rewards sequence $\mathbf{H}^{t-1}$ at episode $t$ (classical assumption in MAB), achieving sub-linear regret in scenarios $\mathcal{S}_{\varsigma+}$. Let $N_{i,T}$ be the number of pulls of arm $i$ by $\mathcal{A}$ up to $T$ episode. By Lemma 7, we have

$$\mathbb{E}_{\varsigma+}[N_{*,T}] = \mathbb{E}_{\varsigma+}[N_{u,T}] = T - o(T). \tag{12}$$

Since the policy $\pi_t^{\mathcal{A}}$ of $\mathcal{A}$ is possibly dependent on $\mathbf{H}^{t-1}$ but independent on the sequences of instantaneous preferences $\mathbf{c}^t$ and preferences means $\bar{\mathbf{c}}^t$, for $t \in (0, T\}$, $i \in [K]$ we have

$$\mathbb{E}_{\varsigma+}[\mathbb{1}_{a_t=i}] - \mathbb{E}_{\varepsilon-}[\mathbb{1}_{a_t=i}]$$

$$= \sum_{\mathbf{a}^{t-1} \in [K]^{t-1}} \int_{\mathbf{r}^{t-1} \in [0,1]^{D \times (t-1)}} \mathbb{P}_{\pi_t^{\mathcal{A}}}(a_t = i | \mathbf{H}^{t-1}, [\mathbf{c}_0]^t, [\mathbf{c}_0]^t) \cdot \mathbb{P}_{\varsigma+}(\mathbf{H}^{t-1}) d\mathbf{r}^{t-1}$$

$$- \sum_{\mathbf{a}^{t-1} \in [K]^{t-1}} \int_{\mathbf{r}^{t-1} \in [0,1]^{D \times (t-1)}} \mathbb{P}_{\pi_t^{\mathcal{A}}}(a_t = i | \mathbf{H}^{t-1}, [\mathbf{c}_{\varepsilon-}]^t, [\mathbf{c}_{\varepsilon-}]^t) \cdot \mathbb{P}_{\varepsilon-}(\mathbf{H}^{t-1}) d\mathbf{r}^{t-1}$$

$$\underset{(a)}{=} \sum_{\mathbf{a}^{t-1} \in [K]^{t-1}} \int_{\mathbf{r}^{t-1} \in [0,1]^{D \times (t-1)}} \mathbb{P}_{\pi_t^{\mathcal{A}}}(a_t = i | \mathbf{H}^{t-1}) \cdot \left( \mathbb{P}_{\varsigma+}(\mathbf{H}^{t-1}) - \mathbb{P}_{\varepsilon-}(\mathbf{H}^{t-1}) \right) d\mathbf{r}^{t-1}, \tag{13}$$

with

$$\mathbb{P}_{\varsigma+}(\mathbf{H}^{t-1}) = \prod_{\tau=1}^{t-1} \left( \mathbb{P}_{\varsigma+}(\mathbf{H}^{\tau-1}) \cdot \mathbb{P}_{\pi_\tau^{\mathcal{A}}}(a_\tau = A_\tau | \mathbf{H}^{\tau-1}) \cdot \mathbb{P}_{\varsigma+}(r_{a_\tau} = \boldsymbol{x}_\tau | a_\tau = A_\tau) \right),$$

$$\mathbb{P}_{\varepsilon-}(\mathbf{H}^{t-1}) = \prod_{\tau=1}^{t-1} \left( \mathbb{P}_{\varepsilon-}(\mathbf{H}^{\tau-1}) \cdot \mathbb{P}_{\pi_\tau^{\mathcal{A}}}(a_\tau = A_\tau | \mathbf{H}^{\tau-1}) \cdot \mathbb{P}_{\varepsilon-}(r_{a_\tau} = \boldsymbol{x}_\tau | a_\tau = A_\tau) \right). \tag{14}$$

where $\boldsymbol{c}_0, \boldsymbol{c}_{\varepsilon-}$ can be any constant vectors such that $\boldsymbol{c}_0 \in \mathcal{C}_0$ and $\boldsymbol{c}_0 \in \mathcal{C}_{\varepsilon-}$. (a) holds since the policy $\pi_t^{\mathcal{A}}$ is independent of $\mathbf{c}^t$, $\bar{\mathbf{c}}^t$ and hence $\mathbb{P}_{\pi_t^{\mathcal{A}}}(a_t = i | \mathbf{H}^{t-1}) = \mathbb{P}_{\pi_t^{\mathcal{A}}}(a_t = i | \mathbf{H}^{t-1}, [\boldsymbol{c}_0]^t, [\boldsymbol{c}_0]^t) = \mathbb{P}_{\pi_t^{\mathcal{A}}}(a_t = i | \mathbf{H}^{t-1}, [\boldsymbol{c}_{\varepsilon-}]^t, [\boldsymbol{c}_{\varepsilon-}]^t)$ (recall the definition of preferences-free algorithm in Definition 1).

Additionally, please note that both scenarios $\mathcal{S}_{\varsigma+}$ and $\mathcal{S}_{\varepsilon-}$ share the same arm reward distributions $\mathcal{P}$, which implies that for any $t \in (0, T]$ and $A \in [K]$, we have

$$\mathbb{P}_{\varsigma+}(r_{a_t} = \boldsymbol{x}_t | a_t = A) = \mathbb{P}_{\varepsilon-}(r_{a_t} = \boldsymbol{x}_t | a_t = A).$$

Combining result above with Eq. 14 and using the fact that $\mathbf{H}^0 := \emptyset$ for both $\mathcal{S}_{\varsigma+}$ and $\mathcal{S}_{\varepsilon-}$, it can be easily verified by induction that $\mathbb{P}_{\varsigma+}(\mathbf{H}^{t-1}) = \mathbb{P}_{\varepsilon-}(\mathbf{H}^{t-1})$. Plugging this back to Eq 13 yields

$$\mathbb{E}_{\varsigma+}[\mathbb{1}_{a_t=i}] - \mathbb{E}_{\varepsilon-}[\mathbb{1}_{a_t=i}]$$

$$= \sum_{\mathbf{a}^{t-1} \in [K]^{t-1}} \int_{\mathbf{r}^{t-1} \in [0,1]^{D \times (t-1)}} \mathbb{P}_{\pi_t^{\mathcal{A}}}(a_t = i | \mathbf{H}^{t-1}) \cdot \left( \mathbb{P}_{\varsigma+}(\mathbf{H}^{t-1}) - \mathbb{P}_{\varepsilon-}(\mathbf{H}^{t-1}) \right)^{0} d\mathbf{r}^{t-1} = 0. \tag{15}$$

By summing over $T$ we can derive that

$$\mathbb{E}_{\varsigma+}[N_{i,T}] = \sum_{t=1}^{T} \mathbb{E}_{\varsigma+}[\mathbb{1}_{a_t=i}] = \sum_{t=1}^{T} \mathbb{E}_{\varepsilon-}[\mathbb{1}_{a_t=i}] = \mathbb{E}_{\varepsilon-}[N_{i,T}].$$

Combining above result with Eq. 12 gives that=
$$\mathbb{E}_{\varsigma+}[N_{u,T}] = \mathbb{E}_{\varepsilon-}[N_{u,T}] = T - o(T) = \Omega(T).$$

However, recall that in scenarios $\mathcal{S}_{\varepsilon^-}$, $u$ is a suboptimal arm, which implies that the regret of $\mathcal{A}$ in $\mathcal{S}_{\varepsilon^-}$ would be at least $\Omega(T)$, i.e.,

$$R(T) = \sum_{i \neq v} \boldsymbol{c}_{\varepsilon^-}^T (\mu_v - \mu_i) \mathbb{E}_{\varepsilon^-}[N_{i,T}]$$

$$> |\varepsilon^-| \mathbb{E}_{\varepsilon^-}[N_{u,T}] = \Omega(T).$$

The analysis above indicates that for the case with two objective-conflicting arms $u, v$, for any preferences-free algorithm $\mathcal{A}$, if there exists a $\varsigma^+ > 0$ such that $\mathcal{A}$ can achieve sub-linear regret in scenarios $\mathcal{S}_{\varsigma^+}$, then it will suffer the regret of the order $\Omega(T)$ in scenarios $\mathcal{S}_{\varepsilon^-}$ for all $\varepsilon^- < 0$, and vice verse (i.e., sub-linear regret in $\varepsilon^- > 0$ while $\Omega(T)$ regret in $\mathcal{S}_{\varsigma^+}$).

Next we extend the solution to the MO-MAB environment containing more than two objective-conflicting arms. Specifically, for each conflicting arm $i$, we can simply select another conflicting arm $j$ to construct a pair, and apply the solution we derived in two-conflicting arms case. By traversing all conflicting arms, we have that for any preferences-free algorithm $\mathcal{A}$ achieving sub-linear regret in a scenarios set $\mathcal{S}_0$ with a subset of conflicting arms $\{a^*\}$ as the optimal, there must exists another scenarios set $\mathcal{S}_0'$ for each arm $i \in \{a^*\}$ such that the arm $i$ is considered as suboptimal and lead to the regret of order $\Omega(T)$. This concludes the proof of Proposition 1.

**Remark B.1.** *As a side-product of the analysis above, we have that:*

*If one MO-MAB environment contains multiple objective-conflicting arms, i.e., $|\mathcal{O}^*| \geq 2$, where $\mathcal{O}^*$ is the Pareto Optimal front. Then for any Pareto-Optimal arm $i \in \mathcal{O}^*$, there exists preferences subsets such that the arm $i$ is suboptimal.*

$\square$

# C   ANALYSES FOR SECTION 5 (KNOWN PREFERENCE)

## C.1   PROOF OF THEOREM 2

For analyzing PRUCB's behaviours in the environment where the preference distribution is possibly dynamic, the main difficulty lies in tracking the potentially changes of the best arm. Specifically, in preference changing environments, the optimal arm is not fixed any more and would change with the changing preference distributions.

We begin with a more general upper bound (Proposition 8) for the learner's behavior using a policy that optimizes the inner product between the reward upper confidence bound (UCB) of arms and an arbitrary dynamic vector $\boldsymbol{b}_t$. It demonstrates that after a sufficiently large number of samples (on the order of $\mathcal{O}(\log T)$) for each arm $i$, for the episodes where the inner product of its rewards expectations with $\boldsymbol{b}_t$ is not highest, the expected number of times arm $i$ is pulled can be well controlled by a constant. The proof of Proposition 8 is provided in Appendix C.1.1.

**Proposition 8.** *Let $\boldsymbol{b}_t \in \mathbb{R}^D$ be an arbitrary bounded vector at time step $t$ with $\|\boldsymbol{b}_t\|_1 \leq M$, define $\mathcal{M}_i := \{t \in [T] \mid i \neq \arg\max_{j \in [K]} \boldsymbol{b}_t^T \boldsymbol{\mu}_j\}, \forall i \in [K]$. For the policy of $a_t = \arg\max \Phi(\boldsymbol{b}_t, \hat{\boldsymbol{r}}_{i,t} + \sqrt{\frac{\log(t/\alpha)}{\max\{1, N_{i,t}\}}} \boldsymbol{e})$, for any arm $i \in [K]$, any subset $\mathcal{M}_i^o \subset \mathcal{M}_i$, we have*

$$\mathbb{E}\left[\sum_{t \in \mathcal{M}_i^o} \mathbb{1}_{\{a_t=i\}}\right] \leq \frac{4M^2 \log\left(\frac{T}{\alpha}\right)}{L_i^2} + \frac{|\mathcal{B}_T^+|\pi^2\alpha^2}{3},$$

*where $L_i = \min_{t \in \mathcal{M}_i^o}\{\max_{j \in [K]\backslash i}\{\boldsymbol{b}_t^T(\boldsymbol{\mu_j} - \boldsymbol{\mu_i})\}\}$, $\mathcal{B}_T^+ := \{[\boldsymbol{b}_1(d), \boldsymbol{b}_2(d), ..., \boldsymbol{b}_T(d)] \neq \boldsymbol{0}, \forall d \in [D]\}$ is the collection set of non-zero $[\boldsymbol{b}(d)]^T$ sequence.*

*Proof of Theorem 2.* Define $\mathcal{T}_i = \{t \in [T] | a_t^* \neq i\}$ be the set of episodes when $i$ serving as a suboptimal arm over $T$. Let $\Delta_{i,t} = \mu_{a_t^*} - \mu_i \in \mathbb{R}^D, \forall t \in [1, T]$ be the gap of expected rewards between suboptimal arm $i$ and best arm $a_t^*$ at time step $t$, $\eta_i^\downarrow = \min_{t \in \mathcal{T}_i}\{\overline{\boldsymbol{c}}_t^T \Delta_{i,t}\}$ and $\eta_i^\uparrow = \max_{t \in \mathcal{T}_i}\{\overline{\boldsymbol{c}}_t^T \Delta_{i,t}\}$ refer to the lower and upper bounds of the expected overall-reward gap between $i$

and $a_t^*$ over $T$ when $i$ serving as a suboptimal arm. Let $\tilde{N}_{i,T}$ denotes the number of times that arm $i$ is played as a suboptimal arm, i.e.,

$$\tilde{N}_{i,T} = \sum_{t=1}^{T} \mathbb{1}_{\{a_t=i\neq a_t^*\}}.$$

Then we can apply Proposition 8 on $\tilde{N}_{i,T}$ for analysis. Specifically, by directly substituting $\boldsymbol{b}_t$ with $\overline{\boldsymbol{c}}_t$, the policy of $a_t$ aligns with that of PRUCB, and it is easy to verify that $\mathcal{M}_i = \mathcal{T}_i$, $L_i = \eta_i^{\downarrow}$. And thus by Proposition 8, we have

$$\mathbb{E}[\tilde{N}_{i,T}] = \mathbb{E}\left[\sum_{t\in\mathcal{T}_i}\mathbb{1}_{\{a_t=i\}}\right] \leq \frac{4\delta^2\log\left(\frac{T}{\alpha}\right)}{\eta_i^{\downarrow 2}} + \frac{|\mathcal{C}_T^+|\pi^2\alpha^2}{3},$$

where $\overline{\mathcal{C}}_T^+ := \{d \in [D] \mid [\overline{\boldsymbol{c}}_1(d), \overline{\boldsymbol{c}}_2(d), ..., \overline{\boldsymbol{c}}_T(d)] \neq [0]^T\}$ is the set of non-zero expected preference sequence on each dimension (objective). By multiplying above result with the corresponding upper-bound of expected gap $\eta_i^{\uparrow}$ and sum over $K$ arms concludes the proof of Theorem 2. $\qquad\square$

### C.1.1 Proof of Proposition 8

We begin with stating a useful central bound below.

**Lemma 9** (Hoeffding's inequality for general bounded random variables (Vershynin, 2018) (Theorem 2.2.6)). *Given independent random variables $\{X_1, ..., X_m\}$ where $a_i \leq X_i \leq b_i$ almost surely (with probability 1) we have:*

$$\mathbb{P}\left(\frac{1}{m}\sum_{i=1}^{m}X_i - \frac{1}{m}\sum_{i=1}^{m}\mathbb{E}[X_i] \geq \epsilon\right) \leq \exp\left(\frac{-2\epsilon^2 m^2}{\sum_{i=1}^{m}(b_i-a_i)^2}\right).$$

*Proof of Proposition 8.* Define $\tilde{a}_t^* = \arg\max_{j\in[K]} \boldsymbol{b}_t^T\boldsymbol{\mu}_j, \forall t \in (0,T]$, for any $\beta \in (0,T]$, we have

$$\sum_{t\in\mathcal{M}_i^o}\mathbb{1}_{\{a_t=i\}} \leq \sum_{t\in\mathcal{M}_i^o}\mathbb{1}_{\{a_t=i,N_{i,t}\leq\beta\}} + \sum_{t\in\mathcal{M}_i^o}\mathbb{1}_{\{a_t=i,N_{i,t}>\beta\}}$$
$$\leq \beta + \sum_{t\in[T]}\mathbb{1}_{\{a_t=i\neq\tilde{a}_t^*,N_{i,t}>\beta\}}. \tag{16}$$

where the first term refers to the event of insufficient sampling (quantified by $\beta$) of arm $i$. , then for the event of second term, we have

$$\{a_t = i \neq \tilde{a}_t^*, N_{i,t} > \beta\}$$

$$\subset \underbrace{\left\{\boldsymbol{b}_t^T\hat{\boldsymbol{r}}_{i,t} > \boldsymbol{b}_t^T\boldsymbol{\mu}_i + \boldsymbol{b}_t^T\boldsymbol{e}\sqrt{\frac{\log(t/\alpha)}{N_{i,t}}}, N_{i,t} > \beta\right\}}_{\tilde{A}_t}$$

$$\cup \underbrace{\left\{\boldsymbol{b}_t^T\hat{\boldsymbol{r}}_{\tilde{a}_t^*,t} < \boldsymbol{b}_t^T\boldsymbol{\mu}_{\tilde{a}_t^*} - \boldsymbol{b}_t^T\boldsymbol{e}\sqrt{\frac{\log(t/\alpha)}{N_{\tilde{a}_t^*,t}}}, N_{i,t} > \beta\right\}}_{\tilde{B}_t} \tag{17}$$

$$\cup \underbrace{\left\{\tilde{A}_t^c, \tilde{B}_t^c, \boldsymbol{b}_t^T\hat{\boldsymbol{r}}_{i,t} + \boldsymbol{b}_t^T\boldsymbol{e}\sqrt{\frac{\log(t/\alpha)}{N_{i,t}}} \geq \boldsymbol{b}_t^T\hat{\boldsymbol{r}}_{\tilde{a}_t^*,t} + \boldsymbol{b}_t^T\boldsymbol{e}\sqrt{\frac{\log(t/\alpha)}{N_{\tilde{a}_t^*,t}}}, N_{i,t} > \beta\right\}}_{\tilde{\Gamma}_t}.$$

Specifically, $\tilde{A}_t$ and $\tilde{B}_t$ denote the events where the constructed upper confidence bounds (UCBs) for arm $i$ or the optimal arm $a$ fail to accurately bound their true expected rewards, indicating imprecise

rewards estimation. Meanwhile, $\tilde{\Gamma}_t$ represents the event where the UCBs for both arms effectively bound their expected rewards, yet the UCB of arm $i$ still exceeds that of the arm $\tilde{a}_t^*$ though it yields the maximum value of $\boldsymbol{b}_t^T \boldsymbol{\mu}_{\tilde{a}_t^*}$, leading to pulling of arm $i$. According to (Auer et al., 2002), at least one of these events must occur for an pulling of arm $i$ to happen at time step $t$.

For event $\tilde{\Gamma}_t$, the $\tilde{A}_t^c$ and $\tilde{B}_t^c$ imply

$$\boldsymbol{b}_t^T \boldsymbol{\mu}_i + \boldsymbol{b}_t^T \boldsymbol{e} \sqrt{\frac{\log(t/\alpha)}{N_{i,t}}} \geq \boldsymbol{b}_t^T \hat{\boldsymbol{r}}_{i,t} \quad \text{and} \quad \boldsymbol{b}_t^T \hat{\boldsymbol{r}}_{\tilde{a}_t^*,t} \geq \boldsymbol{b}_t^T \boldsymbol{\mu}_{\tilde{a}_t^*} - \boldsymbol{b}_t^T \boldsymbol{e} \sqrt{\frac{\log(t/\alpha)}{N_{\tilde{a}_t^*,t}}},$$

indicating

$$\boldsymbol{b}_t^T \boldsymbol{\mu}_i + 2\boldsymbol{b}_t^T \boldsymbol{e} \sqrt{\frac{\log(t/\alpha)}{N_{i,t}}} \geq \boldsymbol{b}_t^T \hat{\boldsymbol{r}}_{i,t} + \boldsymbol{b}_t^T \boldsymbol{e} \sqrt{\frac{\log(t/\alpha)}{N_{i,t}}} \geq \boldsymbol{b}_t^T \hat{\boldsymbol{r}}_{\tilde{a}_t^*,t} + \boldsymbol{b}_t^T \boldsymbol{e} \sqrt{\frac{\log(t/\alpha)}{N_{\tilde{a}_t^*,t}}} \geq \boldsymbol{b}_t^T \boldsymbol{\mu}_{\tilde{a}_t^*}$$

$$\implies 2\boldsymbol{b}_t^T \boldsymbol{e} \sqrt{\frac{\log(t/\alpha)}{N_{i,t}}} \geq \boldsymbol{b}_t^T \boldsymbol{\mu}_{\tilde{a}_t^*} - \boldsymbol{b}_t^T \boldsymbol{\mu}_i.$$

Combining above result and relaxing the first and second union sets in Eq. 17 gives:

$$\{a_t = i \neq \tilde{a}_t^*, N_{i,t} > \beta\}$$

$$\subset \left\{ \boldsymbol{b}_t^T \hat{\boldsymbol{r}}_{i,t} > \boldsymbol{b}_t^T \boldsymbol{\mu}_i + \boldsymbol{b}_t^T \boldsymbol{e} \sqrt{\frac{\log(t/\alpha)}{N_{i,t}}} \right\} \cup \left\{ \boldsymbol{b}_t^T \hat{\boldsymbol{r}}_{\tilde{a}_t^*,t} < \boldsymbol{b}_t^T \boldsymbol{\mu}_{\tilde{a}_t^*} - \boldsymbol{b}_t^T \boldsymbol{e} \sqrt{\frac{\log(t/\alpha)}{N_{\tilde{a}_t^*,t}}} \right\}$$

$$\cup \left\{ \boldsymbol{b}_t^T (\boldsymbol{\mu}_{\tilde{a}_t^*} - \boldsymbol{\mu}_i) < 2\|\boldsymbol{b}_t\|_1 \sqrt{\frac{\log(t/\alpha)}{N_{i,t}}}, N_{i,t} > \beta \right\}$$

$$\subset \underbrace{\left\{ \bigcup_{d \in \mathcal{D}_T^+} \left\{ \boldsymbol{b}_t(d) \hat{\boldsymbol{r}}_{i,t}(d) > \boldsymbol{b}_t(d) \boldsymbol{\mu}_i(d) + \boldsymbol{b}_t(d) \sqrt{\frac{\log(t/\alpha)}{N_{i,t}}} \right\} \right\}}_{A_t}$$

$$\cup \underbrace{\left\{ \bigcup_{d \in \mathcal{D}_T^+} \left\{ \boldsymbol{b}_t(d) \hat{\boldsymbol{r}}_{\tilde{a}_t^*,t}(d) < \boldsymbol{b}_t(d) \boldsymbol{\mu}_{\tilde{a}_t^*}(d) - \boldsymbol{b}_t(d) \sqrt{\frac{\log(t/\alpha)}{N_{\tilde{a}_t^*,t}}} \right\} \right\}}_{B_t}$$

$$\cup \underbrace{\left\{ \boldsymbol{b}_t^T (\boldsymbol{\mu}_{\tilde{a}_t^*} - \boldsymbol{\mu}_i) < 2\|\hat{\boldsymbol{c}}_t\|_1 \sqrt{\frac{\log(t/\alpha)}{N_{i,t}}}, N_{i,t} > \beta, \boldsymbol{b}_t^T \Delta_i > \eta_i - \epsilon \right\}}_{\Gamma_t},$$

(18)

where $\mathcal{D}_T^+ := \{d | [\boldsymbol{b}_1, \boldsymbol{b}_2, ..., \boldsymbol{b}_T](d) \in \mathcal{B}_T^+\}$, and $\mathcal{B}_T^+ := \{[\boldsymbol{b}_1(d), \boldsymbol{b}_2(d), ..., \boldsymbol{b}_T(d)] \neq \boldsymbol{0}, \forall d \in [D]\}$ is the collection set of non-zero $[\boldsymbol{b}(d)]^T$ sequence.

Then on event $A_t$, by applying Hoeffding's Inequality (Lemma 9), for any $d \in [D]$, we have

$$\mathbb{P}\left( \boldsymbol{b}_t(d) \hat{\boldsymbol{r}}_{i,t}(d) > \boldsymbol{b}_t(d) \boldsymbol{\mu}_i(d) + \boldsymbol{b}_t(d) \sqrt{\frac{\log(t/\alpha)}{N_{i,t}}} \right) = \mathbb{P}\left( \hat{\boldsymbol{r}}_{i,t}(d) - \boldsymbol{\mu}_i(d) > \sqrt{\frac{\log(t/\alpha)}{N_{i,t}}} \right)$$

$$\leq \exp\left( \frac{-2N_{i,t}^2 \log(t/\alpha)}{N_{i,t} \sum_{\iota=1}^{N_{i,t}} (1-0)^2} \right)$$

$$= \exp\left( -2\log(t/\alpha) \right) = \left( \frac{\alpha}{t} \right)^2,$$

(19)

which yields the upper bound of $\mathbb{P}(A_t)$ as

$$\mathbb{P}(A_t) \leq \sum_{d \in \mathcal{D}_T^+} \mathbb{P}\left( \boldsymbol{b}_t(d)\hat{\boldsymbol{r}}_{i,t}(d) > \boldsymbol{b}_t(d)\boldsymbol{\mu}_i(d) + \boldsymbol{b}_t(d)\sqrt{\frac{\log(t/\alpha)}{N_{i,t}}} \right) \leq |\mathcal{B}_T^+|\left(\frac{\alpha}{t}\right)^2, \qquad (20)$$

and similarly,

$$\mathbb{P}(B_t) \leq \sum_{d \in \mathcal{D}_T^+} \mathbb{P}\left( \boldsymbol{b}_t(d)\hat{\boldsymbol{r}}_{\tilde{a}_t^*,t}(d) < \boldsymbol{b}_t(d)\boldsymbol{\mu}_{\tilde{a}_t^*}(d) - \boldsymbol{b}_t(d)\sqrt{\frac{\log(t/\alpha)}{N_{\tilde{a}_t^*,t}}} \right) \leq |\mathcal{B}_T^+|\left(\frac{\alpha}{t}\right)^2. \qquad (21)$$

Next we investigate the event $\Gamma_t := \left\{ \boldsymbol{b}_t^T \Delta_i < 2\|\boldsymbol{b}_t\|_1 \sqrt{\frac{\log(t/\alpha)}{N_{i,t}}}, N_{i,t} > \beta \right\}$. Let $\beta = \frac{4M^2 \log(T/\alpha)}{L_i^2}$. Since $N_{i,t} \geq \beta$ and recall that $\boldsymbol{b}_t^T (\boldsymbol{\mu}_{\tilde{a}_t^*} - \boldsymbol{\mu}_i) \geq L_i$, we have,

$$2\|\boldsymbol{b}_t\|_1 \sqrt{\frac{\log(t/\alpha)}{N_{i,t}}} \leq 2\|\boldsymbol{b}_t\|_1 \sqrt{\frac{\log(t/\alpha)}{\beta}} \leq 2M\sqrt{\frac{\log(T/\alpha)}{\beta}} = L_i \leq \boldsymbol{b}_t^T (\boldsymbol{\mu}_{\tilde{a}_t^*} - \boldsymbol{\mu}_i), \qquad (22)$$

implying that the event $\Gamma_t$ has $\mathbb{P}$-probability 0. By combining Eq. 16 with Eq. 17, 20 and 21, the expectation of LHS term in Eq. 24 can be upper-bounded as follows:

$$\mathbb{E}\left[ \sum_{t \in \mathcal{M}_i^o} \mathbb{1}_{\{a_t=i\}} \right] = \mathbb{E}\left[ \sum_{t=1}^T \mathbb{1}_{\{a_t=i\neq\tilde{a}_t^*\}} \right]$$

$$\leq \frac{4M^2 \log(T/\alpha)}{L_i^2} + |\mathcal{B}_T^+|\alpha^2 \sum_{t=1}^T t^{-2} \qquad (23)$$

$$\underset{(a)}{\leq} \frac{4M^2 \log(T/\alpha)}{L_i^2} + |\mathcal{B}_T^+|\frac{\pi^2\alpha^2}{3},$$

where (a) holds by the convergence of sum of reciprocals of squares that $\sum_{t=1}^\infty t^{-2} = \frac{\pi^2}{6}$. This concludes the proof.

$\square$

## C.2 REMARKS OF THEOREM 2

**Remark C.1.** *If the distribution of $\boldsymbol{c}_t$ is stationary with known $\overline{\boldsymbol{c}}$, each arm can be viewed as having a stationary reward distributed with mean of $\overline{\boldsymbol{c}}^T \boldsymbol{\mu}_i \in \mathbb{R}$, and the goal is to maximizing accumulative reward. This reduces the problem to a standard MAB framework. By treating $\eta_i^\uparrow = \eta_i^\downarrow = \overline{\boldsymbol{c}}^T \Delta_i$ as the reward gap between arm $i$ and the best arm $a^*$, and $\delta$ as the upper-bound of reward $\boldsymbol{c}_t^T \boldsymbol{r}_{a_t,t}$ in each round $t$, our result in Theorem 2 matches the typical UCB bounds (Auer et al., 2002).*

**Remark C.2.** *Interestingly, the standard stochastic MAB can also be seen as a special case of PAMO-MAB with known preferences. Specifically, a $K$-armed stochastic bandit with reward means $x_1, \ldots, x_K$ is equivalent to the MO-MAB case where $\exists j \in [D]$ s.t., $\boldsymbol{\mu}_i(j) = x_i, \forall i \in [K]$ and $\overline{\boldsymbol{c}}_t = \boldsymbol{e}_j$ (the $j$-th standard basis vector). In this case, obviously the best arm $a_t^* = \arg\max_{i\in[K]} x_i$. Note $|\overline{\mathcal{C}}_T^+| = 1$, $\eta_i^\downarrow = \eta_i^\uparrow = \Delta_i(j) = x_{a_t^*} - x_i$, the result in Theorem 2 can be rewrite as: $R(T) \leq \sum_{i=1}^K \frac{4}{x_{a_t^*}-x_i} \log\left(\frac{T}{\alpha}\right) + O(1)$, which recovers the bound in standard MAB (Auer et al., 2002).*

Specifically, the remarks above illustrate that under stationary and known preference environments, by introducing the preference-aware optimization, PAMO-MAB can be related to a standard MAB and is solvable using conventional techniques. This insight also provides a foundation for the algorithm design and regret analysis in the unknown preference cases, where we will show that under precise preference estimation, the unknown preference problem can be reduced to the known case but narrowed overall-reward gap.

# D ANALYSES FOR SECTION 6 (UNKNOWN PREFERENCE)

## D.1 REGRET OF PRUCB-SPM: THEOREM 3 (STATIONARY PREFERENCE)

The presented Theorem 3 establishes the upper bound of regret $R(T)$ for PRUCB-SPM under stationary preference environment. For the convenience of the reader, we re-state some notations that will be used in the following before going to proof. In the case where both reward $\boldsymbol{r}_t$ and preference $\boldsymbol{c}_t$ follow fixed distributions with mean vectors of $\boldsymbol{\mu}$ and $\overline{\boldsymbol{c}}$, the optimal arm $a_t^* = \arg\max_{i \in [K]} \overline{\boldsymbol{c}}^T \boldsymbol{\mu}_i$ remains the same in each step, and thus we use $a^*$ to denote the optimal arm for simplicity. Let $\eta_i = \overline{\boldsymbol{c}}^T \Delta_i$ denote the expected overall-reward gap between arm $i$ and best arm $a^*$, where $\Delta_i = \mu_{a^*} - \mu_i \in \mathbb{R}^D$.

### D.1.1 PROOF SKETCH OF THEOREM 3

We analyze the expected number of times in $T$ that one suboptimal arm $i \neq a^*$ is played, denoted by $N_{i,T}$. Since regret performance is affected by both reward and preference estimates, we introduce a hyperparameter $\epsilon_t$ to quantify the accuracy of the empirical estimation $\hat{\boldsymbol{c}}_t$.

The key idea is that by using $\epsilon_t$ to measure the closeness of the preference estimation $\hat{\boldsymbol{c}}_t$ to the true expected vector $\overline{\boldsymbol{c}}$, the event of pulling a suboptimal arm can be decomposed into two disjoint sets based on whether $\hat{\boldsymbol{c}}_t$ is sufficiently accurate, as determined by $\epsilon_t$. And the parameter $\epsilon_t$ can be tuned to optimize the final regret. This decomposition allows us to address the problem of joint impact from the preference and reward estimate errors, analyzing the undesirable behaviors of leaner caused by estimation errors of reward $\hat{r}$ and preference $\hat{c}$ independently.

For suboptimal pulls induced by error of $\hat{r}$, we show that the pseudo episode set $\mathcal{M}_i$ where the sub-optimal arm $i$ is considered suboptimal under the preference estimate align with the true suboptimal episode set $[T]$, and the best arm within $\mathcal{M}_i$ is consistently identified as better than arm $i$. Using this insight, we show that this case can be transferred to a new preference known instance with a narrower overall-reward gap w.r.t $\epsilon_t$.

For suboptimal pulls due to error of $\hat{c}$, we first relax the suboptimal event set to an overall-reward estimation error set, eliminating the joint dependency on reward and preference from action $a_t$. Then we develop a tailored-made error bound (Lemma 10) on preference estimation, which transfers the original error set to a uniform imprecise estimation set on preference, such that a tractable formulation of the estimation deviation can be constructed.

### D.1.2 PROOF OF THEOREM 3

*Proof.* Let $N_{i,T}$ denote the expected number of times in $T$ that the suboptimal arm $i \neq a^*$ is played. We first analyze the upper-bound over $N_{i,T}$, and then derive the final regret $R(T)$ by $R(T) = \sum_{i \neq a^*} \Delta_i N_{i,T}$. The proof consists of several steps.

**Step-1 ($N_{i,T}$ Decomposition with Parameter $\epsilon_t$):**

For any $i \neq a^*$, any time step $t \in [T]$, with a hyper-parameter $0 < \epsilon_t \leq \eta_i$ introduced, we can formulate the the number of times the suboptimal arm $i$ is played as follows:

$$N_{i,T} = \sum_{t=1}^{T} \mathbb{1}_{\{a_t=i\}} = \underbrace{\sum_{t=1}^{T} \mathbb{1}_{\{a_t=i, \hat{\boldsymbol{c}}_t^T \mu_{a^*} > \hat{\boldsymbol{c}}_t^T \mu_i + \eta_i - \epsilon_t\}}}_{N_{i,T}^{\tilde{r}}: \text{ Suboptimal pulls caused by imprecise reward estimation}} + \underbrace{\sum_{t=1}^{T} \mathbb{1}_{\{a_t=i, \hat{\boldsymbol{c}}_t^T \mu_{a^*} \leq \hat{\boldsymbol{c}}_t^T \mu_i + \eta_i - \epsilon_t\}}}_{N_{i,T}^{\tilde{c}}: \text{ Suboptimal pullings caused by imprecise preference estimation}} \cdot$$

(24)

The technical idea behind is that by introducing $\epsilon_t$ to measure the closeness of the preference estimate $\hat{\boldsymbol{c}}_t$ to the true expected vector $\overline{\boldsymbol{c}}$ (i.e., the gap between $\hat{\boldsymbol{c}}_t^T \Delta_i$ and $\overline{\boldsymbol{c}}^T \Delta_i$), we can decouple the undesirable behaviors caused by either reward estimation error or preference estimation error. Specifically, we set $\epsilon_t = \min \left\{ \epsilon_0, \delta \|\Delta_i\|_2 \sqrt{\frac{D \log(t)}{t}} \right\}$, where $0 < \epsilon_0 \leq \eta_i$ is the parameter of proof

that can be optimized by regret, $\delta\|\Delta_i\|_2\sqrt{\frac{D\log(t)}{t}}$ asymptotically converges to 0 as $t$ increases. Let $N_{i,T}^{\widetilde{r}}$ and $N_{i,T}^{\widetilde{c}}$ denote the times of suboptimal pulling induced by imprecise reward estimation and preference estimation (shown in Eq. 24). We use $\mathbb{E}_{\epsilon_t}$ and $\mathbb{P}_{\epsilon_t}$ to denote the probability distribution and expectation under parameter $\epsilon_t$. Next, we will study these two terms separately.

**Step-2 (Bounding $N_{i,T}^{\hat{r}}$):**

Define $\mathcal{M}_i$ as the set of episodes that arm $i$ achieves suboptimal expected overall-reward under preference estimation $\hat{c}_t$, i.e., $\mathcal{M}_i := \{t \in [T] \mid i \neq \arg\max_{j\in[K]} \hat{c}_t^T \boldsymbol{\mu}_j\}$. Since for the event regarding $N_{i,T}^{\hat{r}}$, we have $\hat{c}_t^T \Delta_i > \eta_i - \epsilon_t \geq 0$ holds for all $t \in [T]$, which implies that $a^*$ still yields a better result than $i$ given the estimated preference coefficient $\hat{c}_t$ over time horizon $T$. Thus the suboptimal pulling of arm $i$ is attributed to the imprecise rewards estimations of arms. Additionally, we have $\mathcal{M}_i = [T]$ since arm $i$ is at least worse than $a^*$ under the preference estimation $\hat{c}_t$ for all episode $t \in [T]$. Hence for $N_{i,T}^{\widetilde{r}}$ we have

$$N_{i,T}^{\widetilde{r}} = \sum_{t=1}^{T} \mathbb{1}_{\{a_t=i, \hat{c}_t^T\Delta_i > \eta_i - \epsilon_t\}} = \sum_{t\in\mathcal{M}_i} \mathbb{1}_{\{a_t=i, \hat{c}_t^T\Delta_i > \eta_i - \epsilon_t\}} \tag{25}$$

Let $L_i = \min_{t\in\mathcal{M}_i}\{\max_{j\in[K]\setminus i}\{\hat{c}_t^T(\boldsymbol{\mu_j} - \boldsymbol{\mu_i})\}\}$, $\hat{\mathcal{C}}_T^+ := \{[\hat{c}_1(d), ..., \hat{c}_T(d)] \neq \mathbf{0}, \forall d \in [D]\}$ be the collection set of non-zero preference estimation sequence. Recall that PRUCB-SPM leverages $\hat{c}_t$ for overall-reward UCB optimization, i.e., $a_t = \arg\max \Phi(\hat{c}_t, \hat{r}_{i,t} + \sqrt{\frac{\log(t/\alpha)}{\max\{1,N_{i,t}\}}}\boldsymbol{e})$. By Proposition 8, we have

$$\mathbb{E}_\epsilon\left[\sum_{t\in\mathcal{M}_i} \mathbb{1}_{\{a_t=i, \hat{c}_t^T\Delta_i > \eta_i - \epsilon\}}\right] \leq \mathbb{E}\left[\sum_{t\in\mathcal{M}_i} \mathbb{1}_{\{a_t=i\}}\right] \leq \frac{4\delta^2\log\left(\frac{T}{\alpha}\right)}{L_i^2} + \frac{|\hat{\mathcal{C}}_T^+|\pi^2\alpha^2}{3}. \tag{26}$$

Additionally, since $\hat{c}_t^T\Delta_i > \eta_i - \epsilon_t \geq 0$ holds for all $t \in [T]$, it implies that

$$L_i = \min_{t\in\mathcal{M}_i}\{\max_{j\in[K]\setminus i}\{\hat{c}_t^T(\boldsymbol{\mu_j} - \boldsymbol{\mu_i})\}\} \geq \min_{t\in\mathcal{M}_i} \hat{c}_t^T\Delta_i > \eta_i - \epsilon_t \geq \eta_i - \epsilon_0.$$

Plugging above result into Eq. 26, and by $|\hat{\mathcal{C}}_T^+| \leq D$, we have the expectation of $N_{i,T}^{\widetilde{r}}$ in Eq. 24 can be upper-bounded as follows:

$$\begin{aligned}
\mathbb{E}_{\epsilon_t}\left[N_{i,T}^{\widetilde{r}}\right] &= \mathbb{E}_{\epsilon_t}\left[\sum_{t\in\mathcal{M}_i}\mathbb{1}_{\{a_t=i, \hat{c}_t^T\Delta_i > \eta_i - \epsilon_t\}}\right] \\
&\leq \frac{4\delta^2\log(T/\alpha)}{(\eta_i - \epsilon_0)^2} + D\frac{\pi^2\alpha^2}{3}.
\end{aligned} \tag{27}$$

**Step-3 (Bounding $N_{i,T}^{\widetilde{c}}$):**

We begin with stating one tailored-made preference estimation error bound which will be utilized in our derivation.

**Lemma 10.** *For any non-zero vectors $\Delta, \overline{c} \in \mathbb{R}^k$, and all $\epsilon \in \mathbb{R}$, if $\overline{c}^T\Delta > \epsilon$, then for any vector $c'$ s.t, $c'^T\Delta = \epsilon$, we have*

$$\|\overline{c} - c'\|_2 \geq \frac{\overline{c}^T\Delta - \epsilon}{\|\Delta\|_2}.$$

Please see Appendix D.1.3 for the proof of Lemma 10

Firstly we relax the instantaneous event set of $N_{i,T}^{\widetilde{c}}$ in Eq. 24 into a pure estimation error case as:

$$\left\{ a_t = i \neq a^*, \hat{\boldsymbol{c}}_t^T \mu_{a^*} \leq \hat{\boldsymbol{c}}_t^T \mu_i + \eta_i - \epsilon_t \right\} \subset \left\{ \hat{\boldsymbol{c}}_t^T \mu_{a^*} \leq \hat{\boldsymbol{c}}_t^T \mu_i + \eta_i - \epsilon_t \right\} = \left\{ \hat{\boldsymbol{c}}_t^T \Delta_i \leq \eta_i - \epsilon_t \right\}. \tag{28}$$

Then, according to Lemma 10 above, we can transfer the original overall-reward gap estimation error to the preference estimation error. More specifically, since $\overline{\boldsymbol{c}}^T \Delta_i > \eta_i - \epsilon_t$ always holds, for any $t \in (0, T]$, by applying Lemma 10, we have

$$\left\{ \hat{\boldsymbol{c}}_t^T \Delta_i \leq \eta_i - \epsilon_t \right\} \subset \left\{ \|\overline{\boldsymbol{c}} - \hat{\boldsymbol{c}}_t\|_2 \geq \frac{\overline{\boldsymbol{c}}^T \Delta_i - (\eta_i - \epsilon_t)}{\|\Delta_i\|_2} \right\}$$
$$\subset \left\{ \|\overline{\boldsymbol{c}} - \hat{\boldsymbol{c}}_t\|_2 \geq \frac{\epsilon_t}{\|\Delta_i\|_2} \right\}. \tag{29}$$

$$\implies \mathbb{P}_{\epsilon_t} \left( a_t = i \neq a^*, \hat{\boldsymbol{c}}_t^T \mu_{a^*} \leq \hat{\boldsymbol{c}}_t^T \mu_i + \eta_i - \epsilon_t \right) \leq \mathbb{P}_{\epsilon_t} \left( \|\overline{\boldsymbol{c}} - \hat{\boldsymbol{c}}_t\|_2 \geq \frac{\epsilon_t}{\|\Delta_i\|_2} \right). \tag{30}$$

Next we aim to upper-bound the RHS term of Eq. 30. Since $\epsilon_t$ follows different values at different episodes $t$, we consider it by ① $\epsilon_t = \epsilon_0$ and ② $\epsilon_t = \delta\|\Delta_i\|_2 \sqrt{\frac{D \log(t)}{t}}$ separately. Let $t_{\epsilon_0} = \min\{t' \mid \epsilon_0 \geq \delta\|\Delta_i\|_2 \sqrt{\frac{D \log(t)}{t}}, \forall t > t'\}$. Due to $\lim_{t \to \infty} \sqrt{\frac{\log t}{t}} = 0$, we have $t_{\epsilon_0}$ does exist. More specifically, by the fact that $\log(t) < t^{\frac{1}{5}}, \forall t > 0$, for any $t \geq (\frac{\sqrt{D}\delta\|\Delta_i\|_2}{\epsilon_0})^{\frac{5}{2}}$, we can derive that

$$\epsilon_0 \geq \delta\|\Delta_i\|_2 \sqrt{D t^{-\frac{4}{5}}} > \delta\|\Delta_i\|_2 \sqrt{D \frac{\log(t)}{t}} \implies t_{\epsilon_0} \leq (\frac{\sqrt{D}\delta\|\Delta_i\|_2}{\epsilon_0})^{\frac{5}{2}}.$$

where the first inequality holds by the monotonic decreasing of $\sqrt{t^{-\frac{4}{5}}}$, and the second inequality holds by $\frac{\log(t)}{t} < \frac{t^{1/5}}{t}, \forall t > 0$.

① Hence for $t \leq \lfloor t_{\epsilon_0} \rfloor$, we have $\epsilon_0 \leq \delta\|\Delta_i\|_2 \sqrt{\frac{D \log(t)}{t}}$ and thus

$$\sum_{t=1}^{\lfloor t_{\epsilon_0} \rfloor} \mathbb{P}_{\epsilon_t} \left( \|\overline{\boldsymbol{c}} - \hat{\boldsymbol{c}}_t\|_2 \geq \frac{\epsilon_t}{\|\Delta_i\|_2} \right) \underset{(a)}{=} \sum_{t=1}^{\lfloor t_{\epsilon_0} \rfloor} \mathbb{P}_{\epsilon_t} \left( \|\overline{\boldsymbol{c}} - \hat{\boldsymbol{c}}_t\|_2 \geq \frac{\epsilon_0}{\|\Delta_i\|_2} \right) \leq t_{\epsilon_0} \leq (\frac{\sqrt{D}\delta\|\Delta_i\|_2}{\epsilon_0})^{\frac{5}{2}}, \tag{31}$$

where (a) holds by the definition of $\epsilon_t$, i.e., $\forall t \leq \lfloor t_{\epsilon_0} \rfloor, \epsilon_t = \min\left\{\epsilon_0, \delta\|\Delta_i\|_2 \sqrt{\frac{D \log(t)}{t}}\right\} = \epsilon_0$.

*Please note that the probability of the event $\{\|\overline{\boldsymbol{c}} - \hat{\boldsymbol{c}}_t\|_2 \geq \frac{\epsilon_0}{\|\Delta_i\|_2}\}$ can be further bounded using tail bounds such as Hoeffding's inequality or Bernstein's inequality. And due to $\frac{\epsilon_0}{\|\Delta_i\|_2} > 0$ as a constant, the union probability over $\lfloor t_{\epsilon_0} \rfloor$ episodes can be bounded with a constant by the convergence of geometric series (as detailed in Eq. 43). However, for computational convenience and to keep the final solution concise, we simply treat the union probability as $\lfloor t_{\epsilon_0} \rfloor$ here.*

② On the other hand, for $t > \lfloor t_{\epsilon_0} \rfloor$, we have $\epsilon_0 \geq \delta\|\Delta_i\|_2 \sqrt{\frac{D \log(t)}{t}}$ holds, which yields

$$\mathbb{P}_{\epsilon_t} \left( \|\overline{\boldsymbol{c}} - \hat{\boldsymbol{c}}_t\|_2 \geq \frac{\epsilon_t}{\|\Delta_i\|_2} \right) \underset{(a)}{=} \mathbb{P}_{\epsilon_t} \left( \|\overline{\boldsymbol{c}} - \hat{\boldsymbol{c}}_t\|_2 \geq \delta\sqrt{\frac{D \log(t)}{t}} \right)$$
$$= \mathbb{P}_{\epsilon_t} \left( \sum_{d=1}^{D} (\overline{\boldsymbol{c}}(d) - \hat{\boldsymbol{c}}_t(d))^2 \geq \frac{D\delta^2 \log(t)}{t} \right) \tag{32}$$
$$\underset{(b)}{\leq} \sum_{d=1}^{D} \mathbb{P}_{\epsilon_t} \left( |\overline{\boldsymbol{c}}(d) - \hat{\boldsymbol{c}}_t(d)| \geq \delta\sqrt{\frac{\log(t)}{t}} \right)$$

where (a) holds by the definition of $\epsilon_t$, (b) holds since union bound and the fact that there must be at least one objective $d \in [D]$ satisfying $(\overline{\boldsymbol{c}}(d) - \hat{\boldsymbol{c}}_t(d))^2 \geq \frac{1}{D} \frac{D\delta^2 \log(t)}{t}$, otherwise the event would fail.

Note that for all $t \in (0, T]$, $\boldsymbol{c}_t$ follows same the distribution, and the deviation is exactly the radius of the preference confidence ellipse, thus we can use a tail bound for the confidence interval on empirical mean of i.i.d. sequence. Applying the the Hoeffding's inequality (Lemma 9), the probability for each objective $d \in [D]$ can be upper-bounded as follows:

$$\mathbb{P}_{\epsilon_t} \left( |\overline{\boldsymbol{c}}(d) - \hat{\boldsymbol{c}}_t(d)| \geq \delta \sqrt{\frac{\log(t)}{t}} \right) \leq 2 \exp \left( -\frac{2\delta^2 t^2 \log(t)}{t \sum_{\tau=1}^t \delta^2} \right) = \frac{2}{t^2}. \tag{33}$$

Plugging above result back to Eq. 32 and summing over $(\lfloor t_{\epsilon_0} \rfloor, T]$ yield

$$\sum_{t=\lfloor t_{\epsilon_0} \rfloor + 1}^T \mathbb{P}_{\epsilon_t} \left( \|\overline{\boldsymbol{c}} - \hat{\boldsymbol{c}}_t\|_2 \geq \frac{\epsilon_t}{\|\Delta_i\|_2} - \delta \sqrt{\frac{\log(t)}{t}} \right) \leq \sum_{t=\lfloor t_{\epsilon_0} \rfloor + 1}^T \frac{2D}{t^2} \leq \frac{D\pi^2}{3}, \tag{34}$$

where the first inequality holds by the convergence of sum of reciprocals of squares that $\sum_{t=1}^\infty t^{-2} = \frac{\pi^2}{6}$. By combining Eq. 31, Eq. 34 with Eq. 30, we can obtain the upper-bound for the expectation of $N_{i,T}^{\widetilde{\boldsymbol{c}}}$ in Eq. 24 as follows:

$$\begin{aligned}
\mathbb{E}_{\epsilon_t} \left[ N_{i,T}^{\widetilde{\boldsymbol{c}}} \right] &= \mathbb{E}_{\epsilon_t} \left[ \sum_{t=1}^T \mathbb{1}_{\{a_t = i \neq a^*, \hat{\boldsymbol{c}}_t^T \mu_{a^*} \leq \hat{\boldsymbol{c}}_t^T \mu_i + \eta_i - \epsilon_t\}} \right] \\
&= \sum_{t=1}^T \mathbb{P}_{\epsilon_t} \left( a_t = i \neq a^*, \hat{\boldsymbol{c}}_t^T \mu_{a^*} \leq \hat{\boldsymbol{c}}_t^T \mu_i + \eta_i - \epsilon_t \right) \\
&\leq \sum_{t=1}^{\lfloor t_{\epsilon_0} \rfloor} \mathbb{P}_{\epsilon_t} \left( \|\overline{\boldsymbol{c}} - \hat{\boldsymbol{c}}_t\|_2 \geq \frac{\epsilon_t}{\|\Delta_i\|_2} \right) + \sum_{t=\lfloor t_{\epsilon_0} \rfloor + 1}^T \mathbb{P}_{\epsilon_t} \left( \|\overline{\boldsymbol{c}} - \hat{\boldsymbol{c}}_t\|_2 \geq \frac{\epsilon_t}{\|\Delta_i\|_2} \right) \\
&\leq \left( \frac{\sqrt{D}\delta\|\Delta_i\|_2}{\epsilon_0} \right)^{\frac{5}{2}} + \frac{D\pi^2}{3} \quad \text{(by Eq. 31 and Eq. 34)}.
\end{aligned} \tag{35}$$

**Step-4 (Final $R(T)$ Derivation and Optimization over $\epsilon_0$):**

Combining Eq.24 with the corresponding upper-bounds of $\mathbb{E}_\epsilon \left[ N_{i,T}^{\widetilde{r}} \right]$ (Eq.27) and $\mathbb{E}_\epsilon \left[ N_{i,T}^{\widetilde{\boldsymbol{c}}} \right]$ (Eq.35), we can get

$$\mathbb{E}_\epsilon[N_{i,T}] \leq \frac{4\delta^2 \log(T/\alpha)}{(\eta_i - \epsilon_0)^2} + \frac{D\pi^2 \alpha^2}{3} + \left( \frac{\sqrt{D}\delta\|\Delta_i\|_2}{\epsilon_0} \right)^{\frac{5}{2}} + \frac{D\pi^2}{3}, \tag{36}$$

Note that for any $i \neq a^*$, the parameter $\epsilon_0 \in (0, \eta_i)$ can be optimally selected so as to minimize the RHS of Eq. 36. For simplicity, taking $\epsilon_0 = \frac{1}{\sqrt{D}+1}\eta_i$ yields

$$\mathbb{E}[N_{i,T}] \leq \frac{4(\delta + \frac{\delta}{\sqrt{D}})^2 \log\left(\frac{T}{\alpha}\right)}{\eta_i^2} + \frac{D\pi^2 \alpha^2}{3} + \left( \frac{(D + \sqrt{D})\delta\|\Delta_i\|_2}{\eta_i} \right)^{\frac{5}{2}} + \frac{D\pi^2}{3}.$$

Since $\sqrt{D} + D \leq 2D$ holds for all $D \geq 1$, we can replace $\sqrt{D} + D$ with $2D$ in result above for a simpler form. Multiplying the results above by the expected overall-reward gap $\eta_i$ for all suboptimal arms $i \neq a^*$ and summing them up, we can derive the regret of PRUCB-SPM follows the upper bound below,

$$R(T) \leq \sum_{i \neq a^*} \frac{4(\delta + \frac{\delta}{\sqrt{D}})^2 \log\left(\frac{T}{\alpha}\right)}{\eta_i} + \frac{D\pi^2 \alpha^2 \eta_i}{3} + \frac{4\sqrt{2}(D\delta\|\Delta_i\|_2)^{\frac{5}{2}}}{\eta_i^{3/2}} + \frac{D\pi^2 \eta_i}{3}.$$

which concludes the proof of Theorem 3. $\qquad\square$

### D.1.3 PROOF OF LEMMA 10

*Proof of Lemma 10.* Let $\phi_\epsilon$ be the set of solution such that $\boldsymbol{x}^T\Delta = \epsilon$, $\phi_{\overline{\boldsymbol{c}}^T\Delta}$ be the solution set of $\boldsymbol{x}^T\Delta = \overline{\boldsymbol{c}}^T\Delta$, i.e.,

$$\phi_\epsilon := \left\{ \boldsymbol{x} \mid \boldsymbol{x}^T\Delta = \epsilon \right\}$$
$$\phi_{\overline{\boldsymbol{c}}^T\Delta} := \left\{ \boldsymbol{x} \mid \boldsymbol{x}^T\Delta = \overline{\boldsymbol{c}}^T\Delta \right\},$$

where $\phi_\epsilon$ and $\phi_{\overline{\boldsymbol{c}}^T\Delta}$ can be viewed as two hyperplanes share the same normal vector of $\Delta$. Let $\overline{\boldsymbol{c}}_{\phi_\epsilon}$ be the projection of vector $\overline{\boldsymbol{c}}$ on hyperplane $\phi_\epsilon$. Apparently, $(\overline{\boldsymbol{c}}_{\phi_\epsilon} - \overline{\boldsymbol{c}}) \perp \phi_\epsilon$, and thus we have

$$\|\overline{\boldsymbol{c}}_{\phi_\epsilon} - \overline{\boldsymbol{c}}\|_2 = \frac{\overline{\boldsymbol{c}}^T\Delta}{\|\Delta\|_2} - \frac{\epsilon}{\|\Delta\|_2}, \tag{37}$$

which is also the distance between the parallel hyperplanes $\phi_\epsilon$ and $\phi_{\overline{\boldsymbol{c}}^T\Delta}$. By the principle of distance between points on parallel hyperplanes, we have for any $\hat{\boldsymbol{c}} \in \phi_\epsilon$, the distance between $\hat{\boldsymbol{c}}$ and $\overline{\boldsymbol{c}}$ is always greater than or equal to the shortest distance between the hyperplanes $\phi_\epsilon$ and $\phi_{\overline{\boldsymbol{c}}^T\Delta}$, i.e.,

$$\|\hat{\boldsymbol{c}} - \overline{\boldsymbol{c}}\|_2 \geq \|\overline{\boldsymbol{c}}_{\phi_\epsilon} - \overline{\boldsymbol{c}}\|_2 = \frac{\overline{\boldsymbol{c}}^T\Delta - \epsilon}{\|\Delta\|_2} \tag{38}$$

$\square$

### D.2 PROOF OF THEOREM 4 (STATIONARY PREFERENCE UNDER STOCHASTIC CORRUPTION)

*Proof.* Let $N_{i,T}$ denotes expected number of times each suboptimal arm $i \neq a^*$ being pulled under statistical preference corruptions $\boldsymbol{z}_t$ within $T$ time horizon. We first analyze $N_{i,T}$ and then derive the final regret bound of $R(T)$. The proof follows similar steps as Theorem 3 in Appendix D.1.2.

**Step-1 ($N_{i,T}$ Decomposition with Parameter $\epsilon_t$):**

Similarly, we leverage a parameter $\epsilon_t$ measuring the estimation accuracy of $\hat{\boldsymbol{c}}_t$, and decompose the suboptimal arm pulling event into two disjoint sets by whether the preference estimation $\hat{\boldsymbol{c}}_t$ is sufficiently precise, as quantified by $\epsilon_t > 0$. In this case, we set $\epsilon_t$ as a constant: $\epsilon_t = \epsilon$, and decompose the $N_{i,T}$ as follow:

$$N_{i,T} = \sum_{t=1}^T \mathbb{1}_{\{a_t=i\neq a^*\}} = \underbrace{\sum_{t=1}^T \mathbb{1}_{\{a_t=i\neq a^*, \hat{\boldsymbol{c}}_t^T\mu_{a^*} > \hat{\boldsymbol{c}}_t^T\mu_i + \eta_i - \epsilon\}}}_{N_{i,T}^{\tilde{r}}: \text{ Suboptimal pulls caused by imprecise } \textbf{reward estimation}} + \underbrace{\sum_{t=1}^T \mathbb{1}_{\{a_t=i\neq a^*, \hat{\boldsymbol{c}}_t^T\mu_{a^*} \leq \hat{\boldsymbol{c}}_t^T\mu_i + \eta_i - \epsilon\}}}_{N_{i,T}^{\tilde{c}}: \text{ Suboptimal pulls caused by imprecise } \textbf{preference estimation}} \cdot$$

Please note that in this case, the empirical estimation of preference is computed by the potentially manipulated preference feedback by corruption attacker (stochastic or adversarial), i.e.,

$$\hat{\boldsymbol{c}}_t = \frac{1}{t}\sum_{\tau=1}^t \widetilde{\boldsymbol{c}}_\tau = \frac{1}{t}\sum_{\tau=1}^t (\boldsymbol{c}_\tau + \boldsymbol{z}_\tau).$$

**Step-2 (Bounding $N_{i,T}^{\tilde{r}}$):**

Since term $N_{i,T}^{\tilde{r}}$ counts the number of undesired pulls of suboptimal arm $i \neq a^*$ under the assumption of $\hat{\boldsymbol{c}}_t^T\Delta_i > \eta_i - \epsilon > 0$. In this case, $a^*$ is still a better arm than i given the estimated preference vector $\hat{\boldsymbol{c}}_t$ though it was corrupted either by stochastic or adversarial corruptions. Thus it is easy to verify that the result of $N_{i,T}^{\tilde{r}}$ (Eq. 27) in proof of Theorem 3 (Appendix D.1.2, Step-1) still holds under both stochastic and adversarial corruptions, i.e.,

$$\mathbb{E}_\epsilon\left[N_{i,T}^{\tilde{r}}\right] \leq \frac{4\delta^2 \log(T/\alpha)}{(\eta_i - \epsilon)^2} + |\mathcal{C}_T^+|\frac{\pi^2\alpha^2}{3}. \tag{39}$$

**Step-3 (Bounding $N_{i,T}^{\tilde{c}}$):**

We begin with stating one concentration bound that will be utilized in our derivation. Please see Appendix D.2.2 for the proof of Lemma 11.

**Lemma 11** (Variant of Bernstein's inequality). *Let $\{X_1, ..., X_m\}$ be non-negative and independent, identically distributed random variables, with expected value of $\mathbb{E}[X]$ and variance of $\mathbb{V}\mathrm{ar}[X]$. Suppose that $X_i \leq M$ almost surely for all $i$. Then, for any positive $\epsilon$,*

$$\mathbb{P}\left(\left|\frac{1}{m}\sum_{i=1}^{m} X_i - \mathbb{E}[X]\right| \geq \epsilon\right) \leq 2\exp\left(\frac{-\epsilon^2 m}{2\mathbb{V}\mathrm{ar}[X] + \frac{2}{3}M\epsilon}\right).$$

Please see Appendix D.2.2 for the proof of Lemma 11.

By relaxing the the original event set and applying Lemma 10, we have:

$$\left\{a_t = i \neq a^*, \hat{\boldsymbol{c}}_t^T \mu_{a^*} \leq \hat{\boldsymbol{c}}_t^T \mu_i + \eta_i - \epsilon\right\} \subset \left\{\hat{\boldsymbol{c}}_t^T \Delta_i \leq \eta_i - \epsilon\right\}$$

$$\subset \left\{\|\overline{\boldsymbol{c}} - \hat{\boldsymbol{c}}_t\|_2 \geq \frac{\overline{\boldsymbol{c}}^T \Delta_i - (\eta_i - \epsilon)}{\|\Delta_i\|_2}\right\}$$

$$\underset{(a)}{\subset} \left\{\|\overline{\boldsymbol{c}} + \overline{\boldsymbol{z}} - \hat{\boldsymbol{c}}_t\|_2 + \|\overline{\boldsymbol{z}}\|_2 \geq \frac{\overline{\boldsymbol{c}}^T \Delta_i - (\eta_i - \epsilon)}{\|\Delta_i\|_2}\right\}$$

$$= \left\{\|\overline{\boldsymbol{c}} + \overline{\boldsymbol{z}} - \hat{\boldsymbol{c}}_t\|_2 \geq \frac{\epsilon}{\|\Delta_i\|_2} - \|\overline{\boldsymbol{z}}\|_2\right\},$$

where (a) holds by the triangle inequality that

$$\|\overline{\boldsymbol{c}} - \hat{\boldsymbol{c}}_t\|_2 = \|\overline{\boldsymbol{c}} + \overline{\boldsymbol{z}} - \hat{\boldsymbol{c}}_t - \overline{\boldsymbol{z}}\|_2 \leq \|\overline{\boldsymbol{c}} + \overline{\boldsymbol{z}} - \hat{\boldsymbol{c}}_t\|_2 + \|\overline{\boldsymbol{z}}\|_2.$$

Thus we have

$$\mathbb{P}_\epsilon\left(a_t = i \neq a^*, \hat{\boldsymbol{c}}_t^T \mu_{a^*} \leq \hat{\boldsymbol{c}}_t^T \mu_i + \eta_i - \epsilon\right) \leq \mathbb{P}_\epsilon\left(\|\overline{\boldsymbol{c}} + \overline{\boldsymbol{z}} - \hat{\boldsymbol{c}}_t\|_2 \geq \frac{\epsilon - \|\Delta_i\|_2\|\overline{\boldsymbol{z}}\|_2}{\|\Delta_i\|_2}\right)$$

$$= \mathbb{P}_\epsilon\left(\sum_{d=1}^{D}\left((\overline{\boldsymbol{c}} + \overline{\boldsymbol{z}})(d) - \hat{\boldsymbol{c}}_t(d)\right)^2 \geq \frac{\epsilon - \|\Delta_i\|_2\|\overline{\boldsymbol{z}}\|_2}{\|\Delta_i\|_2}\right)$$

$$\leq \sum_{d=1}^{D} \mathbb{P}_\epsilon\left(\left|(\overline{\boldsymbol{c}} + \overline{\boldsymbol{z}})(d) - \hat{\boldsymbol{c}}_t(d)\right| \geq \frac{\epsilon - \|\Delta_i\|_2\|\overline{\boldsymbol{z}}\|_2}{\sqrt{D}\|\Delta_i\|_2}\right). \tag{40}$$

where the last inequality holds by the union bound and the fact that there must exist at least one dimension $d \in [D]$ satisfying $\left((\overline{\boldsymbol{c}} + \overline{\boldsymbol{z}})(d) - \hat{\boldsymbol{c}}_t(d)\right)^2 \geq \frac{\epsilon - \|\Delta_i\|_2\|\overline{\boldsymbol{z}}\|_2}{D\|\Delta_i\|_2}$, otherwise the event would fail. Recall that $\boldsymbol{c}_t$ and $\boldsymbol{z}_t$ are independent, for all $t \in (0, T]$, $\widetilde{\boldsymbol{c}}_t(d) = \boldsymbol{c}_t(d) + \boldsymbol{z}_t(d), \forall d \in [D]$ follows the distribution as the convolution of the distributions of $\boldsymbol{c}_t(d)$ and $\boldsymbol{z}_t(d)$, which has the mean and variance of $\overline{\boldsymbol{c}}(d) + \overline{\boldsymbol{z}}(d)$ and $\sigma_c^2 + \sigma_z^2$ respectively. By the definition of $\hat{\boldsymbol{c}}_t$ in PUCB-SPM, we can apply a tail bound to upper bound the probability (in Eq. 40) that the empirical mean $\hat{\boldsymbol{c}}_t(d)$ of bounded random variables $\widetilde{\boldsymbol{c}}_t(d)$ deviates from its expected value $\overline{\boldsymbol{c}}(d) + \overline{\boldsymbol{z}}(d)$. Let $B_{\epsilon,i} = \epsilon - \|\overline{\boldsymbol{z}}\|_2\|\Delta_i\|_2$, next we consider two cases as follows.

**Case ①:** $B_{\epsilon,i} \leq 0$. In this case, it is evident that $|(\overline{\boldsymbol{c}} + \overline{\boldsymbol{z}})(d) - \hat{\boldsymbol{c}}_t(d)| \geq 0 \geq \frac{\epsilon - \|\Delta_i\|_2\|\overline{\boldsymbol{z}}\|_2}{\sqrt{D}\|\Delta_i\|_2} = \frac{B_{\epsilon,i}}{\sqrt{D}\|\Delta_i\|_2}$ strictly holds for all $t \in (0, T]$, indicating that $\mathbb{P}_\epsilon\left(a_t = i \neq a^*, \hat{\boldsymbol{c}}_t^T \mu_{a^*} \leq \hat{\boldsymbol{c}}_t^T \mu_i + \eta_i - \epsilon\right) = 1$. Summing over $T$ derives the result that

$$\mathbb{E}_\epsilon\left[N_{i,T}^{\widetilde{c}}\right] = \mathbb{E}_\epsilon\left[\sum_{t=1}^{T} \mathbb{1}_{\{a_t = i \neq a^*, \hat{\boldsymbol{c}}_t^T \mu_{a^*} \leq \hat{\boldsymbol{c}}_t^T \mu_i + \eta_i - \epsilon\}}\right]$$

$$= \sum_{t=1}^{T} \mathbb{P}_\epsilon\left(a_t = i \neq a^*, \hat{\boldsymbol{c}}_t^T \mu_{a^*} \leq \hat{\boldsymbol{c}}_t^T \mu_i + \eta_i - \epsilon\right) = \Omega(T). \tag{41}$$

**Case ②:** $B_{\epsilon,i} > 0$. Since $B_{\epsilon,i}$ is a constant deviation, by applying the the variant of Bernstein's inequality (Lemma 11) on event $\{|(\overline{\boldsymbol{c}} + \overline{\boldsymbol{z}})(d) - \hat{\boldsymbol{c}}_t(d)| \geq \frac{B_{\epsilon,i}}{\sqrt{D}\|\Delta_i\|_2}\}$, the probability for each objective

$d \in [D]$ can be upper-bounded as follows:

$$\mathbb{P}_\epsilon \left( |(\overline{c} + \overline{z})(d) - \hat{c}_t(d)| \geq \frac{B_{\epsilon,i}}{\sqrt{D} \|\Delta_i\|_2} \right) \leq 2 \exp \left( -\frac{B_{\epsilon,i}^2}{2D\|\Delta_i\|_2^2(\sigma_c^2 + \sigma_z^2) + \frac{2}{3}(\delta + \delta_z)B_{\epsilon,i}\sqrt{D}\|\Delta_i\|_2} t \right),$$

where $\sigma_c^2$ and $\sigma_z^2$ are the variance upper-bounds of preference and corruption distributions for each objective, $\delta$ and $\delta_z$ are the upper-bounds of $\|c_t\|_1$ and $\|z_t\|_1$. Plugging back to Eq.40 yields the result of

$$\mathbb{P}_\epsilon \left( a_t = i \neq a^*, \hat{c}_t^T \mu_{a^*} \leq \hat{c}_t^T \mu_i + \eta_i - \epsilon \right)$$

$$\leq 2D \exp \left( -\frac{B_{\epsilon,i}^2}{2D\|\Delta_i\|_2^2(\sigma_c^2 + \sigma_z^2) + \frac{2}{3}(\delta + \delta_z)B_{\epsilon,i}\sqrt{D}\|\Delta_i\|_2} t \right).$$

Summing over $T$ derives the upper-bound for the expectation of $N_{i,T}^{\tilde{c}}$ under stochastic corruptions:

$$\mathbb{E}_\epsilon \left[ N_{i,T}^{\tilde{c}} \right] = \mathbb{E}_\epsilon \left[ \sum_{t=1}^T \mathbb{1}_{\{a_t = i \neq a^*, \hat{c}_t^T \mu_{a^*} \leq \hat{c}_t^T \mu_i + \eta_i - \epsilon\}} \right]$$

$$= \sum_{t=1}^T \mathbb{P}_\epsilon \left( a_t = i \neq a^*, \hat{c}_t^T \mu_{a^*} \leq \hat{c}_t^T \mu_i + \eta_i - \epsilon \right)$$

$$\leq 2D \sum_{t=1}^T \exp \left( -\frac{B_{\epsilon,i}^2}{2D\|\Delta_i\|_2^2(\sigma_c^2 + \sigma_z^2) + \frac{2}{3}(\delta + \delta_z)B_{\epsilon,i}\sqrt{D}\|\Delta_i\|_2} t \right) \qquad (42)$$

$$\overset{(a)}{\leq} \frac{2D}{\exp \left( \frac{B_{\epsilon,i}^2}{2D\|\Delta_i\|_2^2(\sigma_c^2 + \sigma_z^2) + \frac{2}{3}(\delta + \delta_z)B_{\epsilon,i}\sqrt{D}\|\Delta_i\|_2} \right) - 1}$$

$$\leq \frac{4D^2\|\Delta_i\|_2^2(\sigma_c^2 + \sigma_z^2)}{B_{\epsilon,i}^2} + \frac{4D^{\frac{3}{2}}(\delta + \delta_z)\|\Delta_i\|_2}{3B_{\epsilon,i}} \quad \text{(by } e^x \geq x + 1, \forall x \geq 0\text{)}.$$

where (a) holds since for any $a > 0$, we have

$$\sum_{t=1}^T \left( e^{-a} \right)^t = \sum_{t=0}^{T-1} e^{-a} \cdot \left( e^{-a} \right)^t \leq \sum_{t=0}^\infty e^{-a} \cdot \left( e^{-a} \right)^t$$

$$= \frac{e^{-a}}{1 - e^{-a}} \quad \text{(by closed form of the geometric series)} \qquad (43)$$

$$= \frac{1}{e^a - 1}.$$

**Step-4 (Final Derivation and Trade-Off over $\epsilon_0$):**

Combining the results of Eq. 39, Eq. 41 and Eq. 42 yields

① if $\exists i \neq a^*$, s.t., $B_{\epsilon,i} \leq 0$, then $\mathbb{E}_\epsilon[N_{i,T}] = \Omega(T)$;

② else if $B_{\epsilon,i} > 0, \forall i \neq a^*$, then

$$\mathbb{E}_\epsilon[N_{i,T}] \leq \underbrace{\frac{4\delta^2 \log\left(\frac{T}{\alpha}\right)}{(\eta_i - \epsilon)^2} + \frac{D\pi^2\alpha^2}{3}}_{\substack{\textit{Suboptimal pulls caused by imprecise} \\ \textit{reward estimation}}} + \underbrace{\frac{4D^2\|\Delta_i\|_2^2(\sigma_c^2 + \sigma_z^2)}{B_{\epsilon,i}^2} + \frac{4D^{\frac{3}{2}}(\delta + \delta_z)\|\Delta_i\|_2}{3B_{\epsilon,i}}}_{\substack{\textit{Suboptimal pulls caused by} \\ \textit{imprecise \textbf{preference} estimation}}},$$

Note that the RHS of result above can be minimized by selecting an appropriate $\epsilon$. Moreover, there is a trade-off between robust tolerance to the corruption level $z$ and the final regret. Specifically, a

larger $\epsilon$ provides a more robust threshold to the corruption $\boldsymbol{z}$ due to the increased $B_{\epsilon,i}$. However, this would also lead to higher regret caused by error from the reward estimation.

For both satisfied final regret and robust performance, we set $\epsilon = \frac{\eta_i}{1+\frac{1}{D}}$ and thus $B_i = \frac{\eta_i}{1+\frac{1}{D}} - \|\overline{\boldsymbol{z}}\|_2 \|\Delta_i\|_2$. Therefore, if $B_i > 0, \forall i \neq a^*$, then we have

$$
\mathbb{E}[N_{i,T}] \leq \frac{4(D+1)^2 \delta^2 \log\left(\frac{T}{\alpha}\right)}{\eta_i^2} + \frac{D\pi^2 \alpha^2}{3} + \frac{4D^2 \|\Delta_i\|_2^2 (\sigma_c^2 + \sigma_z^2)}{B_i^2} + \frac{4D^{\frac{3}{2}}(\delta + \delta_z)\|\Delta_i\|_2}{3B_i},
$$

otherwise $\mathbb{E}[N_{i,T}] = \Omega(T)$.

Multiplying the results above by the expected overall-reward gap $\eta_i$ for all suboptimal arms $i \neq a^*$ and summing them up yields the final regret $R(T)$ upper bound in Theorem 4.

$\square$

### D.2.1 TIGHTNESS OF ATTACK TOLERANCE

**Remark D.1** (Tightness of attack tolerance). *Theorem 4 shows a tight attack tolerance threshold for PRUCB-SPM against stochastic preference attack. Note that there exists a minimax lower bound for the attack tolerance: if $\eta_i - |\overline{\boldsymbol{z}}^T \Delta_i| \leq 0$, then for any policy $\pi$, $\inf_\pi \sup_{\mathcal{C} \times \mathcal{R}} R(T) = \Omega(T)$, since in this case, there exists a set of $\overline{\boldsymbol{z}}$ that can close the overall-reward gap between arms $i$ and $a^*$, making arm $i$ appear optimal over $a^*$. Our algorithm presents a slightly relaxed threshold $B_i = \eta_i/(1+1/D) - \|\overline{\boldsymbol{z}}\|_2 \|\Delta_i\|_2$. Here, $\eta_i/(1+1/D)$ acts as a lower confidence bound for the overall-reward gap $\eta_i$ due to preference estimation error. By Cauchy–Schwarz inequality, $\|\overline{\boldsymbol{z}}\|_2 \|\Delta_i\|_2$ is an upper bound for $|\overline{\boldsymbol{z}}^T \Delta_i|$. This implies that the attack tolerance $B_i$ of PRUCB-SPM matches the attack tolerance in minimax lower bound up to a constant factor of $1/(1+1/D)$.*

### D.2.2 PROOF OF LEMMA 11

**Lemma 12** (Bernstein inequality for bounded distributions (Vershynin, 2018) (Theorem 2.8.4)). *Given independent zero-mean random variables $\{X_1, ..., X_m\}$ where $|X_i| \leq M$ almost surely (with probability 1) for all $i$, then for all positive $\epsilon$:*

$$
\mathbb{P}\left(\sum_{i=1}^m X_i \geq \epsilon\right) \leq \exp\left(\frac{-\frac{1}{2}\epsilon^2}{\sum_{i=1}^m \mathbb{E}[X_i^2] + \frac{1}{3}M\epsilon}\right). \tag{44}
$$

*Proof of Lemma 11.* Let $Y_i = X_i - \mathbb{E}[X_i]$, apparently $Y_1, ..., Y_m$ are i.i.d. random variables with zero mean, and for all $i$, $|Y_i| \leq M$ almost surely. By plugging $Y_i$ into Eq. 44 (Lemma 12), for any positive $\epsilon_0$ we have

$$
\mathbb{P}\left(\sum_{i=1}^m Y_i \geq \epsilon_0\right) \leq \exp\left(\frac{-\frac{1}{2}\epsilon_0^2}{\sum_{i=1}^m \mathbb{E}[Y_i^2] + \frac{1}{3}M\epsilon_0}\right). \tag{45}
$$

$$
\implies \mathbb{P}\left(\sum_{i=1}^m (X_i - \mathbb{E}[X_i]) \geq \epsilon_0\right) \leq \exp\left(\frac{-\frac{1}{2}\epsilon_0^2}{\sum_{i=1}^m \mathbb{E}[(X_i - \mathbb{E}[X_i])^2] + \frac{1}{3}M\epsilon_0}\right)
$$
$$
\underset{(a)}{=} \exp\left(\frac{-\frac{1}{2}\epsilon_0^2}{m\mathbb{V}\mathrm{ar}[X] + \frac{1}{3}M\epsilon_0}\right). \tag{46}
$$

where (a) holds since $\mathbb{E}[(X_i - \mathbb{E}[X_i])^2] = \mathbb{E}[X_i^2 - 2X_i\mathbb{E}[X_i] + \mathbb{E}[X_i]^2] = \mathbb{E}[X_i^2] - \mathbb{E}[X_i]^2 = \mathbb{V}\mathrm{ar}[X_i]$. Let $\epsilon = \frac{\epsilon_0}{m}$, we have

$$\mathbb{P}\left(\frac{1}{m}\sum_{i=1}^{m}X_i - \mathbb{E}[X] \geq \epsilon\right) = \mathbb{P}\left(\sum_{i=1}^{m}(X_i - \mathbb{E}[X_i]) \geq \epsilon_0\right)$$

$$\leq \exp\left(\frac{-\frac{1}{2}(m\epsilon)^2}{m\mathbb{V}\mathrm{ar}[X] + \frac{1}{3}Mm\epsilon}\right) \tag{47}$$

$$= \exp\left(\frac{-\epsilon^2 m}{2\mathbb{V}\mathrm{ar}[X] + \frac{2}{3}M\epsilon}\right).$$

Then using the symmetry property of confidence interval, we can derive the desired result as Lemma 11. $\qquad\square$

### D.3 ANALYSIS FOR STATIONARY PREFERENCE UNDER ADVERSARIAL CORRUPTION

#### D.3.1 ADVERSARY FOR STOCHASTIC PREFERENCE-AWARE MO-MAB

In this section, we consider the preference with stationary distributions but under arbitrary adversarial corruptions. We inherit the assumptions in Theorem 3 but define an adversary that would alter the preference observations. Specifically, the user preference on each objective is independently drawn from a fixed and unknown distribution, and the preference observation after each episode is then possibly manipulated by an adversary: $\tilde{c}_t = c_t + z_t$, with $z_t$ denoting the adversarial corruption component.

Formally, the protocol between learner and adversary, at each round $t = 1...T$, is as follows:

1. Stochastic preference $c_t(d)$ on each objective $d \in [D]$ is independently drawn from a stationary distribution $\mathcal{F}_d^c$, stochastic reward $r_{i,t}(d)$ is drawn independently from a stationary distribution $\mathcal{F}_{i,d}^r$ for each arm $i \in [K]$ and each objective $d \in [D]$.

2. The learner computes a distribution $\omega_t$ over $K$ arms under historical reward and preference observations, and picks arm $a_t \sim \omega_t$ for acting.

3. (*Attacker*) The adversary returns a corrupted preference vector $\tilde{c}_t = c_t + z_t$, where $z_t$ is the adversarial corruption component.

4. The learner observes reward $r_t$ and corrupted preference feedback $\tilde{c}_t$.

**Corruption Budget.** We refer to $\|\tilde{c}_t - c_t\|_1$ as the amount of corruption injected in round $t$. The total attack budget of the adversary is given by $\sum_{t=1}^{T}\|\tilde{c}_t - c_t\|_1 = \sum_{t=1}^{T}\|z_t\|_1 \leq Z$.

In Theorem D.2 below, we we provide the regret performance of PRUCB-SPM under the adversarial corruptions. The proof of which is provided in Appendix D.3.2.

**Theorem 13** (Regret). *Inherit the assumptions in Theorem 3 but the revealed feedback after episode is under adversarial corruptions. For any attack budget $Z \geq 0$, PRUCB-SPM has*

$$R(T) \leq \sum_{i \neq a^*}\left(\frac{4(\delta + \frac{\delta}{\sqrt{D}})^2}{\eta_i}\log(\frac{T}{\alpha}) + \frac{2(1 + \sqrt{D})}{\|\Delta_i\|_2}Z\right) + O(1).$$

**Remark D.2.** *Theorem implies that the algorithm PRUCB-SPM attains a sub-linear regret as long as the adversarial corruption budget level $Z = o(T)$. In particular, PRUCB-SPM achieves the same order of regret as the uncorrupted setting when $Z = O(\log T)$. It effectively demonstrates that PRUCB-SPM also has strong robustness against adversarial attack.*

#### D.3.2 PROOF OF THEOREM D.2 (ADVERSARIAL CORRUPTIONS)

*Proof.* Let $N_{i,T}$ be the expected number of times each suboptimal arm $i$ being pulled under adversarial preference corruption $z_t$ within $T$ time horizon. We first analyze the performance regarding $N_{i,T}$, and then extend the solution to the final regret $R(T)$. The proof is similar with the case of stochastic corruption.

**Step-1 ($N_{i,T}$ Decomposition with Parameter $\epsilon$):**

Firstly, we decompose the suboptimal arm pulling event using parameter $\epsilon > 0$ as:

$$N_{i,T} = \sum_{t=1}^{T} \mathbb{1}_{\{a_t=i\neq a^*\}} = \underbrace{\sum_{t=1}^{T} \mathbb{1}_{\{a_t=i\neq a^*, \hat{c}_t^T \mu_{a^*} > \hat{c}_t^T \mu_i + \eta_i - \epsilon\}}}_{N_{i,T}^{\tilde{r}}: \text{ \textit{Suboptimal pulls caused by imprecise}} \atop \textbf{\textit{reward estimation}}} + \underbrace{\sum_{t=1}^{T} \mathbb{1}_{\{a_t=i\neq a^*, \hat{c}_t^T \mu_{a^*} \leq \hat{c}_t^T \mu_i + \eta_i - \epsilon\}}}_{N_{i,T}^{\tilde{c}}: \text{ \textit{Suboptimal pulls caused by}} \atop \text{\textit{imprecise} \textbf{\textit{preference estimation}}}}.$$

where the empirical estimation of preference is computed by the potentially manipulated preference feedback by adversarial attacker, i.e.,

$$\hat{c}_t = \frac{1}{t} \sum_{\tau=1}^{t} \widetilde{c}_\tau = \frac{1}{t} \sum_{\tau=1}^{t} (c_\tau + z_\tau).$$

**Step-2 (Bounding $N_{i,T}^{\tilde{r}}$):**

From the analysis of Theorem 4, we have that the result of $N_{i,T}^{\tilde{r}}$ in Eq. 39 (**Step-1**, Appendix D.2) still holds under both stochastic and adversarial corruptions, and thus

$$\mathbb{E}_\epsilon \left[ N_{i,T}^{\tilde{r}} \right] \leq \frac{4\delta^2 \log(T/\alpha)}{(\eta_i - \epsilon)^2} + |\mathcal{C}_T^+| \frac{\pi^2 \alpha^2}{3}. \tag{48}$$

**Step-3 (Bounding $N_{i,T}^{\tilde{c}}$):**

Let $\hat{c}_t^{\mathcal{S}}$ and $\hat{c}_t$ be the empirical mean vector of the stochastic ground-truth preference and the empirical mean vector of the actual (adversely corrupted) preference feedback after $t$ episodes respectively. By relaxing the the original event set and applying Lemma 10, we have:

$$
\begin{aligned}
\left\{ a_t = i \neq a^*, \hat{c}_t^T \mu_{a^*} \leq \hat{c}_t^T \mu_i + \eta_i - \epsilon \right\} &\subset \left\{ \hat{c}_t^T \Delta_i \leq \eta_i - \epsilon \right\} \\
&\subset \left\{ \|\overline{c} - \hat{c}_t\|_2 \geq \frac{\overline{c}^T \Delta_i - (\eta_i - \epsilon)}{\|\Delta_i\|_2} \right\} \\
&= \left\{ \|\overline{c} - \hat{c}_t^{\mathcal{S}} + \hat{c}_t^{\mathcal{S}} - \hat{c}_t\|_2 \geq \frac{\epsilon}{\|\Delta_i\|_2} \right\} \\
&\underset{(a)}{\subseteq} \left\{ \underbrace{\|\overline{c} - \hat{c}_t^{\mathcal{S}}\|_2}_{\text{Term 1}} + \underbrace{\|\hat{c}_t^{\mathcal{S}} - \hat{c}_t\|_2}_{\text{Term 2}} \geq \frac{\epsilon}{\|\Delta_i\|_2} \right\},
\end{aligned}
\tag{49}
$$

where (a) holds by the triangle inequality. Next we analyze the probabilities regarding two terms separately.

*Step-3-i (Bounding Term 1):* For Term 1, recall that $\overline{c}$ is the mean of statistic ground-truth preference $c_t$, we can thus establish the probability upper-bound on the event of Term 1 using the tail bound.

Specifically, by the variant of Bernstein's inequality (Lemma 11), we have

$$
\begin{aligned}
\mathbb{P}_\epsilon \left( \underbrace{\|\overline{c} - \hat{c}_t^{\mathcal{S}}\|_2}_{\text{Term 1}} \geq \frac{\epsilon}{2\|\Delta_i\|_2} \right) &\leq \sum_{d=1}^{D} \mathbb{P}_\epsilon \left( |\overline{c}(d) - \hat{c}_t(d)| \geq \frac{\epsilon}{2\sqrt{D}\|\Delta_i\|_2} \right) \\
&\leq 2D \exp\left( \frac{-\epsilon^2 t}{8D\|\Delta_i\|_2^2 \sigma_c^2 + \frac{4}{3}\delta\epsilon\sqrt{D}\|\Delta_i\|_2} \right),
\end{aligned}
\tag{50}
$$

where $\sigma_c^2$ is the variance upper-bound of preference distribution for each objective, $\delta$ is the upper-bound of $\|c_t\|_1$.

*Step-3-ii (Bounding Term 2):* For Term2, we compare the actual (corrupted) empirical means $\hat{c}_t$ with the ground-truth empirical means $\hat{c}_t^{\mathcal{S}}$. Since the corrupted empirical means can be altered by at most

absolute corruption $\frac{Z}{t}$ for each episode $t$, we show the event of Term 2 can only hold for a limited number of episodes.

Specifically, since the absolute corruption budget is at most $Z$, we have

$$\|\sum_{\tau=1}^{t}\widetilde{c}_{\tau} - \sum_{\tau=1}^{t}c_{\tau}\|_1 \leq \sum_{\tau=1}^{t}\|\widetilde{c}_{\tau} - c_{\tau}\|_1 \leq Z, \forall t \in (0,T],$$

and therefore

$$\underbrace{\|\hat{c}_t^{\mathcal{S}} - \hat{c}_t\|_2}_{\text{Term 2}} = \left\|\frac{1}{t}\sum_{\tau=1}^{t}c_{\tau} - \frac{1}{t}\sum_{\tau=1}^{t}\widetilde{c}_{\tau}\right\|_2 \leq \left\|\frac{1}{t}\sum_{\tau=1}^{t}c_{\tau} - \frac{1}{t}\sum_{\tau=1}^{t}\widetilde{c}_{\tau}\right\|_1 \leq \frac{Z}{t},$$

indicating that the corrupted empirical means $\hat{c}_t$ can be altered from the ones of ground-truth $\hat{c}_t^{\mathcal{S}}$ by at most absolute corruption $\frac{Z}{t}$ up to episode $t$. Hence, the event of Term 2 only holds for limited number of episode and will fail for sufficiently large $t$. Specifically, let $T_z = \left\lfloor \frac{2Z}{\epsilon\|\Delta_i\|_2} \right\rfloor$, then we have $\frac{\epsilon}{2\|\Delta_i\|_2} > \frac{Z}{t}, \forall t > T_z$. In this case, the event of Term 2 would hold at most up to $T_z$ episodes, i.e.,

$$\mathbb{P}_\epsilon\left( \underbrace{\|\hat{c}_t^{\mathcal{S}} - \hat{c}_t\|_2}_{\text{Term 2}} \geq \frac{\epsilon}{2\|\Delta_i\|_2} \right) = \begin{cases} 1, & \text{if } t \leq T_z; \\ 0, & \text{if } t > T_z. \end{cases} \tag{51}$$

*Step-3-iii (Union Bound on Term 1 and Term 2):* By union bound over $T$ episodes with Term 1 and Term 2 derives the upper-bound for the expectation of $N_{i,T}^{\tilde{c}}$ in adversarial corruptions case:

$$
\begin{aligned}
\mathbb{E}_\epsilon\left[N_{i,T}^{\tilde{c}}\right] &= \mathbb{E}_\epsilon\left[ \sum_{t=1}^{T} \mathbb{1}_{\{a_t=i\neq a^*, \hat{c}_t^T\mu_{a^*} \leq \hat{c}_t^T\mu_i + \eta_i - \epsilon\}} \right] \\
&= \sum_{t=1}^{T} \mathbb{P}_\epsilon\left( a_t = i \neq a^*, \hat{c}_t^T\mu_{a^*} \leq \hat{c}_t^T\mu_i + \eta_i - \epsilon \right) \\
&\underset{(a)}{\leq} \sum_{t=1}^{T} \mathbb{P}_\epsilon\left( \underbrace{\|\bar{c} - \hat{c}_t^{\mathcal{S}}\|_2}_{\text{Term 1}} + \underbrace{\|\hat{c}_t^{\mathcal{S}} - \hat{c}_t\|_2}_{\text{Term 2}} \geq \frac{\epsilon}{\|\Delta_i\|_2} \right) \\
&\leq \sum_{t=1}^{T} \mathbb{P}_\epsilon\left( \underbrace{\|\bar{c} - \hat{c}_t^{\mathcal{S}}\|_2}_{\text{Term 1}} \geq \frac{\epsilon}{2\|\Delta_i\|_2} \right) + \sum_{t=1}^{T} \mathbb{P}_\epsilon\left( \underbrace{\|\hat{c}_t^{\mathcal{S}} - \hat{c}_t\|_2}_{\text{Term 2}} \geq \frac{\epsilon}{2\|\Delta_i\|_2} \right) \quad (52) \\
&\underset{(b)}{\leq} 2D \sum_{t=1}^{T} \exp\left( \frac{-\epsilon^2 t}{8D\|\Delta_i\|_2^2\sigma_c^2 + \frac{4}{3}\delta\epsilon\sqrt{D}\|\Delta_i\|_2} \right) + T_z \\
&\underset{(c)}{\leq} \frac{2D}{\exp\left( \frac{\epsilon^2}{8D\|\Delta_i\|_2^2\sigma_c^2 + \frac{4}{3}\delta\epsilon\sqrt{D}\|\Delta_i\|_2} \right) - 1} + \left\lfloor \frac{2Z}{\epsilon\|\Delta_i\|_2} \right\rfloor \\
&\underset{(d)}{\leq} \frac{16D^2\|\Delta_i\|_2^2\sigma_c^2}{\epsilon^2} + \frac{8D^{\frac{3}{2}}\delta\|\Delta_i\|_2}{3\epsilon} + \left\lfloor \frac{2Z}{\epsilon\|\Delta_i\|_2} \right\rfloor.
\end{aligned}
$$

where (a) holds by Eq. 49, (b) holds by Eq. 50 and Eq. 51, (c) holds by the convergence of geometric series in Eq. 43, (d) holds by the fact that $e^x \geq x + 1, \forall x \geq 0$.

**Step-4 (Final R(T) Derivation):** Combine above result with Eq. 48, and choosing $\epsilon = \frac{\eta_i}{1+\sqrt{D}}$ yields

$$\mathbb{E}[N_{i,T}] \leq \underbrace{\frac{4(\delta + \frac{\delta}{\sqrt{D}})^2 \log(\frac{T}{\alpha})}{\eta_i^2} + \frac{D\pi^2\alpha^2}{3}}_{\substack{\textit{Suboptimal pulls caused by} \\ \textit{imprecise \textbf{reward estimation}}}}$$

$$+ \underbrace{\frac{16(D + D^{\frac{3}{2}})^2 \|\Delta_i\|_2^2 \sigma_c^2}{\eta_i^2} + \frac{8(D^2 + D^{\frac{3}{2}})\|\Delta_i\|_2\delta}{3\eta_i} + \left\lfloor \frac{2(1+\sqrt{D})Z}{\eta_i\|\Delta_i\|_2} \right\rfloor}_{\substack{\textit{Suboptimal pulls caused by} \\ \textit{imprecise \textbf{preference estimation}}}},$$

Multiplying the results above by the expected overall-reward gap $\eta_i$ for all suboptimal arms $i \neq a^*$ and summing them up conclude the proof of Theorem D.2.

$\square$

### D.4 REGRET OF PRUCB-APM: THEOREM 5 (NON-STATIONARY PREFERENCE)

The Theorem 5 establishes the upper bound of regret $R(T)$ for PRUCB-APM under abruptly preference changing environment. Note that in this case, the optimal arm is no longer fixed and can change with the abruptly shifting preference distributions, which introduces new challenges for the proof.

Let $a_t^*$ be the dynamic oracle at time step $t$, $\Delta_{i,t} = \boldsymbol{\mu}_{a_t^*} - \boldsymbol{\mu}_{i,t} \in \mathbb{R}^D, \forall t \in [1,T]$ be the gap of expected rewards between suboptimal arm $i$ and best arm $a_t^*$ at time step $t$. Define $\mathcal{T}_i = \{t \in [T]|a_t^* \neq i\}$ be the set of episodes when arm $i$ serving as a suboptimal arm over $T$. $\eta_i^\downarrow = \min_{t \in \mathcal{T}_i}\{\overline{\boldsymbol{c}}_t^T \Delta_{i,t}\}$ refers to the lower bound of the expected gap of overall-rewards between $i$ and $a_t^*$ over $T$. $\|\Delta_i^\uparrow\|_2 = \max_{\{t,j\}\in[T]\times[K]/i} \|\boldsymbol{\mu}_{i,t} - \boldsymbol{\mu}_{j,t}\|_2$ denotes the largest Euclidean distance between the expected rewards of arm $i$ and other arms over $T$.

#### D.4.1 PROOF SKETCH OF THEOREM 5

We follow the proof lines of Theorem 3. The main difficulty is that due to changes of preference distribution, the local empirical mean $\hat{\boldsymbol{c}}_t$ now would be a biased estimator of the expected preference $\overline{\boldsymbol{c}}_t$. It leads to the use of a tail bound on the deviation between $\hat{\boldsymbol{c}}_t$ and $\overline{\boldsymbol{c}}_t$ infeasible in bounding $\tilde{N}_{i,T}^{\tilde{c}}$. To address this problem, we employ proof techniques from (Garivier & Moulines, 2008) which consider sliding windows with and without breakpoints separately. For sliding windows without breakpoints, the estimation bias of $\hat{\boldsymbol{c}}_t$ vanishes entirely. In the case of sliding windows with breakpoints, the worst-case expected regret scales linearly with the product of the number of breakpoints and the length of the sliding window.

#### D.4.2 PROOF OF THEOREM 5

*Proof.* Let $\tilde{N}_{i,T} = \sum_{t=1}^T \mathbb{1}_{\{a_t = i \neq a_t^*\}}$. be the number of pulls of each arm $i$ when it serves as a suboptimal arm within horizon $T$. We first analyze $\tilde{N}_{i,T}$ and then extend to the final regret $R(T)$. The proof consists of several steps.

**Step-1 ($\tilde{N}_{i,T}$ Decomposition with Parameter $\epsilon_t$):**

Let $\epsilon_t = \min\{\epsilon_0, \delta\|\Delta_{i,t}\|_2\sqrt{\frac{D\log(t\wedge\tau)}{t\wedge\tau}}\}$, with $0 < \epsilon_0 \leq \eta_i^\downarrow$. Then we can decompose the the number of times the suboptimal arm $i$ is played as follows:

$$\tilde{N}_{i,T} = \sum_{t=1}^{T} \mathbb{1}_{\{a_t = i \neq a_t^*\}} = \underbrace{\sum_{t=1}^{T} \mathbb{1}_{\{a_t = i \neq a_t^*, \hat{\boldsymbol{c}}_t^T \mu_{a_t^*} > \hat{\boldsymbol{c}}_t^T \mu_i + \eta_i^\downarrow - \epsilon_t\}}}_{\tilde{N}_{i,T}^{\tilde{r}}: \text{ Suboptimal pulls caused by imprecise reward estimation}} + \underbrace{\sum_{t=1}^{T} \mathbb{1}_{\{a_t = i \neq a_t^*, \hat{\boldsymbol{c}}_t^T \mu_{a_t^*} \leq \hat{\boldsymbol{c}}_t^T \mu_i + \eta_i^\downarrow - \epsilon_t\}}}_{\tilde{N}_{i,T}^{\tilde{c}}: \text{ Suboptimal pulls caused by imprecise preference estimation}} .$$
(53)

**Step-2 (Bounding $\tilde{N}_{i,T}^{\tilde{r}}$):**

Define $\mathcal{M}_i$ as the set of episodes that arm $i$ achieves suboptimal expected overall-reward under the preference estimation $\hat{\boldsymbol{c}}_t$, i.e., $\mathcal{M}_i := \{t \in [T] \mid i \neq \arg\max_{j \in [K]} \hat{\boldsymbol{c}}_t^T \mu_j\}$. Let $L_i = \min_{t \in \mathcal{T}_i}\{\max_{j \in [K] \setminus i}\{\hat{\boldsymbol{c}}_t^T(\mu_j - \mu_i)\}\}$, $\hat{\mathcal{C}}_T^+ := \{[\hat{\boldsymbol{c}}_1(d), \hat{\boldsymbol{c}}_2(d), ..., \hat{\boldsymbol{c}}_T(d)] \neq \boldsymbol{0}, \forall d \in [D]\}$ is the collection set of preference estimation sequence.

For the event concerning $\tilde{N}_{i,T}^{\tilde{r}}$, we have $\hat{\boldsymbol{c}}_t^T \Delta_{i,t} > \eta_i^\downarrow - \epsilon_t \geq 0$ holding for all $t \in \mathcal{T}_i$. This implies that, for any episode $t_i \in \mathcal{T}_i$, $a_t^*$ would still yield a better result than $i$ given the current preference estimation $\hat{\boldsymbol{c}}_{t_i}$, indicating $t_i \in \mathcal{M}_i$ as well. Therefore, we can conclude that $\mathcal{T}_i \subset \mathcal{M}_i$. Moreover, recall that PRUCB-APM also leverages $\hat{\boldsymbol{c}}_t$ for optimistic arm selection, i.e., $a_t = \arg\max f(\hat{\boldsymbol{c}}_t, \hat{\boldsymbol{r}}_{i,t} + \sqrt{\frac{\log(t/\alpha)}{\max\{1, N_{i,t}\}}} \boldsymbol{e})$. By Proposition 8, we have

$$\mathbb{E}_{\epsilon_t}\left[\tilde{N}_{i,T}^{\tilde{r}}\right] = \mathbb{E}_{\epsilon_t}\left[\sum_{t \in \mathcal{T}_i} \mathbb{1}_{\{a_t = i, \hat{\boldsymbol{c}}_t^T \Delta_{i,t} > \eta_i^\downarrow - \epsilon_t\}}\right] \leq \mathbb{E}\left[\sum_{t \in \mathcal{T}_i} \mathbb{1}_{\{a_t = i\}}\right] \leq \frac{4\delta^2 \log\left(\frac{T}{\alpha}\right)}{L_i^2} + \frac{|\hat{\mathcal{C}}_T^+|\pi^2 \alpha^2}{3}.$$
(54)

Additionally, since $\hat{\boldsymbol{c}}_t^T \Delta_{i,t} > \eta_i^\downarrow - \epsilon_t \geq \eta_i^\downarrow - \epsilon_0 > 0$ holds for all $t \in \mathcal{T}_i$, it implies that

$$L_i = \min_{t \in \mathcal{T}_i}\{\max_{j \in [K] \setminus i}\{\hat{\boldsymbol{c}}_t^T(\mu_j - \mu_i)\}\} \geq \min_{t \in \mathcal{T}_i} \hat{\boldsymbol{c}}_t^T \Delta_{i,t} \geq \eta_i^\downarrow - \epsilon_t \geq \eta_i^\downarrow - \epsilon_0.$$

Plugging above result into Eq. 54, and by $|\hat{\mathcal{C}}_T^+| \leq D$, we have the expectation of $N_{i,T}^{\tilde{r}}$ in Eq. 24 can be upper-bounded as follows:

$$\mathbb{E}_{\epsilon_t}\left[\tilde{N}_{i,T}^{\tilde{r}}\right] = \mathbb{E}_{\epsilon_t}\left[\sum_{t \in \mathcal{T}_i} \mathbb{1}_{\{a_t = i, \hat{\boldsymbol{c}}_t^T \Delta_i > \eta_i^\downarrow - \epsilon_t\}}\right] \leq \frac{4\delta^2 \log(T/\alpha)}{(\eta_i^\downarrow - \epsilon_0)^2} + D\frac{\pi^2 \alpha^2}{3}.$$
(55)

**Step-3 (Bounding $\tilde{N}_{i,T}^{\tilde{c}}$):**

Next we analyze the upper bound of $\tilde{N}_{i,T}^{\tilde{c}}$. By the sliding window estimation fashion, $\tilde{N}_{i,T}^{\tilde{c}}$ can be decomposed and upper bounded as follows:

$$\sum_{t=1}^{T} \mathbb{1}_{\{\hat{\boldsymbol{c}}_t^T \mu_{a_t^*} \leq \hat{\boldsymbol{c}}_t^T \mu_i + \eta_i^\downarrow - \epsilon_t\}} \leq \psi_T \tau + \sum_{t \in \mathcal{W}_\tau} \mathbb{1}_{\{\hat{\boldsymbol{c}}_t^T \mu_{a_t^*} \leq \hat{\boldsymbol{c}}_t^T \mu_i + \eta_i^\downarrow - \epsilon_t\}},$$
(56)

and $\mathcal{W}_\tau$ is the set of all time instances where the distributions of $\boldsymbol{c}_t$ within the sliding window remain the same, i.e., $\mathcal{W}_\tau := \{t \mid \overline{\boldsymbol{c}}_s = \overline{\boldsymbol{c}}_t, \forall s \in (t - \tau, t]\}$. Since $\overline{\boldsymbol{c}}_t^T \Delta_{i,t} \geq \eta_i^\downarrow > \eta_i^\downarrow - \epsilon_t$ always holds, for any $t \in \mathcal{W}_\tau$, by applying Lemma 10, we have

$$\left\{\hat{\boldsymbol{c}}_t^T \mu_{a_t^*} \leq \hat{\boldsymbol{c}}_t^T \mu_i + \eta_i^\downarrow - \epsilon_t\right\} = \left\{\hat{\boldsymbol{c}}_t^T \Delta_{i,t} \leq \eta_i^\downarrow - \epsilon_t\right\}$$

$$\underset{(a)}{\subset} \left\{\|\overline{\boldsymbol{c}}_t - \hat{\boldsymbol{c}}_t\|_2 \geq \frac{\overline{\boldsymbol{c}}_t^T \Delta_{i,t} - (\eta_i^\downarrow - \epsilon_t)}{\|\Delta_{i,t}\|_2}\right\}$$

$$= \left\{\|\overline{\boldsymbol{c}}_t - \hat{\boldsymbol{c}}_t\|_2 \geq \frac{\epsilon_t}{\|\Delta_{i,t}\|_2}\right\},$$
(57)

where (a) holds by Lemma 10. Since the sliding-window length $\tau$ is a tuning parameter of PRUCB-APM, which can be sufficiently large, we thus assume $\tau > \lfloor (\frac{\sqrt{D}\delta\|\Delta_i^{\uparrow}\|_2}{\epsilon_0})^{\frac{5}{2}} \rfloor = t_{\epsilon_0}$, with $\|\Delta_i^{\uparrow}\|_2 = \max_{j \in [K]/i} \|\boldsymbol{\mu}_i - \boldsymbol{\mu}_j\|_2$. Then for any $t \in \mathcal{W}_\tau > t_{\epsilon_0}$, we have

$$
\begin{aligned}
\mathbb{P}_{\epsilon_t}\left(\hat{\boldsymbol{c}}_t^T \mu_{a_t^*} \leq \hat{\boldsymbol{c}}_t^T \mu_i + \eta_i^{\downarrow} - \epsilon_t\right) &\leq \mathbb{P}_{\epsilon_t}\left(\|\overline{\boldsymbol{c}}_t - \hat{\boldsymbol{c}}_t\|_2 \geq \frac{\epsilon_t}{\|\Delta_{i,t}\|_2}\right) \\
&\underset{(a)}{=} \mathbb{P}_{\epsilon_t}\left(\|\overline{\boldsymbol{c}}_t - \hat{\boldsymbol{c}}_t\|_2 \geq \delta\sqrt{\frac{D \log(t \wedge \tau)}{t \wedge \tau}}\right) \\
&= \mathbb{P}_{\epsilon_t}\left(\sqrt{\sum_{d \in [D]} \left(\overline{\boldsymbol{c}}_t(d) - \hat{\boldsymbol{c}}_t(d)\right)^2} \geq \delta\sqrt{\frac{D \log(t \wedge \tau)}{t \wedge \tau}}\right) \\
&\underset{(b)}{\leq} \sum_{d \in [D]} \mathbb{P}\left(|\overline{\boldsymbol{c}}_t(d) - \hat{\boldsymbol{c}}_t(d)| \geq \delta\sqrt{\frac{\log(t \wedge \tau)}{t \wedge \tau}}\right),
\end{aligned}
\tag{58}
$$

where (a) holds by the definition of $\epsilon_t$ and $\epsilon_0 > \delta\sqrt{\frac{D \log(t \wedge \tau)}{t \wedge \tau}}, \forall t > t_{\epsilon_0}$; (b) holds since union bound and the fact that there must be at least one objective $d \in [D]$ satisfying $(\overline{\boldsymbol{c}}(d) - \hat{\boldsymbol{c}}_t(d))^2 \geq \frac{1}{D}(\delta\sqrt{\frac{D \log(t \wedge \tau)}{t \wedge \tau}})^2$, otherwise the event would fail. Note that for any $t \in \mathcal{W}_\tau$, the distribution of $\boldsymbol{c}_t$ remains the same with those of previous instances within its $(\tau \wedge t)$-length sliding window. We can thus employ a tail bound for measuring the deviation on the empirical mean (i.e., $\hat{\boldsymbol{c}}_t$) of i.i.d. sequence $\boldsymbol{c}_{t-\tau}, ..., \boldsymbol{c}_{t-1}$. Using the Hoeffding's inequality (Lemma 9), the probability for any objective $d \in [D]$, any $t \in \mathcal{W}_\tau > t_{\epsilon_0}$ can be upper-bounded as follows:

$$
\begin{aligned}
\mathbb{P}\left(|\overline{\boldsymbol{c}}_t(d) - \hat{\boldsymbol{c}}_t(d)| \geq \delta\sqrt{\frac{\log(t \wedge \tau)}{t \wedge \tau}}\right) &\leq 2\exp\left(-\frac{2\delta^2(\tau \wedge t)^2 \log(\tau \wedge t)}{(\tau \wedge t)\sum_{i=1}^{\tau \wedge t}\delta^2}\right) \\
&= 2\exp\left(-2\log(\tau \wedge t)\right) \\
&= \frac{2}{(\tau \wedge t)^2}.
\end{aligned}
\tag{59}
$$

Plugging back to Eq. 58 yields

$$
\mathbb{P}\left(\hat{\boldsymbol{c}}_t^T \mu_{a_t^*} \leq \hat{\boldsymbol{c}}_t^T \mu_i + \eta_i^{\downarrow} - \epsilon\right) \leq \frac{2D}{(\tau \wedge t)^2}.
\tag{60}
$$

By combining Eq. 56 with Eq. 60, we can derive the upper-bound for $\tilde{N}_{i,T}^{\tilde{c}}$ as follows:

$$
\begin{aligned}
\mathbb{E}\left[\sum_{t=1}^{T} \mathbb{1}_{\{\hat{\boldsymbol{c}}_t^T \mu_{a_t^*} \leq \hat{\boldsymbol{c}}_t^T \mu_i + \eta_i^{\downarrow} - \epsilon_t\}}\right] \\
\leq \psi_T \tau + \sum_{t=1}^{t_{\epsilon_0}} 1 + 2D \sum_{t=t_{\epsilon_0}+1}^{T} \frac{1}{(\tau \wedge t)^2} \\
= \psi_T \tau + t_{\epsilon_0} + 2D \sum_{t=t_{\epsilon_0}+1}^{\tau} \frac{1}{t^2} + 2D \sum_{t=\tau+1}^{T} \frac{1}{\tau^2} \\
\leq \psi_T \tau + \left(\frac{\sqrt{D}\delta\|\Delta_i^{\uparrow}\|_2}{\epsilon_0}\right)^{\frac{5}{2}} + \frac{D\pi^2}{3} + \frac{2D(T-\tau)}{\tau^2}.
\end{aligned}
\tag{61}
$$

**Step-4 (Final $R(T)$ Derivation and Optimization over $\epsilon_0$ and $\tau$):**

Combining Eq.53 with the corresponding upper-bounds of expected $\tilde{N}_{i,T}^{\tilde{r}}$ (Eq.55) and $\tilde{N}_{i,T}^{\tilde{c}}$ (Eq.61) we can get

$$\mathbb{E}[\tilde{N}_{i,T}] \leq \frac{4\delta^2 \log(T/\alpha)}{(\eta_i^\downarrow - \epsilon_0)^2} + D\frac{\pi^2\alpha^2}{3} + \psi_T\tau + \left(\frac{\sqrt{D}\delta\|\Delta_i^\uparrow\|_2}{\epsilon_0}\right)^{\frac{5}{2}} + \frac{D\pi^2}{3} + \frac{2D(T-\tau)}{\tau^2}. \quad (62)$$

Similar with Theorem 3, the parameter $\epsilon \in (0, \eta_i)$ can be optimally selected so as to minimize the RHS of Eq. 62. Following the setup in the proof of Theorem 3, we choose $\epsilon_0 = \frac{\eta_i^\downarrow}{1+\sqrt{D}}$ and have

$$\mathbb{E}[\tilde{N}_{i,T}] \leq \underbrace{\frac{4(\delta + \frac{\delta}{\sqrt{D}})^2 \log(T/\alpha)}{(\eta_i^\downarrow)^2} + D\frac{\pi^2\alpha^2}{3}}_{\substack{\textit{Suboptimal pulls caused by imprecise} \\ \textbf{\textit{reward estimation}}}}$$

$$\underbrace{+ \psi_T\tau + \frac{2D(T-\tau)}{\tau^2} + \left(\frac{2D\delta\|\Delta_i^\uparrow\|_2}{\eta_i^\downarrow}\right)^{\frac{5}{2}} + \frac{D\pi^2}{3}}_{\substack{\textit{Suboptimal pulls caused by} \\ \textit{imprecise } \textbf{\textit{preference estimation}}}}.$$

Multiplying the results above by the upper-bound of expected overall-reward gap $\eta_i^\uparrow = \max_{t \in \mathcal{T}_i}\{\overline{c}_t^T \Delta_{i,t}\}$ for all arms $i \in [K]$ and summing them up yields the desired result of Theorem 5. $\qquad\square$

**Corollary 13.1.** *If the horizon $T$ and the number of breakpoints $\psi_T$ are known in advance, the window size $\tau$ can be chosen so as to minimize the $\mathbb{E}[\tilde{N}_{i,T}]$. For simplicity and consistency cross $K$ arms, we select $\tau$ by optimizing the term $\psi_T\tau + \frac{2DT}{\tau^2}$. Specifically, taking $\tau = \left(\frac{4DT}{\psi_T}\right)^{\frac{1}{3}}$ yields*

$$\mathbb{E}[\tilde{N}_{i,T}] \leq \frac{4(\delta + \frac{\delta}{\sqrt{D}})^2 \log(\frac{T}{\alpha})}{(\eta_i^\downarrow)^2} + D\frac{\pi^2\alpha^2}{3} + (4^{\frac{1}{3}} + 2^{-\frac{1}{3}})D^{\frac{1}{3}}\psi_T^{\frac{2}{3}}T^{\frac{1}{3}} + \left(\frac{D\delta\|\Delta_i^\uparrow\|_2}{\eta_i^\downarrow}\right)^{\frac{5}{2}} + \frac{D\pi^2}{3}$$

$$= \mathcal{O}(\log(T) + \psi_T^{\frac{2}{3}}T^{\frac{1}{3}}).$$

*Assuming that $\psi_T = \mathcal{O}(T^\gamma)$ for some $\gamma \in [0, 1)$, then we have the expected number of sub-optimal pulls of arm $i$ is upper-bounded as $\mathcal{O}(T^{(1+2\gamma)/3})$. In particular, if $\gamma = 0$, the number of breakpoints $\psi_T$ is upper-bounded by $\psi$ independently of $T$, then upper-bound is $\mathcal{O}(\log(T) + \psi^{\frac{2}{3}}T^{\frac{1}{3}})$.*

# E  ANALYSES FOR SECTION 7 (HIDDEN PREFERENCE)

Our main result of Theorem 6 in Section 7 indicates that the proposed PUCB-HPM under hidden preference environment achieves sublinear expected regret $R(T) \leq \tilde{\mathcal{O}}(D\sqrt{T})$. To prove this, we need two key components. The first is to show that the value of $\hat{r}_{i,t}$, the matrix of $\Upsilon_t$, and the region of $\Theta_t$ are good estimators of $\mu_i$, $\mathbb{E}[\Upsilon_t]$ and $\overline{c}$ respectively. The second is to show that as long as the aforementioned high-probability event holds, we have some control on the growth of the regret. We show the analyses regarding these two components in the following sections.

## E.1  UNIFORM CONFIDENCE BOUND FOR ESTIMATIONS

**Proposition 14.** *For any $\lambda > 0$, if set $\beta_t = \left(\sqrt{\lambda} + \sqrt{D\log\left(1 + \frac{t-1}{\lambda}\right) + 4\log\left(\frac{\pi t}{\sqrt{2}\vartheta}\right)}\right)^2$ and $\alpha = \sqrt{\frac{8\vartheta}{KD(D+3)\pi^2}}$, for all $t \in (1, T]$, with probability at least $1 - \vartheta$, we have following events hold*

*simultaneously:*

*Event A:* $\{\overline{c} \in \Theta_t\}$,

*Event B:* $\left\{ |\boldsymbol{\mu}_i(d) - \hat{\boldsymbol{r}}_{i,t}(d)| \le \sqrt{\dfrac{\log\left(\frac{t}{\alpha}\right)}{N_{i,t}}}, \forall i \in [K], \forall d \in [D] \right\}$,

*Event C:* $\left\{ \mathbb{E}\left[ \displaystyle\sum_{\iota \in \mathcal{T}_{i,t-1}} \boldsymbol{r}_{i,\iota}\boldsymbol{r}_{i,\iota}^T \right](m,n) - \displaystyle\sum_{\iota \in \mathcal{T}_{i,t-1}} \left(\boldsymbol{r}_{i,\iota}\boldsymbol{r}_{i,\iota}^T\right)(m,n) \le \sqrt{N_{i,t}\log\left(\dfrac{t}{\alpha}\right)}, \right.$

$\left. \forall i \in [K], \forall m \in [D], \forall n \in [m,D] \right\}$,

*where $\mathcal{T}_{i,t}$ is the set of episodes that arm $i$ is pulled within $t$ steps.*

Proposition 14 shows that by proper parameter settings of PUCB-HPM, the Events A, B, C hold simultaneously with high probability. To proof 14, we study the uniform confidence bound of Proposition 14 by considering the Events $A$, $B$ and $C$ separately.

*Proof of Proposition 14.* **Step-1 (Confidence analysis of Event A):**

First we state two lemmas from (Abbasi-Yadkori et al., 2011) that will be utilized in our confidence analysis of Event A:

**Lemma 15** (Self-Normalized Bound for Vector-Valued Martingales (Abbasi-Yadkori et al., 2011), Theorem 1). *Let $\{\mathcal{F}_t\}_{t=0}^{\infty}$ be a filtration, and let $\{\zeta_t\}_{t=1}^{\infty}$ be a real-valued stochastic process such that $\zeta_t$ is $\mathcal{F}_t$-measurable, $\mathbb{E}[\zeta_t \mid \mathcal{F}_{t-1}] = 0$ and $\zeta_t$ is conditionally $R$-sub-Gaussian for some $R \ge 0$. Let $\{\boldsymbol{X}_t\}_{t=1}^{\infty}$ be an $\mathbb{R}^d$-valued stochastic process such that $\boldsymbol{X}_t$ is $\mathcal{F}_{t-1}$-measurable. Assume that $\boldsymbol{V} \in \mathbb{R}^{d \times d}$ is a positive definite matrix, and define $\overline{\boldsymbol{V}}_t = \boldsymbol{V} + \sum_{\iota=1}^{t} \boldsymbol{X}_\iota \boldsymbol{X}_\iota^T$. Then for any $\vartheta \ge 0$, with probability at least $1 - \vartheta$, for all $t \ge 1$, we have*

$$\|\sum_{\iota=1}^{t} \zeta_\iota \boldsymbol{X}_\iota\|_{\overline{\boldsymbol{V}}_t^{-1}}^2 \le 2R^2 \log\left( \frac{\det\left(\overline{\boldsymbol{V}}_t\right)^{\frac{1}{2}} \det\left(\boldsymbol{V}\right)^{-\frac{1}{2}}}{\vartheta} \right).$$

**Lemma 16** (Determinant-Trace Inequality (Abbasi-Yadkori et al., 2011), Lemma 10). *Suppose $\boldsymbol{X}_1, ..., \boldsymbol{X}_t \in \mathbb{R}^d$ and $\|\boldsymbol{X}_\iota\|_2 \le L, \forall \iota \in [1,t]$. Let $\overline{\boldsymbol{V}}_t = \lambda \boldsymbol{I} + \sum_{\iota=1}^{t} \boldsymbol{X}_\iota \boldsymbol{X}_\iota^T$ for some $\lambda > 0$, then*

$$\det\left(\overline{\boldsymbol{V}}_t\right) \le \left(\lambda + \frac{tL^2}{d}\right)^d.$$

Define $\zeta_t = g_{a_t,t} - \overline{c}^T \boldsymbol{r}_{a_t,t} = \boldsymbol{c}_t^T \boldsymbol{r}_{a_t,t} - \overline{c}^T \boldsymbol{r}_{a_t,t}$. By the definition of $\hat{c}_t$ and $\zeta_t$, for $t \ge 2$ we have

$$\begin{aligned}
\hat{c}_t - \overline{c} &= \Upsilon_t^{-1} \sum_{\iota=1}^{t-1} g_{a_\iota,\iota} \boldsymbol{r}_{a_\iota,\iota} - \overline{c} \\
&= \Upsilon_t^{-1} \sum_{\iota=1}^{t-1} \boldsymbol{r}_{a_\iota,\iota}(\overline{c}^T \boldsymbol{r}_{a_\iota,\iota} + \zeta_\iota) - \overline{c} \\
&= \Upsilon_t^{-1} \left( \sum_{\iota=1}^{t-1} \boldsymbol{r}_{a_\iota,\iota}\boldsymbol{r}_{a_\iota,\iota}^T \right)\overline{c} + \Upsilon_t^{-1} \sum_{\iota=1}^{t-1} \zeta_\iota \boldsymbol{r}_{a_\iota,\iota} - \overline{c} \qquad (63) \\
&= \Upsilon_t^{-1} \left( \Upsilon_t - \lambda \boldsymbol{I} \right)\overline{c} - \overline{c} + \Upsilon_t^{-1} \sum_{\iota=1}^{t-1} \zeta_\iota \boldsymbol{r}_{a_\iota,\iota} \\
&= -\lambda \Upsilon_t^{-1}\overline{c} + \Upsilon_t^{-1} \sum_{\iota=1}^{t-1} \zeta_\iota \boldsymbol{r}_{a_\iota,\iota}.
\end{aligned}$$

Following the above results, we can bound $\|\hat{c}_t - \overline{c}\|_{\Upsilon_t}$ as:

$$
\begin{aligned}
\sqrt{(\hat{c}_t - \overline{c})^T \Upsilon_t (\hat{c}_t - \overline{c})} &= \left\| \Upsilon_t^{\frac{1}{2}} (\hat{c}_t - \overline{c}) \right\|_2 \\
&\underset{(a)}{=} \left\| \Upsilon_t^{\frac{1}{2}} \left( -\lambda \Upsilon_t^{-1} \overline{c} + \Upsilon_t^{-1} \sum_{\iota=1}^{t-1} \zeta_\iota r_{a_\iota,\iota} \right) \right\|_2 \\
&\underset{(b)}{\leq} \left\| \lambda \Upsilon_t^{-\frac{1}{2}} \overline{c} \right\|_2 + \left\| \Upsilon_t^{-\frac{1}{2}} \sum_{\iota=1}^{t-1} \zeta_\iota r_{a_\iota,\iota} \right\|_2 \\
&\underset{(c)}{\leq} \sqrt{\lambda} \|\overline{c}\|_2 + \left\| \Upsilon_t^{-\frac{1}{2}} \sum_{\iota=1}^{t-1} \zeta_\iota r_{a_\iota,\iota} \right\|_2,
\end{aligned}
\tag{64}
$$

where (a) follows from Eq. 63, (b) follows from Triangle Inequality, and (c) holds since $\left\| \Upsilon_t^{-\frac{1}{2}} \right\|_2 \leq \left\| \Upsilon_1^{-\frac{1}{2}} \right\|_2 = \frac{1}{\sqrt{\lambda}}$. The first term above can be immediately bounded by $\sqrt{\lambda}$. We next analyze the second term.

Let $a_{1:t} = \{a_\iota\}_{\iota=1}^t$ be the sequence of historical pulled actions within $t$ steps, $g_{1:t} = \{g_{a_\iota,\iota}\}_{\iota=1}^t$ and $r_{1:t} = \{r_{a_\iota,\iota}\}_{\iota=1}^t$ be the sequences of historical overall scores and reward vectors within $t$ steps respectively, and define the $\sigma$ algebra $\mathcal{F}_{t-1} = \sigma(a_{1:t}, g_{1:t-1}, r_{1:t})$. By definition of $\zeta_t$, note that for any $t \geq 1$,

$$
\begin{aligned}
\mathbb{E}[\zeta_t \mid \mathcal{F}_{t-1}] &= \mathbb{E}[c_t^T r_{a_t,t} \mid \mathcal{F}_{t-1}] - \mathbb{E}[\overline{c}^T r_{a_t,t} \mid \mathcal{F}_{t-1}] \\
&\underset{(a)}{=} \overline{c}^T r_{a_t,t} - \overline{c}^T r_{a_t,t} = 0,
\end{aligned}
$$

where (a) holds since $c_t$ is independent of $\mathcal{F}_{t-1}$ and the conditional expectation fact that $\mathbb{E}(X \mid \mathcal{F}) = X$ if $X \in \mathcal{F}$. Furthermore, by assumption of 1-bounded overall-reward, $-1 \leq \zeta_t \leq 1$ holds almost surely, and hence we can conclude that $\zeta_t$ is conditionally 1-sub-Gaussian. Also, since $r_{a_t,t}$ is $\mathcal{F}_{t-1}$-measurable, by applying Lemma 15, we have with probability at least $1 - \vartheta_t$,

$$
\left\| \Upsilon_t^{-\frac{1}{2}} \sum_{\iota=1}^{t-1} \zeta_\iota r_{a_\iota,\iota} \right\|_2^2 = \left\| \sum_{\iota=1}^{t-1} \zeta_\iota r_{a_\iota,\iota} \right\|_{\Upsilon_t^{-1}}^2 \leq \log \left( \frac{\det(\Upsilon_t) \det(\lambda I)^{-1}}{\vartheta_t^2} \right).
\tag{65}
$$

By Lemma 16, we have

$$
\frac{\det(\Upsilon_t)}{\det(\lambda I)} \leq \frac{\left( \lambda + \frac{(t-1)D}{D} \right)^D}{\lambda^D} = \left( 1 + \frac{(t-1)}{\lambda} \right)^D.
\tag{66}
$$

Combining Eq. 66, Eq. 65 and Eq. 64 yields:

$$
\sqrt{(\hat{c}_t - \overline{c})^T \Upsilon_t (\hat{c}_t - \overline{c})} \leq \sqrt{\lambda} + \sqrt{D \log \left( 1 + \frac{(t-1)}{\lambda} \right) - 2 \log(\vartheta_t)}.
\tag{67}
$$

For $t \geq 1$, define $\vartheta_t = \frac{2\vartheta}{(\pi t)^2}$ be the instantaneous failure probability and plug back into Eq. 67, we have

$$
\sqrt{(\hat{c}_t - \overline{c})^T \Upsilon_t (\hat{c}_t - \overline{c})} \leq \sqrt{\lambda} + \sqrt{D \log \left( 1 + \frac{(t-1)}{\lambda} \right) + 4 \log \left( \frac{\pi t}{\sqrt{2\vartheta}} \right)} = \sqrt{\beta_t},
\tag{68}
$$

indicating $\overline{c} \in \Theta_t$ holds with probability at least $1 - \frac{2\vartheta}{(\pi t)^2}$ at each time step $t$. Hence, by the union bound, we can derive an upper-bound over the failure probability of Event A as

$$\mathbb{P}(A^{\mathrm{c}}) = \mathbb{P}(\exists t, \overline{c} \notin \Theta_t) \le \sum_{t=1}^{\infty} \mathbb{P}(\overline{c} \notin \Theta_t) \le \frac{2\vartheta}{\pi^2} \sum_{t=1}^{\infty} \frac{1}{t^2} \underset{(a)}{=} \frac{2\vartheta}{\pi^2} \frac{\pi^2}{6} = \frac{\vartheta}{3}. \tag{69}$$

where (a) holds by the convergence of sum of reciprocals of squares that

$$\sum_{t=1}^{\infty} t^{-2} = \frac{\pi^2}{6}. \tag{70}$$

Thus we conclude by choosing $\beta_t = \left( \sqrt{\lambda} + \sqrt{D \log\left(1 + \frac{t-1}{\lambda}\right) + 4\log\left(\frac{\pi t}{\sqrt{2\vartheta}}\right)} \right)^2$, Event A holds with probability at least $1 - \frac{\vartheta}{3}$.

**Step-2 (Confidence analysis of Event B):**

For any $i \in [K], d \in [D], t \in (0, T]$, by Hoeffding's Inequality (Lemma 9), we have the instantaneous failure probability of Event B can be bounded as:

$$\mathbb{P}\left( |\hat{r}_{i,t}(d) - \mu_i(d)| > \sqrt{\frac{\log(t/\alpha)}{N_{i,t}}} \right) \le 2\exp\left( \frac{-2N_{i,t}^2 \log(t/\alpha)}{N_{i,t} \sum_{\iota=1}^{N_{i,t}} (1 - 0)^2} \right)$$
$$= 2\exp\left( -2\log(t/\alpha) \right) \tag{71}$$
$$= 2\left( \frac{\alpha}{t} \right)^2,$$

which yields the upper bound of $\mathbb{P}(B^{\mathrm{c}})$ by union bound as

$$\mathbb{P}(B^{\mathrm{c}}) = \mathbb{P}\left( \exists\{i, d, t\}, |\hat{r}_{i,t}(d) - \mu_i(d)| > \sqrt{\frac{2\log(t/\alpha)}{N_{i,t}}} \right)$$
$$\le 2 \sum_{t=1}^{T} \sum_{i=1}^{K} \sum_{d=1}^{D} \mathbb{P}\left( |\hat{r}_{i,t}(d) - \mu_i(d)| > \sqrt{\frac{\log(t/\alpha)}{N_{i,t}}} \right) \tag{72}$$
$$\le 2 \sum_{t=1}^{T} \sum_{i=1}^{K} \sum_{d=1}^{D} \left( \frac{\alpha}{t} \right)^2 \underset{(Eq.\ 70)}{\le} \frac{KD\alpha^2\pi^2}{3},$$

**Step-3 (Confidence analysis of Event C):**

The proof follows similar lines as above. Note that for any $i \in [K], t \in (1, T], m \in [1, D], n \in [m, D]$, we have the instantaneous failure probability of Event C can be bounded as

$$\mathbb{P}\left( \mathbb{E}\left[ \sum_{\iota \in \mathcal{T}_{i,t-1}} \boldsymbol{r}_{i,\iota}\boldsymbol{r}_{i,\iota}^T \right](m, n) - \sum_{\iota \in \mathcal{T}_{i,t-1}} \left( \boldsymbol{r}_{i,\iota}\boldsymbol{r}_{i,\iota}^T \right)(m, n) > \sqrt{N_{i,t} \log\left( \frac{t}{\alpha} \right)} \right)$$
$$= \mathbb{P}\left( \mathbb{E}\left[ \boldsymbol{r}_i\boldsymbol{r}_i^T \right](m, n) - \frac{1}{N_{i,t}} \sum_{\iota \in \mathcal{T}_{i,t-1}} \left( \boldsymbol{r}_{i,\iota}\boldsymbol{r}_{i,\iota}^T \right)(m, n) > \sqrt{\frac{\log(t/\alpha)}{N_{i,t}}} \right)$$
$$\le \exp\left( -\frac{2N_{i,t}^2 \log(t/\alpha)}{N_{i,t}^2 (1-0)^2} \right) = \left( \frac{\alpha}{t} \right)^2. \quad \text{(by Lemma 9 and } (\boldsymbol{r}_{i,\iota}\boldsymbol{r}_{i,\iota}^T)(m, n) \in [0, 1] \text{)}$$

Using union bound, we have $\mathbb{P}(C^{\mathsf{c}})$ as

$$
\begin{aligned}
\mathbb{P}(C^{\mathsf{c}}) &= \mathbb{P}\left(\exists\{i,t,m,n\}, \mathbb{E}\left[\sum_{\iota \in \mathcal{T}_{i,t-1}} \boldsymbol{r}_{i,\iota}\boldsymbol{r}_{i,\iota}^T\right](m,n) - \sum_{\iota \in \mathcal{T}_{i,t-1}}\left(\boldsymbol{r}_{i,\iota}\boldsymbol{r}_{i,\iota}^T\right)(m,n) > \sqrt{N_{i,t}\log\left(\frac{t}{\alpha}\right)}\right) \\
&\leq \sum_{t=1}^{T}\sum_{i=1}^{K}\sum_{m=1}^{D}\sum_{n=m}^{D}\mathbb{P}\left(\mathbb{E}\left[\sum_{\iota \in \mathcal{T}_{i,t-1}} \boldsymbol{r}_{i,\iota}\boldsymbol{r}_{i,\iota}^T\right](m,n) - \sum_{\iota \in \mathcal{T}_{i,t-1}}\left(\boldsymbol{r}_{i,\iota}\boldsymbol{r}_{i,\iota}^T\right)(m,n) > \sqrt{N_{i,t}\log\left(\frac{t}{\alpha}\right)}\right) \\
&\leq \sum_{t=1}^{T}\sum_{i=1}^{K}\sum_{m=1}^{D}\sum_{n=m}^{D}\left(\frac{\alpha}{t}\right)^2 \underset{(Eq.\ 70)}{\leq} \frac{KD(D-1)\alpha^2\pi^2}{12}.
\end{aligned}
\tag{73}
$$

**Step-4 (Union confidence on three Events):**

Combining Eq. 69, Eq. 72 and Eq. 73, and setting $\alpha = \sqrt{\frac{8\vartheta}{KD(D+3)\pi^2}}$, by union bound, we can have the overall failure probability bound of three Events as

$$
\begin{aligned}
\mathbb{P}(A^{\mathsf{c}} \cup B^{\mathsf{c}} \cup C^{\mathsf{c}}) &\leq \mathbb{P}(A^{\mathsf{c}}) + \mathbb{P}(B^{\mathsf{c}}) + \mathbb{P}(C^{\mathsf{c}}) \\
&= \frac{\vartheta}{3} + \left(\frac{KD(D-1)\pi^2}{12} + \frac{4KD\pi^2}{12}\right)\left(\frac{8\vartheta}{KD(D+3)\pi^2}\right) \\
&= \frac{\vartheta}{3} + \frac{2\vartheta}{3} = \vartheta.
\end{aligned}
$$

This concludes the proof of Proposition 14. $\qquad\square$

### E.2 PROOF OF THEOREM 6

*Proof.* Based on the assumptions in Proposition 14, we next show that when Events of A, B, C in Proposition 14 hold (detailed definitions of Events of A, B, C refer to Appendix E.1), the sub-linear regret of PUCB-HPM can be achieved. Please see the detailed proof steps below.

#### E.2.1 STEP-1 (REGRET ANALYSIS AND DECOMPOSITION)

Let $M$ be an arbitrary positive integer, we can express $R(T)$ in a truncated form with respect to $M$ as follows:

$$
R(T) = \sum_{t=1}^{T}\text{regret}_t \leq M + \sum_{t=M+1}^{T}\text{regret}_t,
\tag{74}
$$

where $\text{regret}_t$ denotes the instantaneous regret of PRUCB-HPM at step $t \in [T]$, and the last inequality holds since the fact that the instantaneous regret is upper-bounded by 1 (by Assumption 7.1).

Next, we analyze the instantaneous regret over the truncated time horizon $[M+1, T]$. Let $\tilde{\boldsymbol{c}}_t, a_t$ be the solution of policy such that

$$
\tilde{\boldsymbol{c}}_t^T\left(\hat{\boldsymbol{r}}_{a_t,t} + \sqrt{\frac{\log\left(\frac{t}{\alpha}\right)}{N_{a_t,t}}}\boldsymbol{e}\right) = \max_{\boldsymbol{c}' \in \Theta_t}\max_{i \in [K]}\boldsymbol{c}'^T\left(\hat{\boldsymbol{r}}_{i,t} + \sqrt{\frac{\log\left(\frac{t}{\alpha}\right)}{N_{i,t}}}\boldsymbol{e}\right).
\tag{75}
$$

Please note that since events A and B hold, we have

$$
\overline{\boldsymbol{c}} \in \Theta_t,
\tag{76}
$$

$$
\boldsymbol{\mu}_{a^*}(d) \leq \hat{\boldsymbol{r}}_{a^*,t}(d) + \sqrt{\frac{\log\left(\frac{t}{\alpha}\right)}{N_{a^*,t}}}, \forall d \in [D],
\tag{77}
$$

$$
\hat{\boldsymbol{r}}_{a_t,t}(d) \leq \boldsymbol{\mu}_{a_t}(d) + \sqrt{\frac{\log\left(\frac{t}{\alpha}\right)}{N_{a_t,t}}}, \forall d \in [D],
\tag{78}
$$

Combining Eq. 75 with Eq. 77 implies

$$\tilde{\boldsymbol{c}}_t^T \left( \hat{\boldsymbol{r}}_{a_t,t} + \sqrt{\frac{\log\left(\frac{t}{\alpha}\right)}{N_{a_t,t}}} \boldsymbol{e} \right) \geq \overline{\boldsymbol{c}}^T \left( \hat{\boldsymbol{r}}_{a^*,t} + \sqrt{\frac{\log\left(\frac{t}{\alpha}\right)}{N_{a^*,t}}} \boldsymbol{e} \right) \geq \overline{\boldsymbol{c}}^T \boldsymbol{\mu}_{a^*}. \tag{79}$$

By the definition of regret in Eq. 2 and facts above, we can derive the upper-bound of instantaneous regret as follows:

$$\text{regret}_t = \overline{\boldsymbol{c}} \boldsymbol{\mu}_{a^*} - \overline{\boldsymbol{c}} \boldsymbol{\mu}_{a_t} \underset{(a)}{\leq} \tilde{\boldsymbol{c}}_t^T \left( \hat{\boldsymbol{r}}_{a_t,t} + \sqrt{\frac{\log\left(\frac{t}{\alpha}\right)}{N_{a_t,t}}} \boldsymbol{e} \right) - \overline{\boldsymbol{c}}^T \boldsymbol{\mu}_{a_t}$$

$$\underset{(b)}{\leq} (\tilde{\boldsymbol{c}}_t - \overline{\boldsymbol{c}})^T \boldsymbol{\mu}_{a_t} + 2\|\tilde{\boldsymbol{c}}_t\|_1 \sqrt{\frac{\log\left(\frac{t}{\alpha}\right)}{N_{a_t,t}}}$$

$$\underset{(c)}{\implies} \text{regret}_t \leq \min\left( (\tilde{\boldsymbol{c}}_t - \overline{\boldsymbol{c}})^T \boldsymbol{\mu}_{a_t} + 2\sqrt{\frac{\log\left(\frac{t}{\alpha}\right)}{N_{a_t,t}}}, 1 \right)$$

$$\leq \underbrace{\min\left( (\tilde{\boldsymbol{c}}_t - \overline{\boldsymbol{c}})^T \boldsymbol{\mu}_{a_t}, 1 \right)}_{\text{regret}_t^{\tilde{c}}} + \underbrace{2\sqrt{\frac{\log\left(\frac{t}{\alpha}\right)}{N_{a_t,t}}}}_{\text{regret}_t^{\tilde{r}}} \tag{80}$$

where (a) follows Eq. 79, (b) follows Eq. 78, and (c) holds by the facts that $\text{regret}_t \leq 1$, and the optimistic preference solution $\boldsymbol{c}'$ of policy satisfies $\|\boldsymbol{c}'\|_1 \leq 1$. Interestingly, the derived instantaneous regret above can also be interpreted as the sum of two components:

- $\text{regret}_t^{\tilde{c}}$: Regret caused by the imprecise estimation of preference $\overline{\boldsymbol{c}}$.

- $\text{regret}_t^{\tilde{r}}$: Regret caused by the imprecise estimation of expected reward of arms.

Plugging above results back to Eq. 74, we have

$$R(T) \leq M + \sum_{t=M+1}^{T} \text{regret}_t$$

$$\leq M + \sum_{t=M+1}^{T} \left( \text{regret}_t^{\tilde{c}} + \text{regret}_t^{\tilde{r}} \right) \tag{81}$$

$$\leq M + \underbrace{\sum_{t=M+1}^{T} \min\left( (\tilde{\boldsymbol{c}}_t - \overline{\boldsymbol{c}})^T \boldsymbol{\mu}_{a_t}, 1 \right)}_{R_{M+1:T}^{\tilde{c}}} + \underbrace{\sum_{t=M+1}^{T} 2\sqrt{\frac{\log\left(\frac{t}{\alpha}\right)}{N_{a_t,t}}}}_{R_{M+1:T}^{\tilde{r}}},$$

which also yields two components of $R_{M+1:T}^{\tilde{c}}$ and $R_{M+1:T}^{\tilde{r}}$, denoting the accumulated truncated expected errors caused by the imprecise estimations of preference and reward respectively. Next we analyze two components of $R_{M+1:T}^{\tilde{c}}$ and $R_{M+1:T}^{\tilde{r}}$ separately.

### E.2.2 STEP-2 (UPPER-BOUND OVER $R_{M+1:T}^{\tilde{c}}$)

Before the analysis of term $R_{M+1:T}^{\tilde{c}}$, we first state two useful lemmas that will be utilized in proof:

**Lemma 17.** *Let $M = \left\lfloor \min\left\{ t' \mid (t-1)\sigma_{r\downarrow}^2 + \lambda \geq 2D\sqrt{K(t-1)\log\frac{t}{\alpha}}, \forall t \geq t' \right\} \right\rfloor$, and the assumptions follow those outlined in Proposition 14, then for $t \geq M+1$, $\boldsymbol{\mu} \in \mathbb{R}^D$, and $c \in \Theta_t$,*

$$\left| (\boldsymbol{c} - \hat{\boldsymbol{c}}_t)^T \boldsymbol{\mu} \right| \leq \sqrt{2\beta_t \boldsymbol{\mu}^T \mathbb{E}[\Upsilon_t]^{-1} \boldsymbol{\mu}}.$$

Please see Appendix E.3 for the proof of Lemma 17.

**Lemma 18.** *Let* $M = \left\lfloor \min\left\{ t' \mid (t-1)\sigma_{r\downarrow}^2 + \lambda \geq 2D\sqrt{K(t-1)\log\frac{t}{\alpha}}, \forall t \geq t' \right\} \right\rfloor$, *and assumptions follow those outlined in Proposition 14, then we have:*

$$\sum_{t=M+1}^{T} \min\left( \sqrt{2\beta_t \boldsymbol{\mu}_{a_t}^T \mathbb{E}[\Upsilon_t] \boldsymbol{\mu}_{a_t}}, \frac{1}{2} \right) \leq \sqrt{\frac{\beta_T D}{2\log(5/4)}(T-M)\log\left( 1 + \frac{1+\sigma_{r\uparrow}^2}{\lambda}(T-M) \right)}.$$

Please see Appendix E.4 for the proof of Lemma 18.

Define $M = \left\lfloor \min\left\{ t' \mid (t-1)\sigma_{r\downarrow}^2 + \lambda \geq 2D\sqrt{K(t-1)\log\frac{t}{\alpha}}, \forall t \geq t' \right\} \right\rfloor$. Please note that for $\sigma_{r\downarrow}^2 > 0$, we have $\lim_{t\to\infty} \frac{2D\sqrt{K(t-1)\log\frac{t}{\alpha}}}{\sigma_{r\downarrow}^2(t-1)} = \lim_{t\to\infty} C_1 \sqrt{\frac{\log(t)-C_2}{t-1}} = 0$ since as $t$ increase, $\sqrt{\log(t)}$ grows very slowly compared to $\sqrt{t-1}$. Hence for sufficiently large $t'$, the inequality $(t-1)\sigma_{r\downarrow}^2 + \lambda \geq 2D\sqrt{K(t-1)\log\frac{t}{\alpha}}, \forall t \geq t'$ holds, which implies that such an $M$ does indeed exist. By Lemma 17, for any $t \in [M+1, T]$, we have

$$
\begin{aligned}
\text{regret}_t^{\tilde{c}} &= \min\left( (\tilde{\boldsymbol{c}}_t - \overline{\boldsymbol{c}})^T \boldsymbol{\mu}_{a_t}, 1 \right) \\
&\underset{(a)}{\leq} \min\left( \left| (\tilde{\boldsymbol{c}}_t - \hat{\boldsymbol{c}}_t)^T \boldsymbol{\mu}_{a_t} \right| + \left| (\hat{\boldsymbol{c}}_t - \overline{\boldsymbol{c}})^T \boldsymbol{\mu}_{a_t} \right|, 1 \right) \\
&\underset{(b)}{\leq} \min\left( 2\sqrt{2\beta_t \boldsymbol{\mu}_{a_t}^T \mathbb{E}[\Upsilon_t]^{-1} \boldsymbol{\mu}_{a_t}}, 1 \right) \qquad \text{(by Lemma 17)} \\
&= 2\min\left( \sqrt{2\beta_t \boldsymbol{\mu}_{a_t}^T \mathbb{E}[\Upsilon_t]^{-1} \boldsymbol{\mu}_{a_t}}, \frac{1}{2} \right).
\end{aligned}
\tag{82}
$$

where (a) holds since

$$
\begin{aligned}
(\tilde{\boldsymbol{c}}_t - \overline{\boldsymbol{c}})^T \boldsymbol{\mu}_{a_t} = (\tilde{\boldsymbol{c}}_t - \hat{\boldsymbol{c}}_t + \hat{\boldsymbol{c}}_t - \overline{\boldsymbol{c}})^T \boldsymbol{\mu}_{a_t} &= (\tilde{\boldsymbol{c}}_t - \hat{\boldsymbol{c}}_t)^T \boldsymbol{\mu}_{a_t} + (\hat{\boldsymbol{c}}_t - \overline{\boldsymbol{c}})^T \boldsymbol{\mu}_{a_t} \\
&\leq \left| (\tilde{\boldsymbol{c}}_t - \hat{\boldsymbol{c}}_t)^T \boldsymbol{\mu}_{a_t} \right| + \left| (\hat{\boldsymbol{c}}_t - \overline{\boldsymbol{c}})^T \boldsymbol{\mu}_{a_t} \right|,
\end{aligned}
$$

(b) holds since both $\tilde{\boldsymbol{c}}_t$ and $\overline{\boldsymbol{c}}$ are located within the confidence region $\Theta_t$ and $t > M$, and we can thus apply Lemma 17 on both $|(\tilde{\boldsymbol{c}}_t - \hat{\boldsymbol{c}}_t)^T \boldsymbol{\mu}_{a_t}|$ and $|(\hat{\boldsymbol{c}}_t - \overline{\boldsymbol{c}})^T \boldsymbol{\mu}_{a_t}|$ respectively.

Summing $\text{regret}_t^{\tilde{c}}$ over $[M+1, T]$ and apply Lemma 18 derives the truncated regret component of $R_{M+1:T}^{\tilde{c}}$ as follows:

$$
\begin{aligned}
R_{M+1:T}^{\tilde{c}} &\leq 2 \sum_{t=M+1}^{T} \min\left( \sqrt{2\beta_t \boldsymbol{\mu}_{a_t}^T \mathbb{E}[\Upsilon_t]^{-1} \boldsymbol{\mu}_{a_t}}, \frac{1}{2} \right) \\
&\leq 2\sqrt{\frac{\beta_T D}{2\log(5/4)}(T-M)\log\left( 1 + \frac{1+\sigma_{r\uparrow}^2}{\lambda}(T-M) \right)}. \qquad \text{(by Lemma 18)}
\end{aligned}
\tag{83}
$$

### E.2.3 STEP-3 (UPPER-BOUND OVER $R^{\tilde{r}}_{M+1:T}$)

For the truncated regret component $R^{\tilde{r}}_{M+1:T}$ caused by imprecise estimation of reward, we have

$$
\begin{aligned}
R^{\tilde{r}}_{M+1:T} = 2 \sum_{t=M+1}^{T} \sqrt{\frac{\log\left(\frac{t}{\alpha}\right)}{N_{a_t,t}}} &\underset{(a)}{\leq} 2\sqrt{\log\left(\frac{T}{\alpha}\right)} \sum_{i=1}^{K} \sum_{n=N_{i,M+1}}^{N_{i,T}} \sqrt{\frac{1}{n}} \\
&\underset{(b)}{\leq} 2\sqrt{\log\left(\frac{T}{\alpha}\right)} \sum_{i=1}^{K} \sum_{n=N_{i,1}}^{N_{i,T-M}} \sqrt{\frac{1}{n}} \\
&\underset{(c)}{\leq} 2\sqrt{\log\left(\frac{T}{\alpha}\right)} \sum_{i=1}^{K} \sum_{n=1}^{\frac{T-M}{K}} \sqrt{\frac{1}{n}} \\
&\underset{(d)}{\leq} 2\sqrt{\log\left(\frac{T}{\alpha}\right)} \sum_{i=1}^{K} 2\sqrt{\frac{T-M}{K}} \\
&= 4\sqrt{K\log\left(\frac{T}{\alpha}\right)(T-M)}.
\end{aligned}
\tag{84}
$$

Specifically, in step (a), we breakdown the totally truncated horizon by the episodes that each individual arm $i \in [K]$ was pulled, and replace $t$ with upper-bound $T$ in the original numerator. Step (b) trivially holds since $\frac{1}{N_{i,t+M}} \leq \frac{1}{N_{i,t}}$ is strictly true for all $i \in [K]$. Step (c) follows from the fact that the entire sum is maximized when all arms are pulled an equal number of times. (d) holds since the fact that $2\sqrt{n} - 2 \leq \sum_{x=1}^{n} \frac{1}{\sqrt{x}} \leq 2\sqrt{n}$.

### E.2.4 STEP-4 (DERIVING FINAL REGRET)

Based on above results, we can derive the final regret $R(T)$. Specifically, plug Eq. 83 and Eq. 84 back to Eq. 81, define $M = \left\lfloor \min\left\{t' \mid (t-1)\sigma^2_{r\downarrow} + \lambda \geq 2D\sqrt{K(t-1)\log\frac{t}{\alpha}}, \forall t \geq t'\right\}\right\rfloor$, and choose

$$
\beta_t = \left(\sqrt{\lambda} + \sqrt{D\log\left(1 + \frac{t-1}{\lambda}\right) + 4\log\left(\frac{\pi t}{\sqrt{2\vartheta}}\right)}\right)^2 \quad \text{and} \quad \alpha = \sqrt{\frac{8\vartheta}{KD(D+3)\pi^2}},
$$

we have with probability at least $1 - \vartheta$, the expected regret of PUCB-HPM satisfies

$$
R(T) \leq 2\sqrt{\frac{\beta_T D}{2\log(\frac{5}{4})}\log\left(1 + \frac{(1+\sigma^2_{r\uparrow})(T-M)}{\lambda}\right)(T-M)} + 4\sqrt{K\log\left(\frac{T}{\alpha}\right)(T-M)} + M, \tag{85}
$$

which concludes the proof of Theorem 6. $\qquad\square$

### E.3 PROOF OF LEMMA 17

To begin with, we state an essential lemma that will be utilized in the proof of Lemma 17. Specifically, the following lemma characterizes the size of confidence ellipse $\Theta_t$ for preference estimation $\hat{c}_t$ with respect to $\mathbb{E}[\Upsilon_t]$-norm. The detailed proof of Lemma 19 is provided in Appendix E.3.1.

**Lemma 19.** *Let* $M = \left\lfloor \min\left\{t' \mid (t-1)\sigma^2_{r\downarrow} + \lambda \geq 2D\sqrt{K(t-1)\log\frac{t}{\alpha}}, \forall t \geq t'\right\}\right\rfloor$. *Assume Event C in Proposition 14 holds, for* $t \geq M+1$*, and any* $c \in \Theta_t$*,*

$$
(c - \hat{c}_t)^T \mathbb{E}[\Upsilon_t](c - \hat{c}_t) \leq 2\beta_t.
$$

*Proof of Lemma 17.* Let $M = \left\lfloor \min\left\{ t' \mid (t-1)\sigma_{r\downarrow}^2 + \lambda \geq 2D\sqrt{K(t-1)\log\frac{t}{\alpha}}, \forall t \geq t' \right\} \right\rfloor$. By applying Cauchy-Schwarz inequality and Lemma 19, we can obtain for any $t \in (M, T]$, any $\boldsymbol{c} \in \Theta_t$,

$$
\begin{aligned}
\left| (\boldsymbol{c} - \hat{\boldsymbol{c}}_t)^T \boldsymbol{\mu} \right| &= \left| (\boldsymbol{c} - \hat{\boldsymbol{c}}_t)^T \mathbb{E}[\Upsilon_t]^{\frac{1}{2}} \mathbb{E}[\Upsilon_t]^{-\frac{1}{2}} \boldsymbol{\mu} \right| \\
&= \left| \left( \mathbb{E}[\Upsilon_t]^{\frac{1}{2}} (\boldsymbol{c} - \hat{\boldsymbol{c}}_t) \right)^T \mathbb{E}[\Upsilon_t]^{-\frac{1}{2}} \boldsymbol{\mu} \right| \\
&\underset{(a)}{\leq} \left\| \mathbb{E}[\Upsilon_t]^{\frac{1}{2}} (\boldsymbol{c} - \hat{\boldsymbol{c}}_t) \right\|_2 \left\| \mathbb{E}[\Upsilon_t]^{-\frac{1}{2}} \boldsymbol{\mu} \right\|_2 \\
&= \left\| \mathbb{E}[\Upsilon_t]^{\frac{1}{2}} (\boldsymbol{c} - \hat{\boldsymbol{c}}_t) \right\|_2 \sqrt{\boldsymbol{\mu}^T \mathbb{E}[\Upsilon_t]^{-1} \boldsymbol{\mu}} \\
&\underset{(b)}{\leq} \sqrt{2\beta_t} \sqrt{\boldsymbol{\mu}^T \mathbb{E}[\Upsilon_t]^{-1} \boldsymbol{\mu}},
\end{aligned}
\tag{86}
$$

where inequality (a) follows Cauchy-Schwarz, (b) holds by applying Lemma 19. $\qquad\square$

### E.3.1 PROOF OF LEMMA 19

Before the proof, we state two lemmas that will be utilized in the derivation as follows.

**Lemma 20** (Eigenvalues of Sums of Hermitian Matrices (Fulton, 2000), Eq.(11)). *Let $\boldsymbol{A}$ and $\boldsymbol{B}$ are $n \times n$ Hermitian matrices with eigenvalues $a_1 > a_2 > ... > a_n$ and $b_1 > b_2 > ... > b_n$. Let $\boldsymbol{C} = \boldsymbol{A} + \boldsymbol{B}$ and the eigenvalues of $\boldsymbol{C}$ are $c_1 > c_2 > ... > c_n$, then we have*

$$
c_{n-i-j} \geq a_{n-i} + b_{n-j}, \forall i, j \in [0, n-1].
$$

**Lemma 21** (Eigenvalue Bounds on Quadratic Forms). *Assuming $A \in \mathbb{R}^{n \times n}$ is symmetric, then for any $\boldsymbol{x} \in \mathbb{R}^n$, the quadratic form is bounded by the product of the minimum and maximum eigenvalues of $A$ and the square of the norm of $\boldsymbol{x}$:*

$$
\max(\boldsymbol{\lambda_A}) \|\boldsymbol{x}\|_2^2 \geq \boldsymbol{x}^T \boldsymbol{A} \boldsymbol{x} \geq \min(\boldsymbol{\lambda_A}) \|\boldsymbol{x}\|_2^2,
$$

*where $\boldsymbol{\lambda_A}$ is the eigenvalues of $\boldsymbol{A}$.*

The detailed proof of Lemma 21 can be found in Appendix E.3.2.

*Proof of Lemma 19.* First, let's recall the definitions of $\mathbb{E}[\Upsilon_t]$ and $\Upsilon_t$ for $t \in (2, T]$:

$$
\begin{aligned}
\mathbb{E}[\Upsilon_t] &= \sum_{\iota=1}^{t-1} \mathbb{E}[\boldsymbol{r}_{a_{\iota,\iota}} \boldsymbol{r}_{a_{\iota,\iota}}^T] + \lambda \boldsymbol{I} = \sum_{i=1}^{K} \mathbb{E}\left[ \sum_{\iota \in \mathcal{T}_{i,t-1}} \boldsymbol{r}_{i,\iota} \boldsymbol{r}_{i,\iota}^T \right] + \lambda \boldsymbol{I} \\
&= \sum_{i=1}^{K} N_{i,t} \mathbb{E}[\boldsymbol{r}_i \boldsymbol{r}_i^T] + \lambda \boldsymbol{I} = \sum_{i=1}^{K} N_{i,t} \left( \boldsymbol{\mu}_i \boldsymbol{\mu}_i^T + \Sigma_{\boldsymbol{r},\boldsymbol{i}} \right) + \lambda \boldsymbol{I},
\end{aligned}
\tag{87}
$$

$$
\Upsilon_t = \sum_{i=1}^{K} \sum_{\iota=1}^{N_{i,t}} \boldsymbol{r}_{i,\iota} \boldsymbol{r}_{i,\iota}^T + \lambda \boldsymbol{I}.
$$

where $\Sigma_{\boldsymbol{r},i} = \begin{bmatrix} \sigma_{r,i,1}^2 & & 0 \\ & \ddots & \\ 0 & & \sigma_{r,i,D}^2 \end{bmatrix}_{d \times d}$ denotes the covariance matrix of reward.

Due to the assumption that event C holds, we have $\forall i \in [K], \forall m \in [D], \forall n \in [D]$,

$$
\mathbb{E}\left[ \sum_{\iota \in \mathcal{T}_{i,t-1}} \boldsymbol{r}_{i,\iota} \boldsymbol{r}_{i,\iota}^T \right](m, n) - \sqrt{N_{i,t} \log\left(\frac{t}{\alpha}\right)} \leq \sum_{\iota \in \mathcal{T}_{i,t-1}} \left( \boldsymbol{r}_{i,\iota} \boldsymbol{r}_{i,\iota}^T \right)(m, n),
$$

By the definition of $\Theta_t$ and symmetry of $\mathbb{E}[\Upsilon_t]$ and $\Upsilon_t$, for any $\boldsymbol{c} \in \Theta_t$, we can easily get

$$\beta_t \geq (\boldsymbol{c} - \hat{\boldsymbol{c}}_t)^T \left( \sum_{i=1}^{K} \left( N_{i,t} \mathbb{E}[\boldsymbol{r}_i \boldsymbol{r}_i^T] - \sqrt{N_{i,t} \log\left(\frac{t}{\alpha}\right)} \boldsymbol{e} \boldsymbol{e}^T \right) + \lambda \boldsymbol{I} \right) (\boldsymbol{c} - \hat{\boldsymbol{c}}_t)$$

$$= (\boldsymbol{c} - \hat{\boldsymbol{c}}_t)^T \left( \sum_{i=1}^{K} \left( N_{i,t} \left( \boldsymbol{\mu}_i \boldsymbol{\mu}_i^T + \Sigma_{\boldsymbol{r},\boldsymbol{i}} \right) - \sqrt{N_{i,t} \log\left(\frac{t}{\alpha}\right)} \boldsymbol{e} \boldsymbol{e}^T \right) + \lambda \boldsymbol{I} \right) (\boldsymbol{c} - \hat{\boldsymbol{c}}_t). \tag{88}$$

Next we make a preliminary analysis over the norm-distances of $\|\boldsymbol{c} - \hat{\boldsymbol{c}}_t\|_{\sum_{i=1}^{K} \left(N_{i,t} \boldsymbol{\mu}_i \boldsymbol{\mu}_i^T\right)}^2$ and $\|\boldsymbol{c} - \hat{\boldsymbol{c}}_t\|_{\sum_{i=1}^{K} \left(N_{i,t} \Sigma_{\boldsymbol{r},\boldsymbol{i}}\right)}^2$ respectively.

Let

$$p = \underset{i \in [K]}{\arg\min} \, (\boldsymbol{c} - \hat{\boldsymbol{c}}_t)^T \boldsymbol{\mu}_i \boldsymbol{\mu}_i^T (\boldsymbol{c} - \hat{\boldsymbol{c}}_t)$$

$$q = \underset{j \in [K]}{\arg\max} \, (\boldsymbol{c} - \hat{\boldsymbol{c}}_t)^T \boldsymbol{\mu}_j \boldsymbol{\mu}_j^T (\boldsymbol{c} - \hat{\boldsymbol{c}}_t),$$

and we can obtain

$$(\boldsymbol{c} - \hat{\boldsymbol{c}}_t)^T \left( (t-1) \boldsymbol{\mu}_p \boldsymbol{\mu}_p^T \right) (\boldsymbol{c} - \hat{\boldsymbol{c}}_t)$$

$$\leq (\boldsymbol{c} - \hat{\boldsymbol{c}}_t)^T \left( \sum_{i=1}^{K} N_{i,t} \boldsymbol{\mu}_i \boldsymbol{\mu}_i^T \right) (\boldsymbol{c} - \hat{\boldsymbol{c}}_t)$$

$$\leq (\boldsymbol{c} - \hat{\boldsymbol{c}}_t)^T \left( (t-1) \boldsymbol{\mu}_q \boldsymbol{\mu}_q^T \right) (\boldsymbol{c} - \hat{\boldsymbol{c}}_t).$$

By the continuity of norm-distance, result above implies that $\exists w_1 \in [0,1]$, such that

$$(\boldsymbol{c} - \hat{\boldsymbol{c}}_t)^T \left( \sum_{i=1}^{K} N_{i,t} \boldsymbol{\mu}_i \boldsymbol{\mu}_i^T \right) (\boldsymbol{c} - \hat{\boldsymbol{c}}_t) = (\boldsymbol{c} - \hat{\boldsymbol{c}}_t)^T \left( (t-1) \tilde{\boldsymbol{\mu}} \tilde{\boldsymbol{\mu}}^T \right) (\boldsymbol{c} - \hat{\boldsymbol{c}}_t), \tag{89}$$

where $\tilde{\boldsymbol{\mu}} = w_1 \boldsymbol{\mu}_p + (1 - w_1) \boldsymbol{\mu}_q$. Similarly, for $\|\boldsymbol{c} - \hat{\boldsymbol{c}}_t\|_{\sum_{i=1}^{K} (N_{i,t} \Sigma_{\boldsymbol{r},\boldsymbol{i}})}^2$, since the covariance matrices $\Sigma_{\boldsymbol{r},\boldsymbol{i}}, \forall i \in [K]$ are diagonal, by Lemma 21, we have

$$\xi_{\min}\left( \sum_{i=1}^{K} N_{i,t} \Sigma_{\boldsymbol{r},\boldsymbol{i}} \right) \|\boldsymbol{c} - \hat{\boldsymbol{c}}_t\|_2^2 \leq (\boldsymbol{c} - \hat{\boldsymbol{c}}_t)^T \left( \sum_{i=1}^{K} N_{i,t} \Sigma_{\boldsymbol{r},\boldsymbol{i}} \right) (\boldsymbol{c} - \hat{\boldsymbol{c}}_t) \leq \xi_{\max}\left( \sum_{i=1}^{K} N_{i,t} \Sigma_{\boldsymbol{r},\boldsymbol{i}} \right) \|\boldsymbol{c} - \hat{\boldsymbol{c}}_t\|_2^2,$$

where $\xi_{\min}(\sum_{i=1}^{K} N_{i,t} \Sigma_{\boldsymbol{r},\boldsymbol{i}})$ denotes the minimum eigenvalue of matrix $\sum_{i=1}^{K} N_{i,t} \Sigma_{\boldsymbol{r},\boldsymbol{i}}$, while $\xi_{\max}(\sum_{i=1}^{K} N_{i,t} \Sigma_{\boldsymbol{r},\boldsymbol{i}})$ denotes the corresponding maximum one. We will also use $\xi(\cdot)$ to denote the eigenvalue calculator for a matrix in the following part. By the continuity of nor-distance, result above implies that there exist a constant $\tilde{\xi}_t \in \left[ \xi_{\min}(\sum_{i=1}^{K} N_{i,t} \Sigma_{\boldsymbol{r},\boldsymbol{i}}), \xi_{\max}(\sum_{i=1}^{K} N_{i,t} \Sigma_{\boldsymbol{r},\boldsymbol{i}}) \right]$, such that

$$\xi_{\min}\left( \sum_{i=1}^{K} N_{i,t} \Sigma_{\boldsymbol{r},\boldsymbol{i}} \right) \|\boldsymbol{c} - \hat{\boldsymbol{c}}_t\|_2^2$$

$$\leq \tilde{\xi}_t \|\boldsymbol{c} - \hat{\boldsymbol{c}}_t\|_2^2 = (\boldsymbol{c} - \hat{\boldsymbol{c}}_t)^T \left( \sum_{i=1}^{K} N_{i,t} \Sigma_{\boldsymbol{r},\boldsymbol{i}} \right) (\boldsymbol{c} - \hat{\boldsymbol{c}}_t)$$

$$\leq \xi_{\max}\left( \sum_{i=1}^{K} N_{i,t} \Sigma_{\boldsymbol{r},\boldsymbol{i}} \right) \|\boldsymbol{c} - \hat{\boldsymbol{c}}_t\|_2^2,$$

Note that $\sum_{i=1}^{K} N_{i,t} \Sigma_{\boldsymbol{r},\boldsymbol{i}}$ is diagonal, we have $\xi_{\min}(\sum_{i=1}^{K} N_{i,t} \Sigma_{\boldsymbol{r},\boldsymbol{i}}) = \min_{d \in [D]} \sum_{i=1}^{K} N_{i,t} \sigma_{r,i,d} \geq (t-1) \sigma_{r\downarrow}^2$, and similarly, $\xi_{\max}(\sum_{i=1}^{K} N_{i,t} \Sigma_{\boldsymbol{r},\boldsymbol{i}}) \leq (t-1) \sigma_{r\uparrow}^2$. Define $\tilde{\sigma}_{r,t}^2 = \frac{\tilde{\xi}_t}{(t-1)}$, and we have $\tilde{\sigma}_{r,t}^2 \in [\sigma_{r\downarrow}^2, \sigma_{r\uparrow}^2]$ and satisfies

$$(t-1) \tilde{\sigma}_{r,t}^2 \|\boldsymbol{c} - \hat{\boldsymbol{c}}_t\|_2^2 = (\boldsymbol{c} - \hat{\boldsymbol{c}}_t)^T \left( \sum_{i=1}^{K} N_{i,t} \Sigma_{\boldsymbol{r},\boldsymbol{i}} \right) (\boldsymbol{c} - \hat{\boldsymbol{c}}_t). \tag{90}$$

By plugging above result back into the Eq 88 and using the definition in Eq 87, we have

$$\beta_t \geq (\boldsymbol{c} - \hat{\boldsymbol{c}}_t)^T \left( \sum_{i=1}^{K} \left( N_{i,t} \mathbb{E}[\boldsymbol{r}_i \boldsymbol{r}_i^T] - \sqrt{N_{i,t} \log\left(\frac{t}{\alpha}\right)} \boldsymbol{e}\boldsymbol{e}^T \right) + \lambda \boldsymbol{I} \right) (\boldsymbol{c} - \hat{\boldsymbol{c}}_t)$$

$$\underset{(a)}{=} (\boldsymbol{c} - \hat{\boldsymbol{c}}_t)^T \left( (t-1)\tilde{\boldsymbol{\mu}}\tilde{\boldsymbol{\mu}}^T + \left((t-1)\tilde{\sigma}_{r,t}^2 + \lambda\right) \boldsymbol{I} - \sum_{i=1}^{K} \sqrt{N_{i,t} \log\left(\frac{t}{\alpha}\right)} \boldsymbol{e}\boldsymbol{e}^T \right) (\boldsymbol{c} - \hat{\boldsymbol{c}}_t)$$

$$\underset{(b)}{\geq} (\boldsymbol{c} - \hat{\boldsymbol{c}}_t)^T \left( \underbrace{(t-1)\tilde{\boldsymbol{\mu}}\tilde{\boldsymbol{\mu}}^T}_{\boldsymbol{A}_t} + \underbrace{\left((t-1)\tilde{\sigma}_{r,t}^2 + \lambda\right) \boldsymbol{I}}_{\boldsymbol{B}_t} - \underbrace{\sqrt{K(t-1) \log\left(\frac{t}{\alpha}\right)} \boldsymbol{e}\boldsymbol{e}^T}_{\boldsymbol{C}_t} \right) (\boldsymbol{c} - \hat{\boldsymbol{c}}_t).$$

$$(91)$$

where (a) holds by Eq. 89 and Eq. 90, (b) holds since the squared root term is maximized when $N_{i,t} = (t-1)/K, \forall i \in [K]$. Note that $\boldsymbol{B}_t$ is diagonal matrix, and $-\boldsymbol{C}_t$ is rank-1 matrix yields one eigenvalue of $-\sqrt{K(t-1) \log\left(\frac{t}{\alpha}\right)}\|\boldsymbol{e}\|_2^2 = -D\sqrt{K(t-1) \log\left(\frac{t}{\alpha}\right)}$ and $D-1$ eigenvalues of 0, we have

$$\xi_{\min}(B_t - C_t) = (t-1)\tilde{\sigma}_{r,t}^2 + \lambda - D\sqrt{K(t-1) \log\left(\frac{t}{\alpha}\right)}.$$

Due to $t \geq M+1$, we can trivially derive $(t-1)\tilde{\sigma}_{r,t}^2 + \lambda \geq (t-1)\sigma_{r\downarrow}^2 + \lambda \geq 2D\sqrt{K(t-1) \log\frac{t}{\alpha}}$, implying that the minimum eigenvalue $\xi_{\min}(B_t - C_t) \geq 0$ and the matrix $\boldsymbol{B}_t - \boldsymbol{C}_t$ is a positive semi- definite matrix, and thus $\boldsymbol{A}_t + \boldsymbol{B}_t - \boldsymbol{C}_t$ is positive-definite. Also note that $\boldsymbol{A}_t + \boldsymbol{B}_t - \boldsymbol{C}_t$ is symmetric, by Lemma 21, we can derive that

$$\xi_{\min}\left(\boldsymbol{A}_t + \boldsymbol{B}_t - \boldsymbol{C}_t\right) \|\boldsymbol{c} - \hat{\boldsymbol{c}}_t\|_2^2 \leq (\boldsymbol{c} - \hat{\boldsymbol{c}}_t)^T \left(\boldsymbol{A}_t + \boldsymbol{B}_t - \boldsymbol{C}_t\right) (\boldsymbol{c} - \hat{\boldsymbol{c}}_t) \leq \beta_t$$

$$\underset{(a)}{\Longrightarrow} \|\boldsymbol{c} - \hat{\boldsymbol{c}}_t\|_2^2 \leq \frac{\beta_t}{\xi_{\min}\left(\boldsymbol{A}_t + \boldsymbol{B}_t - \boldsymbol{C}_t\right)}, \tag{92}$$

where $\xi_{\min}\left(\boldsymbol{A}_t + \boldsymbol{B}_t - \boldsymbol{C}_t\right)$ is the minimum eigenvalue of $\boldsymbol{A}_t + \boldsymbol{B}_t - \boldsymbol{C}_t$, and the implication (a) holds since $\xi_{\min}\left(\boldsymbol{A}_t + \boldsymbol{B}_t - \boldsymbol{C}_t\right) > 0$ due to the positive-definite of $\boldsymbol{A}_t + \boldsymbol{B}_t - \boldsymbol{C}_t$.

Note that $\boldsymbol{A}_t$ is rank-1 matrix and $\boldsymbol{B}_t$ is diagonal matrix, we can trivially derive that $\boldsymbol{A}_t + \boldsymbol{B}_t$ has one eigenvalue of $(t-1)(\|\mu\|_2^2 + \tilde{\sigma}_{r,t}^2) + \lambda$ and $D-1$ eigenvalues of $(t-1)\tilde{\sigma}_{r,t}^2 + \lambda$. Also, $-\boldsymbol{C}_t$ has one eigenvalue of $-\sqrt{K(t-1) \log\left(\frac{t}{\alpha}\right)}\|\boldsymbol{e}\|_2^2 = -D\sqrt{K(t-1) \log\left(\frac{t}{\alpha}\right)}$ and $D-1$ eigenvalues of 0.

Since $\boldsymbol{A}_t + \boldsymbol{B}_t$ and $-\boldsymbol{C}_t$ are both symmetric, by applying Lemma 20, we have

$$\xi_{\min}\left(\boldsymbol{A}_t + \boldsymbol{B}_t - \boldsymbol{C}_t\right) \geq \xi_{\min}\left(\boldsymbol{A}_t + \boldsymbol{B}_t\right) + \xi_{\min}\left(-\boldsymbol{C}_t\right)$$

$$= (t-1)\tilde{\sigma}_{r,t}^2 + \lambda - D\sqrt{K(t-1) \log\left(\frac{t}{\alpha}\right)}$$

Plugging above result back into Eq. 93, we have

$$\|\boldsymbol{c} - \hat{\boldsymbol{c}}_t\|_2^2 \leq \frac{\beta_t}{\xi_{\min}\left(\boldsymbol{A}_t + \boldsymbol{B}_t - \boldsymbol{C}_t\right)} \leq \frac{\beta_t}{(t-1)\tilde{\sigma}_{r,t}^2 + \lambda - D\sqrt{K(t-1) \log\left(\frac{t}{\alpha}\right)}}. \tag{93}$$

Again, since $t \geq M+1$ holds, the denominator of the final term is strictly positive. Combining above result with Eq. 91 and rearranging the terms, for $t \geq M+1$, we can obtain

$$(\boldsymbol{c} - \hat{\boldsymbol{c}}_t)^T \left( \boldsymbol{A}_t + \boldsymbol{B}_t \right) (\boldsymbol{c} - \hat{\boldsymbol{c}}_t) \leq \beta_t + (\boldsymbol{c} - \hat{\boldsymbol{c}}_t)^T \boldsymbol{C}_t \left( \boldsymbol{c} - \hat{\boldsymbol{c}}_t \right)$$

$$\underset{(a)}{\leq} \beta_t + \xi_{\max} \left( \boldsymbol{C}_t \right) \|\boldsymbol{c} - \hat{\boldsymbol{c}}_t\|_2^2$$

$$\underset{(b)}{\leq} \beta_t + \frac{\beta_t D \sqrt{K(t-1) \log \left( \frac{t}{\alpha} \right)}}{(t-1)\tilde{\sigma}_{r,t}^2 + \lambda - D \sqrt{K(t-1) \log \left( \frac{t}{\alpha} \right)}} \qquad (94)$$

$$= \beta_t + \frac{\beta_t}{\frac{(t-1)\tilde{\sigma}_{r,t}^2 + \lambda}{D \sqrt{K(t-1) \log \left( \frac{t}{\alpha} \right)}} - 1}$$

$$\underset{(c)}{\leq} 2\beta_t,$$

where (a) follows from Lemma 21, (b) holds since Eq. 93 and $\xi_{\max} \left( \boldsymbol{C}_t \right) = -\xi_{\min} \left( -\boldsymbol{C}_t \right) = D\sqrt{K(t-1) \log \left( \frac{t}{\alpha} \right)}$, (c) holds since $(t-1)\tilde{\sigma}_{r,t}^2 + \lambda \geq 2D\sqrt{K(t-1) \log \left( \frac{t}{\alpha} \right)}$ for $t \geq M+1$.

By Eq. 89 and the definition of $\mathbb{E}[\Upsilon_t]$ in Eq. 87, we have

$$\mathbb{E}[\Upsilon_t] = \sum_{i=1}^{K} N_{i,t} \left( \boldsymbol{\mu}_i \boldsymbol{\mu}_i^T + \Sigma_{r,i} \right) + \lambda \boldsymbol{I} = (t-1)\tilde{\boldsymbol{\mu}}\tilde{\boldsymbol{\mu}}^T + (t-1)\tilde{\sigma}_{r,t}^2 \boldsymbol{I} + \lambda \boldsymbol{I} = \boldsymbol{A}_t + \boldsymbol{B}_t,$$

and thus for $t \geq M+1$,

$$(\boldsymbol{c} - \hat{\boldsymbol{c}}_t)^T \mathbb{E}[\Upsilon_t] \left( \boldsymbol{c} - \hat{\boldsymbol{c}}_t \right) \leq 2\beta_t.$$

$\square$

### E.3.2  PROOF OF LEMMA 21

*Proof.* The quadratic form $\boldsymbol{x}^T \boldsymbol{A} \boldsymbol{x}$ can be analyzed by decomposing $\boldsymbol{A}$ using its eigenvalues and eigenvectors. Since $\boldsymbol{A}$ is a symmetric matrix, we can write it as:

$$\boldsymbol{A} = \boldsymbol{Q} \boldsymbol{\Lambda} \boldsymbol{Q}^T,$$

where $\boldsymbol{Q}$ is an orthogonal matrix whose columns are the eigenvectors of $\boldsymbol{A}$, and $\boldsymbol{\Lambda}$ is a diagonal matrix with the eigenvalues $\boldsymbol{\lambda_A}(i)$ on its diagonal. By substituting the eigen-decomposition of $\boldsymbol{A}$, we have

$$\boldsymbol{x}^T \boldsymbol{A} \boldsymbol{x} = \boldsymbol{x}^T \boldsymbol{Q} \boldsymbol{\Lambda} \boldsymbol{Q}^T \boldsymbol{x}.$$

Let $\boldsymbol{y} = \boldsymbol{Q}^T \boldsymbol{x}$, then we have

$$\boldsymbol{x}^T \boldsymbol{A} \boldsymbol{x} = \boldsymbol{y}^T \boldsymbol{\Lambda} \boldsymbol{y} = \sum_{i=1}^{n} \boldsymbol{\lambda_A}(i) \boldsymbol{y}(i)^2 \geq \min\left(\boldsymbol{\lambda_A}\right) \sum_{i=1}^{n} \boldsymbol{y}(i)^2 = \min\left(\boldsymbol{\lambda_A}\right) \|\boldsymbol{y}\|_2^2 \underset{(a)}{=} \min\left(\boldsymbol{\lambda_A}\right) \|\boldsymbol{x}\|_2^2.$$

where (a) follows since $\|\boldsymbol{y}\|_2^2 = \|\boldsymbol{Q}^T \boldsymbol{x}\|_2^2 = \|\boldsymbol{x}\|_2^2$ as $\boldsymbol{Q}$ is orthogonal and preserves the norm. For $\max\left(\boldsymbol{\lambda_A}\right) \|\boldsymbol{x}\|_2^2, \geq \boldsymbol{x}^T \boldsymbol{A} \boldsymbol{x}$, the proof follows similarly and is therefore omitted. $\square$

### E.4  PROOF OF LEMMA 18

Since $\beta_t \geq 1$ and is increasing with $t$, we have

$$\sum_{t=M+1}^{T} \min\left( \sqrt{2\beta_t \boldsymbol{\mu}_{a_t}^T \mathbb{E}[\Upsilon_t]^{-1} \boldsymbol{\mu}_{a_t}}, \frac{1}{2} \right) \leq \sum_{t=M+1}^{T} \min\left( \sqrt{2\beta_T \boldsymbol{\mu}_{a_t}^T \mathbb{E}[\Upsilon_t]^{-1} \boldsymbol{\mu}_{a_t}}, \frac{1}{2}\sqrt{2\beta_T} \right)$$

$$\leq \sqrt{2\beta_T} \sum_{t=M+1}^{T} \min\left( \sqrt{\boldsymbol{\mu}_{a_t}^T \mathbb{E}[\Upsilon_t]^{-1} \boldsymbol{\mu}_{a_t}}, \frac{1}{2} \right). \qquad (95)$$

To derive the upper-bound of term $\sum_{t=M+1}^{T} \min\left(\sqrt{\boldsymbol{\mu}_{a_t}^T \mathbb{E}[\Upsilon_t]^{-1} \boldsymbol{\mu}_{a_t}}, \frac{1}{2}\right)$, we follow the similar techniques for analyzing the sum of instantaneous regret in OFUL (Abbasi-Yadkori et al., 2011). Specifically, we first show that the sum of squared terms $\min\left(\sqrt{\boldsymbol{\mu}_{a_t}^T \mathbb{E}[\Upsilon_t]^{-1} \boldsymbol{\mu}_{a_t}}, \frac{1}{2}\right)^2$ yields an upper-bound sub-linear to $T$, and then extend the result to the sum of $\min\left(\sqrt{\boldsymbol{\mu}_{a_t}^T \mathbb{E}[\Upsilon_t]^{-1} \boldsymbol{\mu}_{a_t}}, \frac{1}{2}\right)$.

We begin with stating the following lemmas from which Lemma 18 follows.

**Lemma 22.** *For any action sequence of $a_1, ..., a_T$ and any $M \in (0, T)$, we have*

$$\det\left(\mathbb{E}[\Upsilon_{T+1}]\right) \geq \det\left(\mathbb{E}[\Upsilon_{M+1}]\right) \prod_{t=M+1}^{T} \left(1 + \frac{\det\left(\Sigma_{r,a_t}\right)}{\det\left(\mathbb{E}[\Upsilon_t]\right)} + \boldsymbol{\mu}_{a_t}^T \mathbb{E}[\Upsilon_t]^{-1} \boldsymbol{\mu}_{a_t}\right).$$

Please see Appendix E.4.1 for the detailed proof of Lemma 22.

**Lemma 23.** *For any action sequence of $a_1, ..., a_T$ with $\|\boldsymbol{\mu}_{a_t}\|_2^2 \leq B, \forall t \in [T]$, then for any $M \in (0, T)$, we have*

$$\log\left(\frac{\det\left(\mathbb{E}[\Upsilon_{T+1}]\right)}{\det\left(\mathbb{E}[\Upsilon_{M+1}]\right)}\right) \leq D \log\left(1 + \frac{B + D\sigma_{r\uparrow}^2}{D\lambda}(T - M)\right).$$

Please see Appendix E.4.2 for the detailed proof of Lemma 23.

*Proof of Lemma 18.* **Step-1:** We first show that the sum of squared terms in Eq. 95 is optimal up to $\mathcal{O}(\log(T - M))$. Specifically,

$$
\begin{aligned}
\sum_{t=M+1}^{T} &\min\left(\sqrt{\boldsymbol{\mu}_{a_t}^T \mathbb{E}[\Upsilon_t]^{-1} \boldsymbol{\mu}_{a_t}}, \frac{1}{2}\right)^2 \\
&= \sum_{t=M+1}^{T} \min\left(\boldsymbol{\mu}_{a_t}^T \mathbb{E}[\Upsilon_t]^{-1} \boldsymbol{\mu}_{a_t}, \frac{1}{4}\right) \\
&\overset{\leq}{\underset{(a)}{}} \sum_{t=M+1}^{T} \frac{1}{4\log(5/4)} \log\left(1 + \min\left(\boldsymbol{\mu}_{a_t}^T \mathbb{E}[\Upsilon_t]^{-1} \boldsymbol{\mu}_{a_t}, \frac{1}{4}\right)\right) \\
&\leq \sum_{t=M+1}^{T} \frac{1}{4\log(5/4)} \log\left(1 + \boldsymbol{\mu}_{a_t}^T \mathbb{E}[\Upsilon_t]^{-1} \boldsymbol{\mu}_{a_t}\right)
\end{aligned}
\tag{96}
$$

where (a) holds since the fact that $\log(1 + x) \geq 4 \log\left(\frac{5}{4}\right) x$ for $x \leq \frac{1}{4}$.

On the other hand, Lemma 22 implies that

$$\log\left(\frac{\det\left(\mathbb{E}[\Upsilon_{T+1}]\right)}{\det\left(\mathbb{E}[\Upsilon_{M+1}]\right)}\right) \geq \sum_{t=M+1}^{T} \log\left(1 + \frac{\det\left(\Sigma_{r,a_t}\right)}{\det\left(\mathbb{E}[\Upsilon_t]\right)} + \boldsymbol{\mu}_{a_t}^T \mathbb{E}[\Upsilon_t]^{-1} \boldsymbol{\mu}_{a_t}\right). \tag{97}$$

Additionally, since $\det\left(\Sigma_{r,a_t}/\mathbb{E}[\Upsilon_t]\right) > 0$ and $\|\boldsymbol{\mu}_i\|_2^2 \leq \|\boldsymbol{\mu}_i\|_1^2 \leq D, \forall i \in [K]$, by Lemma 23, we have

$$D \log\left(1 + \frac{1 + \sigma_{r\uparrow}^2}{\lambda}(T - M)\right) \geq \log\left(\frac{\det\left(\mathbb{E}[\Upsilon_{T+1}]\right)}{\det\left(\mathbb{E}[\Upsilon_{M+1}]\right)}\right) \geq \sum_{t=M+1}^{T} \log\left(1 + \boldsymbol{\mu}_{a_t}^T \mathbb{E}[\Upsilon_t]^{-1} \boldsymbol{\mu}_{a_t}\right). \tag{98}$$

Plugging the above result back into Eq. 96, we can derive a bound up to $\mathcal{O}(\log(T - M))$ on the sum of squared instantaneous regrets in Eq. 95 as:

$$\sum_{t=M+1}^{T} \min\left(\sqrt{\boldsymbol{\mu}_{a_t}^T \mathbb{E}[\Upsilon_t]^{-1} \boldsymbol{\mu}_{a_t}}, \frac{1}{2}\right)^2 \leq \frac{D}{4\log(5/4)} \log\left(1 + \frac{1 + \sigma_{r\uparrow}^2}{\lambda}(T - M)\right). \tag{99}$$

**Step-2:** Given the upper-bound on the sum of squared instantaneous regrets , we next extend it to the sum of instantaneous regrets by using Cauchy-Schwarz inequality. Specifically,

$$\sum_{t=M+1}^{T} \min\left(\sqrt{\boldsymbol{\mu}_{a_t}^T \mathbb{E}[\Upsilon_t]^{-1} \boldsymbol{\mu}_{a_t}}, \frac{1}{2}\right) \underset{(a)}{\leq} \sqrt{(T - M) \sum_{t=M+1}^{T} \min\left(\sqrt{\boldsymbol{\mu}_{a_t}^T \mathbb{E}[\Upsilon_t]^{-1} \boldsymbol{\mu}_{a_t}}, \frac{1}{2}\right)^2}$$

$$\leq \sqrt{\frac{D}{4\log(5/4)}(T - M) \log\left(1 + \frac{1 + \sigma_{r\uparrow}^2}{\lambda}(T - M)\right)}. \tag{100}$$

Plugging above result back into Eq. 95 concludes the proof of Lemma 18. $\qquad\square$

### E.4.1 PROOF OF LEMMA 22

We begin with a lemma that will be utilized in the derivations of Lemma 22:

**Lemma 24** (Determinant of Symmetric PSD Matrices Sum). *Let $\boldsymbol{A} \in \mathbb{R}^{n \times n}$ be a symmetric and positive definite matrix, and $\boldsymbol{B} \in \mathbb{R}^{n \times n}$ be a symmetric and positive (semi-) definite matrix. Then we have*

$$\det(\boldsymbol{A} + \boldsymbol{B}) \geq \det(\boldsymbol{A}) + \det(\boldsymbol{B})$$

*Proof.*

$$\det(\boldsymbol{A} + \boldsymbol{B}) = \det(\boldsymbol{A}) \det\left(\boldsymbol{I} + \boldsymbol{A}^{-\frac{1}{2}} \boldsymbol{B} \boldsymbol{A}^{-\frac{1}{2}}\right). \tag{101}$$

Let $\lambda_1, ..., \lambda_n$ be the eigenvalues of $\boldsymbol{A}^{-\frac{1}{2}} \boldsymbol{B} \boldsymbol{A}^{-\frac{1}{2}}$. Since $\boldsymbol{A}^{-\frac{1}{2}} \boldsymbol{B} \boldsymbol{A}^{-\frac{1}{2}}$ is positive (semi-) definite, we have $\lambda_i \geq 0, \forall i \in [n]$, which implies

$$\det\left(\boldsymbol{I} + \boldsymbol{A}^{-\frac{1}{2}} \boldsymbol{B} \boldsymbol{A}^{-\frac{1}{2}}\right) = \prod_{i=1}^{n}(1 + \lambda_i) \geq 1 + \prod_{i=1}^{n} \lambda_i = \det(\boldsymbol{I}) + \det\left(\boldsymbol{A}^{-\frac{1}{2}} \boldsymbol{B} \boldsymbol{A}^{-\frac{1}{2}}\right). \tag{102}$$

Combining Eq.101 with Eq. 102 concludes the proof.

$\qquad\square$

*Proof of Lemma 22.* For $\Upsilon_t$ and $\mathbb{E}[\Upsilon_t]$, by definition,

$$\Upsilon_{t+1} = \Upsilon_t + \boldsymbol{r}_{a_t,t} \boldsymbol{r}_{a_t,t}^T \quad \text{and} \quad \Upsilon_1 = \lambda \boldsymbol{I},$$
$$\mathbb{E}[\Upsilon_{t+1}] = \mathbb{E}[\Upsilon_t] + \boldsymbol{\mu}_{a_t} \boldsymbol{\mu}_{a_t}^T + \Sigma_{r,a_t}.$$

Since $\mathbb{E}[\Upsilon_t]$ is symmetric and positive definite, we have

$$\det(\mathbb{E}[\Upsilon_{t+1}]) = \det\left(\mathbb{E}[\Upsilon_t] + \boldsymbol{\mu}_{a_t} \boldsymbol{\mu}_{a_t}^T + \Sigma_{r,a_t}\right)$$

$$= \det\left(\mathbb{E}[\Upsilon_t]^{\frac{1}{2}} \left(\boldsymbol{I} + \mathbb{E}[\Upsilon_t]^{-\frac{1}{2}} \left(\boldsymbol{\mu}_{a_t} \boldsymbol{\mu}_{a_t}^T + \Sigma_{r,a_t}\right) \mathbb{E}[\Upsilon_t]^{-\frac{1}{2}}\right) \mathbb{E}[\Upsilon_t]^{\frac{1}{2}}\right)$$

$$= \det(\mathbb{E}[\Upsilon_t]) \det\left(\boldsymbol{I} + \mathbb{E}[\Upsilon_t]^{-\frac{1}{2}} \left(\boldsymbol{\mu}_{a_t} \boldsymbol{\mu}_{a_t}^T + \Sigma_{r,a_t}\right) \mathbb{E}[\Upsilon_t]^{-\frac{1}{2}}\right)$$

$$\underset{(a)}{\geq} \det(\mathbb{E}[\Upsilon_t]) \left(\det\left(\boldsymbol{I} + \mathbb{E}[\Upsilon_t]^{-\frac{1}{2}} \boldsymbol{\mu}_{a_t} \boldsymbol{\mu}_{a_t}^T \mathbb{E}[\Upsilon_t]^{-\frac{1}{2}}\right) + \det\left(\mathbb{E}[\Upsilon_t]^{-\frac{1}{2}} \Sigma_{r,a_t} \mathbb{E}[\Upsilon_t]^{-\frac{1}{2}}\right)\right) \tag{103}$$

where (a) holds since both $\left( \boldsymbol{I} + \mathbb{E}[\Upsilon_t]^{-\frac{1}{2}} \boldsymbol{\mu}_{a_t} \boldsymbol{\mu}_{a_t}^T \mathbb{E}[\Upsilon_t]^{-\frac{1}{2}} \right)$ and $\left( \mathbb{E}[\Upsilon_t]^{-\frac{1}{2}} \Sigma_{r,a_t} \mathbb{E}[\Upsilon_t]^{-\frac{1}{2}} \right)$ are positive definite and applying Lemma 24 yields the result.

Let $\mathbb{E}[\Upsilon_t]^{-\frac{1}{2}} \boldsymbol{\mu}_{a_t} = \boldsymbol{v}_t$, and we observe that

$$\left( \boldsymbol{I} + \boldsymbol{v}_t \boldsymbol{v}_t^T \right) \boldsymbol{v}_t = \boldsymbol{v}_t + \boldsymbol{v}_t \left( \boldsymbol{v}_t^T \boldsymbol{v}_t \right) = \left( 1 + \boldsymbol{v}_t^T \boldsymbol{v} \right) \boldsymbol{v}_t.$$

Hence, $1 + \boldsymbol{v}_t^T \boldsymbol{v}$ is an eigenvalue of $\boldsymbol{I} + \boldsymbol{v}_t \boldsymbol{v}_t^T$. And since $\boldsymbol{v}_t \boldsymbol{v}_t^T$ is a rank-1 matrix, all other eigenvalue of $\boldsymbol{I} + \boldsymbol{v}_t \boldsymbol{v}_t^T$ equal to 1, implying

$$\begin{aligned}
\det \left( \boldsymbol{I} + \mathbb{E}[\Upsilon_t]^{-\frac{1}{2}} \boldsymbol{\mu}_{a_t} \boldsymbol{\mu}_{a_t}^T \mathbb{E}[\Upsilon_t]^{-\frac{1}{2}} \right) &= \det \left( \boldsymbol{I} + \boldsymbol{v}_t \boldsymbol{v}_t^T \right) \\
&= 1 + \boldsymbol{v}_t \boldsymbol{v}_t^T \\
&= 1 + \left( \mathbb{E}[\Upsilon_t]^{-\frac{1}{2}} \boldsymbol{\mu}_{a_t} \right)^T \left( \mathbb{E}[\Upsilon_t]^{-\frac{1}{2}} \boldsymbol{\mu}_{a_t} \right) \\
&= 1 + \boldsymbol{\mu}_{a_t}^T \mathbb{E}[\Upsilon_t]^{-1} \boldsymbol{\mu}_{a_t}.
\end{aligned} \tag{104}$$

Combining Eq. 103 and Eq. 104, we have

$$\det \left( \mathbb{E}[\Upsilon_{t+1}] \right) \geq \det \left( \mathbb{E}[\Upsilon_t] \right) \left( 1 + \boldsymbol{\mu}_{a_t}^T \mathbb{E}[\Upsilon_t]^{-1} \boldsymbol{\mu}_{a_t} + \det \left( \mathbb{E}[\Upsilon_t]^{-\frac{1}{2}} \Sigma_{r,a_t} \mathbb{E}[\Upsilon_t]^{-\frac{1}{2}} \right) \right)$$

The solution of Lemma 22 follows from induction. $\qquad \square$

### E.4.2 PROOF OF LEMMA 23

*Proof.* For the proof of this lemma, we follow the main idea of Determinant-Trace Inequality in OFUL (Abbasi-Yadkori et al., 2011) (Lemma 10). Specifically, by the definition of $\Upsilon_t$, we have

$$\begin{aligned}
\log \left( \frac{\det \left( \mathbb{E}[\Upsilon_{T+1}] \right)}{\det \left( \mathbb{E}[\Upsilon_{M+1}] \right)} \right) &= \log \left( \det \left( \frac{\mathbb{E}[\Upsilon_{M+1}] + \sum_{t=M+1}^T (\boldsymbol{\mu}_{a_t} \boldsymbol{\mu}_{a_t}^T + \Sigma_{r,a_t})}{\mathbb{E}[\Upsilon_{M+1}]} \right) \right) \\
&\overset{(a)}{\leq} \log \left( \det \left( 1 + \frac{\sum_{t=M+1}^T (\boldsymbol{\mu}_{a_t} \boldsymbol{\mu}_{a_t}^T + \Sigma_{r,a_t})}{\lambda \boldsymbol{I}} \right) \right) \\
&= \log \left( \det \left( 1 + \frac{1}{\lambda} \left( \sum_{t=M+1}^T (\boldsymbol{\mu}_{a_t} \boldsymbol{\mu}_{a_t}^T + \Sigma_{r,a_t}) \right) \right) \right),
\end{aligned} \tag{105}$$

where (a) holds since $\det(\mathbb{E}[\Upsilon_{M+1}]) \geq \det(\mathbb{E}[\Upsilon_1]) = \lambda \boldsymbol{I}$. Let $\xi_1, ..., \xi_D$ denote the eigenvalues of $\sum_{t=M+1}^T (\boldsymbol{\mu}_{a_t} \boldsymbol{\mu}_{a_t}^T + \Sigma_{r,a_t})$, and note:

$$\begin{aligned}
\sum_{d=1}^D \xi_d &= \text{Trace} \left( \sum_{t=M+1}^T (\boldsymbol{\mu}_{a_t} \boldsymbol{\mu}_{a_t}^T + \Sigma_{r,a_t}) \right) \\
&= \sum_{t=M+1}^T \text{Trace} \left( \boldsymbol{\mu}_{a_t} \boldsymbol{\mu}_{a_t}^T \right) + \sum_{t=M+1}^T \text{Trace} \left( \Sigma_{r,a_t} \right) \\
&\leq \sum_{t=M+1}^T \| \boldsymbol{\mu}_{a_t} \|_2^2 + (T-M) D \sigma_{r\uparrow}^2 \qquad \text{(by } \sigma_{r,i,d}^2 \leq \sigma_{r\uparrow}^2 \text{)} \\
&\leq (T-M)(B + D \sigma_{r\uparrow}^2). \qquad \text{(by } \| \boldsymbol{\mu}_{a_t} \|_2^2 \leq B \text{)}
\end{aligned} \tag{106}$$

Combining Eq. 105 and Eq. 106 implies

$$\log\left(\frac{\det\left(\mathbb{E}[\Upsilon_{T+1}]\right)}{\det\left(\mathbb{E}[\Upsilon_{M+1}]\right)}\right) \leq \log\left(\det\left(1 + \frac{1}{\lambda}\left(\sum_{t=M+1}^{T}\left(\boldsymbol{\mu}_{a_t}\boldsymbol{\mu}_{a_t}^T + \Sigma_{r,a_t}\right)\right)\right)\right)$$

$$= \log\left(\prod_{i=1}^{D}\left(1 + \frac{\xi_i}{\lambda}\right)\right)$$

$$= D\log\left(\prod_{i=1}^{D}\left(1 + \frac{\xi_i}{\lambda}\right)\right)^{\frac{1}{D}}$$

$$\underset{(a)}{\leq} D\log\left(\frac{1}{D}\sum_{i=1}^{D}\left(1 + \frac{\xi_i}{\lambda}\right)\right)$$

$$\underset{(b)}{\leq} D\log\left(1 + \frac{(T-M)(B + D\sigma_{r\uparrow}^2)}{D\lambda}\right),$$

where (a) follows from the inequality of arithmetic and geometric means, and (b) follows from Eq. 106. □

