# OpenReview forum: "Provably Efficient Multi-Objective Bandit Algorithms under Preference-Centric Customization"
_ICLR.cc/2025/Conference — Submitted to ICLR 2025_

### Official Review · Reviewer_dpuc · 2024-10-29

**Soundness:** 2
**Presentation:** 2
**Contribution:** 2
**Rating:** 3
**Confidence:** 4

**Summary:**

This paper studies the MO-MAB framework, integrating user preferences. It first introduces a formalization of the Preference-Aware MO-MAB and subsequently presents four algorithms tailored to different levels of knowledge regarding user preferences.

**Strengths:**

1. This paper provides a comprehensive examination of the various forms of preferences.

2. The use of multiple figures throughout the paper enhances clarity and readability.

**Weaknesses:**

1. This paper proposes a variety of algorithms and theorems; however, it demonstrates minimal innovation, relying largely on straightforward applications of existing methods. Specific observations are provided below.

2. **Algorithm 1** appears redundant. The primary challenge in bandit problems is the parameter estimation under uncertainty; assuming a known preference does not add any complexity when compared to single-objective bandit problem.

3. **Algorithm 2** addresses stationary preferences where the expectation of preference is constant. Given the independence of reward and preference, there does not seem to be any technical difficulty in managing this scenario.

4. **Algorithm 3** is designed for non-stationary preferences. The sliding window approach employed here is a well-established technique for adapting to dynamic environments in bandit settings. Additionally, the "Restart" method from [1] offers a simpler solution for this type of problem.

 [1] Peng Zhao, Lijun Zhang, Yuan Jiang, and Zhi-Hua Zhou. "A simple approach for non-stationary linear bandits." In *International Conference on Artificial Intelligence and Statistics*, 2020.

5. **Algorithm 4** addresses hidden preferences, whose expectation is fixed. This algorithm estimates the hidden preference by using multiple rewards as features and overall-rewards as reward—another mature approach in the field of bandit algorithms.

6. **Proposition 1**: For preference-free algorithms, it is straightforward to construct an example with a regret bound of $O(T)$ due to conflicting objectives. For instance, consider two arms with expected rewards of $[0,1]$ and $[1,0]$; it is evident that, for any bandit algorithm, the regret for at least one objective is $\Omega(T)$. Does your proof offer any technical novelty? If so, please highlight this aspect.

**Questions:**

Please refer to weaknesses.

---

> ### Author Response · Authors · 2024-11-21
> **Response to W1, W2**
>
> We thank the reviewer for these constructive feedbacks. We hope our point-to-point response can address the reviewer's concerns and clarify our contribution. We have revised the paper accordingly to address the comments.
>
> >  **W1**: This paper proposes a variety of algorithms and theorems; however, it demonstrates minimal innovation, relying largely on straightforward applications of existing methods. Specific observations are provided below.
>
>  **A1**:  We respectfully disagree with the statement that our work has little innovation or contribution. Our paper contains technical novelties in both algorithm design and theoretical analysis.
>
> For example, for the unknown preference case (Alg. 2), we propose a new analytical method to address the joint impact from the preference and reward estimate errors, by introducing a tunable parameter $\epsilon$ to decompose the suboptimal actions into two disjoint sets. Then we characterize the regret for each disjoint component separately. Both these steps and our new developments therein differ from the standard approach (please see A3 for details).
>
> For unknown, non-stationary preference cases (Alg. 3), the dynamic environments lead to a potential mismatch between the dynamic episodes where an arm $i$ acting suboptimally and the episodes where it is deemed suboptimal under preference estimates, making it challenging to analyze the decomposed regret terms based on preference estimates. To address this, we theoretically characterized  the learner’s behavior under UCB-based policy built on preference estimates and prove that it serves as an upper bound for the regret under true dynamic preferences (see A4 for details).
>
> For the hidden preference cases (Alg. 4), we address the new challenge in preference estimation due to random mappings from reward to overall-reward by rigorously exploring the use of regression for preference estimation in this setting. In the regret analysis, the randomness of input rewards makes it challenging to directly bound the expected error term $\sum_{t=1}^{T} |(c_t - \hat{c_t})^{T} \mu_{a_t}|$ using the empirical confidence region $\Vert \Upsilon_t^{\frac{1}{2}} (c - \hat{c}_t) \Vert_2 \leq \sqrt{\beta_t}$. To address this, we derive a novel upper bound based on the confidence region normed by the expected Gram matrix $\mathbb{E}[\Upsilon_t]$, enabling regret analysis involving $\mu$ with new theoretical insights (see A5 for details).
>
> > **W2: Algorithm 1**  appears redundant. The primary challenge in bandit problems is the parameter estimation under uncertainty; assuming a known preference does not add any complexity when compared to a single-objective bandit problem.
>
>  **A2**: Thank you for your feedback. We acknowledge that the assumption of known preferences in Algorithm 1 significantly simplifies the problem. However, this setting serves as a **warm-up** for understanding the structure of the problem and tackling the subsequent more challenging case where preferences are unknown. We have now pointed this out in the revised paper.

---

> ### Author Response · Authors · 2024-11-21
> **Response to W3**
>
> > **W3: Algorithm 2**  for constant preference. Given the independence of reward and preference, there does not seem to be any technical difficulty in managing this scenario.
>
>  **A3**:  We respectfully disagree with this comment since the regret analysis of Algorithm 2 is non-trivial. Note that despite independence between reward and preference, the chosen action depends jointly on both reward and preference estimations. This joint dependency introduces unique challenges in regret analysis for PAMO-MAB problem. Treating reward and preference errors independently would fail to capture their coupled impact on regret, resulting in a worst regret bound with an order of at least $O(\log^2 T)$ (please see the Independent Analysis part below for the details).
> In our work, we propose a new analytical method to address this very issue of having a coupled impact. Specifically, we first decompose the suboptimal actions into two disjoint sets by introducing a tunable parameter $\epsilon$, and then we characterize the regret for each disjoint component separately. Both these steps and our new developments therein are different from the standard MAB analysis. Our new method resolves the overlooked joint impact issue of preference and reward estimation errors, achieving a much tighter regret bound with order of $O(\log⁡T)$ compared to treating them independently, and is near-optimal (for the details see Our Method part below).
>
> We elaborate the details as follows (please note our method is highlighted in Section 6.1, Line 355-Line 362 in main body, and the complete proof sketch can be found in Appendix D.1.1):
> - **Independent Analysis**:
> 	For regret, we have
> $$\begin{aligned}
> R(T)&=sum_{t=1}^{T}\overline{{c}}^T(\mu_{a^*}-\mu_{a_t})=\sum_{t=1}^{T}(\overline{c}-\hat{c_t}+\hat{c_t})^{T}(\mu_{a^*}-\mu_{a_t}-\hat{r_{a^*,t}}+\hat{r_{a^*,t}}-\hat{r_{a_t,t}}+\hat{r_{a_t,t}})
> \end{aligned}$$
> $$\begin{aligned}
> \qquad=\underbrace{\sum_{t=1}^{T}(\overline{c}-\hat{c_t})^T(\mu_{a^*}-\hat{r_{a^*,t}}+\hat{r_{a_t,t}}-\mu_{a_t})}_ {Term1}+\sum_{t=1}^{T}(\hat{c_t}^{T}(\mu_{a^*}-\mu_{a_t})+\overline{c}^{T}(\hat{r_{a^*,t}}-\hat{r_{a_t,t}})).
> \end{aligned}$$
> ByCauchy–Schwarzinequality,wehaveterm1as:
> $$\begin{aligned}
> &\sum_{t=1}^{T}(\overline{c}-\hat{c_t})^T(\mu_{a^*}-\hat{r_{a^*,t}}+\hat{r_{a_t,t}}-\mu_{a_t})
> \end{aligned}$$
> $$\begin{aligned}
> \qquad\leq\sqrt{(\sum_{t=1}^{T}\\|\overline{c}-\hat{c_t}\\|^2)(\sum_{t=1}^{T}\\|(\mu_{a^*}-\hat{r_{a^*,t}})+(\hat{r_{a^*,t}}-\mu_{a^*})\\|^2)}
> \end{aligned}$$
> $$\begin{aligned}
> \qquad\leq\sqrt{\underbrace{\Big(\sum_{t=1}^{T}\sum_{d=1}^{D}(\overline{c}(d)-\hat{c_t}(d))^2\Big)}_ {TermA}\Big(\underbrace{\sum_{t=1}^{T}\sum_{d=1}^{D}(\mu_{a^*}(d)-\hat{r_{a^*,t}}(d))+\hat{r_{a^*,t}}(d)-\mu_{a^*}(d))^2}_ {TermB}\Big)}.
> \end{aligned}$$
> This enables us to analyze the regret by analyzing two errors independently. Since both $\hat{c}_ t$ and $\hat{r}_ {i,t}$ are empirical averages, by Hoeffding's inequality, we have with high probability, $\text{Term A} = O(\log^2 T)$ and $\text{Term B} = O(\log^2 T)$.  Hence, $\text{Term 1} = O(\log^2 T)$ w.h.p., implying the regret derived by naive independent analysis would have at least order of $O(\log^2 T)$.
> - **Our Method**
>     - To tackle the challenge of joint-dependency in PAMO-MAB, we employ a tunable measure $\epsilon$ to decompose the regret into two major parts: (1) regret under precise preference estimation (quantified by $\epsilon$), (2) regret under imprecise preference estimation. This framework systematically disentangles the intertwined effects of estimated preferences and rewards, allowing us to rigorously characterize regrets within each component separately (see Step-1, Eq. (24) in Appendix D.1.2).
>     - For the regret with precise preference estimation, we show that the pseudo episode set $\mathcal{M}_i$ where the suboptimal arm $i$ is considered suboptimal under the preference estimate align with its true suboptimal episode set $\mathcal{T}_i$ (Step-2, Eq. (25) in Appendix D.1.2), and the best arm is consistently identified as better than arm $i$. Using this insight, we transfer this case to a new preference known instance with a narrower overall-reward gap w.r.t $\epsilon$. This transformation allows us to effectively characterize the regret caused by reward estimation error (Eq. (26) in Appendix D.1.2).
>     - For the regret with imprecise preference estimation, we relax the suboptimal event set to an overall-reward estimation error set, eliminating the joint dependency on reward and preference from action $a_t$ (see Step-3, Eq. (28) in Appendix D.1.2). Then we develop a tailored-made error bound on preference estimation (Lemma 10), which transfers the original error set to a uniform imprecise estimation set on preference (Step-3, Eq. (29) in Appendix D.1.2), such that a tractable formulation of the estimation deviation can be constructed (Eq. (30), Appendix D.1.2).
>
>     Our method yields a near-optimal bound ($O(\log⁡T)$).

---

> > ### Comment · Reviewer_dpuc · 2024-11-22
> > **Respond to the Independent Analysis**
> >
> > A similar regret bound can be derived without resorting to your complex analytical approach. Specifically, we can bound the regret as follows:
> >
> > $$R(T)=\sum_{t=1}^{T}\overline{c}^T(\mu_{a^*}-\mu_{a_t})=\sum_{t=1}^{T}(\overline{c}-\hat{c_t}+\hat{c_t})^T(\mu_{a^*}-\mu_{a_t}) $$
> > $$=\sum_{t=1}^{T}(\overline{c}-\hat{c_t})^T(\mu_{a^*}-\mu_{a_t})+\sum_{t=1}^{T}\hat{c_t}^T(\mu_{a^*}-\mu_{a_t})$$
> > $$ \leq O\left(\sqrt{\sum_{t=1}^{T}(\mu_{a^*}-\mu_{a_t})^2}\cdot\log T\right)+O(\sqrt{KT}\cdot\log T)$$
> > $$=O\left(\sqrt{T}\cdot\log T\right)+O(\sqrt{KT}\cdot\log T).$$
> >
> > This is a more straightforward manner allows us to reduce the complexity from $O(\log^2 T)$ to $O(\log T)$. Although the above is a minimax type regret bound, I believe your analysis can be further simplified.

---

> > > ### Author Response · Authors · 2024-11-23
> > > **Response to Reviewer dpuc**
> > >
> > > We appreciate the reviewer’s follow-up and providing the alternative analysis gives an instance-independent bound of $O(\sqrt{KT} \log T)$. However, we demonstrate that this bound is worse than the instance-independent bound $O(\sqrt{KT\log T})$, which can be translated from our result in Theorem 3.
> > >
> > > Specifically, let $S := \\{ i \mid 0<\eta_i\leq\eta\\}$,  we have
> > > $$
> > > R(T) = \sum_{i \in [K]} \eta_i N_{a_i,T} = \sum_{i \in S} \eta_i N_{a_i,T} + \sum_{i \in [K] \setminus S} \eta_i N_{a_i,T}
> > > $$
> > > $$
> > > \qquad \leq \eta\sum_{i \in S} N_{a_i,T} + \sum_{i \in [K] \setminus S} \eta_i N_{a_i,T}
> > > $$
> > > $$
> > > \qquad \leq \eta T + C\frac{K \log T}{\eta},
> > > $$
> > > where the last inequality holds by $\sum_{i \in S} N_{a_i,T} \leq T$, and for $i \in [K] \setminus S$, $\eta_i N_{a_i,T} \leq C\frac{\log T}{\eta_i} \leq C\frac{\log T}{\eta}$ by Theorem 3. Choosing $\eta = \sqrt{\frac{CK \log T}{T}}$ yields
> > > $$R(T) \leq 2\sqrt{CKT\log T} = O(\sqrt{KT\log T}),$$
> > > which is tighter than the alternative bound $O(\sqrt{KT}\log T)$. Please let us know if there is any misunderstanding.

---

> > > > ### Comment · Reviewer_dpuc · 2024-11-23
> > > > **Respond to the instance-independent regret bound**
> > > >
> > > > Based on the $\epsilon$-based analysis presented in the proof of Theorem 3, the regret bound is given by
> > > >
> > > > $$R(T) \leq O\left(\frac{K \log T}{\eta} + K\eta + \frac{K}{\eta^{3/2}} + K\eta\right),$$
> > > >
> > > > which differs from the instance-independent regret bound stated in the authors' response.

---

> ### Author Response · Authors · 2024-11-21
> **Response to W4**
>
> > **W4: Algorithm 3**  is designed for non-stationary preferences. The sliding window approach employed here is a well-established technique for adapting to dynamic environments in bandit settings. Additionally, the "Restart" method from [1] offers a simpler solution for this type of problem.
>
> **A4:** Thank you for your feedback. We clarify that the application of the sliding window method in PAMO-MAB introduces new challenges in regret analysis. In a non-stationary environment, the episodes during which an arm $i$ acts suboptimally are dynamic and may not align with the episodes where arm $i$ is deemed suboptimal based on preference estimates. This creates challenges in regret analysis, as we characterize the regret under estimated preferences, while the mismatch with true suboptimal episodes can lead to inaccurate regret evaluation. To address this, we develop an upper bound characterizing the learner’s behavior under UCB-based policy built on preference estimates and show that it serves as an upper bound for the regret under true dynamic preference (details refer to the bullet in New Regret Analysis in Our Work mentioned below). Moreover, we assert that using the restart method would result in a much larger regret under our setting (refer to the bullet in Restart Method [1] Analysis mentioned below).
>
> We elaborate the details as follows:
> **New Regret Analysis in Our Work**:
> We first decompose the regret into two parts: (1) regret under precise preference estimation, (2) regret under imprecise preference estimation, following the analytical framework mentioned in **A3** (Eq. (53), Step-1 in Appendix D.4.2). For the regret with precise preference estimation, the dependency on the true preference is eliminated, which means we can only characterize the regret based on the pseudo episode set $\mathcal{M}_i$, where arm $i$ is considered suboptimal under the estimated preference. However, due to the dynamic nature of the preferences, the true suboptimal episode set $\mathcal{T}_i$, where arm $i$ is truly suboptimal, is not fixed. This potential misalignment between $\mathcal{M}_i$ and $\mathcal{T}_i$ could lead to incorrect regret analysis if not properly addressed. In our analysis, we show that $\mathcal{M}_i$ forms a superset of $\mathcal{T}_i$ (see Step-2, Eq. (25) in Appendix D.1.2) and establish an upper bound (Proposition 8, Appendix C.1) that characterizes the learner’s behavior within the pseudo episode set $\mathcal{M}_i$ under the UCB-based policy built on preference estimates. This allows us to transform the original problem to a known preference case with a narrower overall-reward gap determined by $\epsilon$.
>
> **Restart Method [1] Analysis**:
> We clarify that using the restart method [1] in our setting will result in a larger regret. The Restart method resets the learning process periodically, which introduces a cold start problem at the beginning of each epoch and increases regret. Assuming no statistical detection is employed and the breakpoints are unknown in advance, under the non-stationary preference settings of PAMO-MAB (following the assumptions in Theorem 6), using the Restart method yields an upper bound of:
> $R(T) \leq \sum_{i=1}^{K} \eta_{i}^{\uparrow}
> \big(
> \frac{4 (\delta + \frac{\delta}{\sqrt{D}} )^2 \log (T/\alpha)}{ (\eta_i^{\downarrow})^2 }
> +
> D \frac{\pi^2 \alpha^2}{3}
> +
> \psi_{T} H
> +
> \frac{D^2(1+\sqrt{D})^2 \delta^2 \Vert \Delta_{i}^{\uparrow} \Vert_2^2}{\eta_i^{\downarrow 2}}
> (\frac{T}{H} - \psi_{T})
> \big),$
>
> where $H$ is the size of each epoch. By setting the restarting period optimally as
> $H = \frac{D (1+\sqrt{D}) \delta \Vert \Delta_{i}^{\uparrow} \Vert_2}{\eta_i^{\downarrow}} \sqrt{\frac{T}{\psi_{T}}}$, Restart achieves the regret of $O(\sqrt{\psi_{T} T})$, which is worse than the dependency on $T$ of the regret of our method $O(\psi_{T}^{\frac{2}{3}} T^{\frac{1}{3}})$.
>
> It is worth noting that for the real deployment of the sliding window approach, we can achieve the same time complexity with the Restart method. Specifically, by utilizing an efficient data structure like a _Deque_ in Python (which offers $O(1)$ time complexity for append and pop operations) and employing an incremental method for updating the preference estimation within the sliding window, we can also achieve $O(1)$ complexity per step. This matches the computational efficiency of the Restart method.
>
> [1] Peng Zhao, Lijun Zhang, Yuan Jiang, and Zhi-Hua Zhou. "A simple approach for non-stationary linear bandits." In  _International Conference on Artificial Intelligence and Statistics_, 2020.

---

> ### Author Response · Authors · 2024-11-21
> **Response to W5**
>
> > **W5: Algorithm 4**  addresses hidden preferences, whose expectation is fixed. This algorithm estimates the hidden preference by using multiple rewards as features and overall-rewards as reward—another mature approach in the field of bandit algorithms.
>
> **A5:** While we acknowledge that there are some similarities in the approaches taken in traditional bandit algorithms, our approach is different in important ways and addresses new technical challenges in both algorithm design and regret analysis:
> - **(a)** Unlike traditional problem with fixed mapping from input features to output rewards (fixed $\theta$), we aim to estimate the parameter (expected preference $\overline{c}$) under **random mapping** (random vector $c_t$). To address this issue, we develop a new analytical framework that adapts linear regression to handle random mappings, proving its effectiveness despite dependent residuals and non-stationarity in the mapping process (refer to the Algorithm design part below for details).
> - **(b)** Unlike traditional settings, with input features $x_{a_t}$ known in advance, in our problem, **both** parameter $c_t$​ and reward $r_{a_t, t}$ are **random variables with unknown means**. For regret analysis, the randomness in the Gram matrix $\Upsilon_t$​ and the unknown distribution of ${r}_ {a_t, t}$ (Eq. (9)) do not allow us to directly characterize the expected error term $\sum_{t=1}^{T}|(c_t - \hat{c_t})^T \mu_{a_t}|$ using the empirical confidence region $\Vert \Upsilon_t^{\frac{1}{2}} (c - \hat{c_t}) \Vert_2 \leq \sqrt{\beta_t}$. To address this, we derived a novel upper bound on the confidence region normed with the expected Gram matrix (Lemma 19, Appendix E.3), allowing the analysis involves $\mu$, and characterize the regret term w.r.t the the expected Gram matrix (Lemma 22, 23, Appendix E.4). Please refer to the Regret analysis part below for details.

---

> ### Author Response · Authors · 2024-11-21
> **Response to W5 (Continued)**
>
> Specifically, we elaborate the key new components as follows.
>
> **Algorithm design**:
> - **Non-trivial Generalization of Regression:** While fixed expected preference $\overline{c}$ to estimate, the choice of regression in our algorithm is a non-trivial generalization of traditional solutions due to the fundamentally different problem formulation caused by random mapping:
> 	- In traditional problems, the output is $g_{a_t, t} = \theta^\top x_{a_t} + \eta_t$, where $\eta_t$​ is an independent sub-Gaussian noise term. This formulation aligns perfectly with linear regression assumptions, where residual errors are independent of predictors.
> 	- In our problem, due to the random mapping, the output $g_{a_t, t} = c_t^\top r_{a_t, t} = (\overline{c} + \zeta_t)^\top r_{a_t, t} = \overline{c}^\top r_{a_t, t} + \zeta_t^\top r_{a_t, t}$, where $\zeta_t$​ is a zero-mean noise term denoting preference randomness. This makes the residual error term $\zeta_t^\top r_ {a_t, t}$​ **dependent on the input features**, violating standard linear regression assumptions.
>
> 	In our work, we demonstrate that the residual error $\zeta_t^\top r_{a_t, t}$​ vanishes in expectation without biasing the predictor (Line 466-Line 470 in manuscript), thereby establishing the feasibility of using standard linear regression for estimating $\overline{c}$ in PAMO-MAB. Moreover, by showing that $\zeta_t^\top r_{a_t, t}$​ is a **zero-mean innovation** ($\mathbb{E}[\zeta_t^\top r_{a_t, t} \mid \mathcal{F}_ {t-1}] = 0$ (Line 2070-Line 2076, Appendix E.1), where $\mathcal{F}_ {t−1} = \sigma(a_{1:t−1}, g_{1:t−1}, r_{1:t})$​ is the filtration of past observations) and conditionally **1-sub-Gaussian distributed**, we prove that the estimation error of $\hat{c_t}$ with respect to $\overline{c}$ can be appropriately controlled (Proposition 14). This analysis further validates the effectiveness of linear regression, even in the presence of random mapping from input to output.
> **Regret analysis**:
> - As mentioned in **A3**, the regret analysis of PAMO-MAB faces the unique challenge of joint dependency on both preferences and rewards. To address this, based on the assumptions in Proposition 14, we also introduce a decomposition to the hidden preference case (see Eq. (80), Appendix E.2.1), which divides two regret components caused by preference estimation error and reward estimation error.
> - For regret w.r.t preference errors, one essential step is to bound $\sum_{t=1}^{T}|(c_t - \hat{c_t})^T \mu_{a_t}|$ to ensure it is sub-linear. The core difficulty lies in the fact that while we have a confidence bound involving $\Vert \Upsilon_t^{\frac{1}{2}} (c - c_t) \Vert_2 \leq \sqrt{\beta_t}$, analyzing ${\mu}^\top \Upsilon_t^{-1} {\mu}$ is non-trivial due to the randomness in $\Upsilon_t$​ and the unknown distribution of ${r}_{a_t, t}$​. This is fundermentally different with traditional problems [2,3] where features $x_t$​ are known and directly incorporated into $\Upsilon_t$. In our analysis:
> 	- We express and bound the term using $\mathbb{E}[\Upsilon_t]$, allowing the analysis involving $\mu$ (Eq. (87) 19, Appendix E.3.1).
> 	- We derive a novel upper-bound on $\| \mathbb{E}[\Upsilon_t]^{\frac{1}{2}} ({c} - \hat{{c}}_t) \|_2​$ (Lemma 19, Appendix E.3) involving the variance of rewards. Specifically, we show that there exists a constant $M \sim \frac{C^2}{\log C^2}$, s.t., the $\| \mathbb{E}[\Upsilon_t]^{\frac{1}{2}} ({c} - \hat{{c}}_t) \|_2​ \leq \sqrt{C \beta_t}$ holds for $t>M$. Based on this, we derive the regret w.r.t preference errors in a truncated form (Eq. (83), (81), Appendix E.2.1, E.2.2). Our novel idea may be of independent interest for other problems where both random variables involved in linear regresion.
>
>  [2] Abbasi-Yadkori et al. Improved algorithms for linear stochastic bandits. NeurIPS, 2011.
>
>  [3] Chu et al. Contextual bandits with linear payoff functions. AISTATS, 2011.

---

> ### Author Response · Authors · 2024-11-21
> **Response to W6**
>
> > **W6: Proposition 1**: For preference-free algorithms, it is straightforward to construct an example with a regret bound of  O(T)  due to conflicting objectives. For instance, consider two arms with expected rewards of  [0,1]  and  [1,0]; it is evident that, for any bandit algorithm, the regret for at least one objective is Ω(T). Does your proof offer any technical novelty? If so, please highlight this aspect.
>
>  **A6**: Thanks for the feedback. While the example provided is simple, **Proposition 1** offers a general and rigorous proof that establishes the linear regret bound for all preference-free algorithms under conflicting objectives, extending beyond specific instances.
>
> Importantly, the proposition provides insights not discussed in previous studies:
> - It formalizes how conflicting objectives universally result in linear regret for at least one objective.
> - It offers a theoretical foundation justifying the need for preference-aware methods.
>
> These contributions demonstrate both rigor and novelty, providing a basis for advancing research on multi-objective bandits.

---

> > ### Comment · Reviewer_dpuc · 2024-11-23
> > **Respond to Proposition 1**
> >
> > The proof of Proposition 1 can be simplified. I propose the following straightforward proof:
> >
> > Consider a multi-objective problem with two arms $\{a,b\}$, where the expected rewards are $[0,1]$ and $[1,0]$, respectively. For any algorithm operating $T$ rounds, let $N_a$ and $N_b$ denote the number of times arms $a$ and $b$ are selected, respectively. By definition, $N_a+N_b=T$.
> >
> > 1. **Case 1**: If the preference is $[1,0]$, arm $b$ is optimal, and the regret is $N_a$.
> > 2. **Case 2**: If the preference is $[0,1]$, arm $a$ is optimal, and the regret is $N_b$.
> >
> > In either case, the regret bound is $\Omega(T)$; otherwise, it would be impossible to satisfy $N_a + N_b = T$. Hence, there exists a subset of preferences such that the regret is $\Omega(T)$.

---

> > > ### Author Response · Authors · 2024-11-24
> > > **Respond to Proposition 1**
> > >
> > > We thank the reviewer for pointing this out and agree that this is a simpler proof for Proposition 1. But please note that the main contribution of Proposition 1 lies in the new insight it provides: emphasizing the necessity of preference-aware bandits to align outputs with user preferences. We have not claimed that the proof of Proposition 1 itself constitutes a key contribution of our work.

---

> ### Author Response · Authors · 2024-11-24
> **Respond to the instance-independent regret bound**
>
> We apologize for presenting a slightly loose upper bound in Theorem 3 for the sake of a concise expression. In fact, this bound can be tightened to $O\left(\sum_{i \in [K]} (\frac{\log T}{\eta_i} + \eta_i + \frac{1}{\eta_i}) \right)$, with a simple modification in Step 2 (Bounding $N_{i,T}^{\widetilde{\boldsymbol{c}}}$) in Appendix D.1.2 (Please see the derivation in the "Tighter Bound" part below).
>
>
> By replacing the second term ($\sum_{i \in [K] \setminus S} \eta_i N_{a_i,T}$) in our previous response with this refined bound $O\left(\sum_{i \in [K]} (\frac{\log T}{\eta_i} + \eta_i + \frac{1}{\eta_i}) \right) \leq O\left( \frac{K\log T}{\eta} + \sum_{i \in K} \eta_i + \frac{K}{\eta} \right)$, we have
> $$
> R(T) \leq \eta T + O\left( \frac{K\log T}{\eta} + \frac{K}{\eta} \right) + O(K\delta).
> $$
> By taking $\eta = \sqrt{\frac{K (1+\log T)}{T}}$, we have
> $$
> R(T) \leq O\left(\sqrt{KT(1+\log T)}\right) + O\left(K\delta\right),$$
> which asymptoticly converges to $O\left(\sqrt{KT\log T}\right)$.
>
>
> **Tighter Bound**
>
> We make a small modification on Step-2 in D.1.2 to derive a tighter bound on $N_{i,T}^{\widetilde{\boldsymbol{c}}}$. Specifically, to upper-bound the RHS term of Eq. (30) in Step-2 (Appendix D.1.2), we can directly set the $\epsilon_t = \epsilon_0$, and apply a concentration inequalities like Bernstein’s inequality to derive the upper bound:
>
> $$
> \mathbb{P}_ {\epsilon_0} \left( \Vert \overline{\boldsymbol{c}} - \hat{\boldsymbol{c}}_ t \Vert_2 \geq \frac{\epsilon_0}{ \Vert \Delta_{i} \Vert_2 } \right)
> = \mathbb{P}_ {\epsilon_0} \left( \sum_{d = 1}^{D} \left( \overline{\boldsymbol{c}} (d) - \hat{c}_ t (d) \right)^2 \geq \frac{\epsilon_0^2}{ \Vert \Delta_{i} \Vert_2^2 } \right)
> $$
> $$
>  \qquad \qquad\leq \sum_{d = 1}^{D} \mathbb{P}_ {\epsilon_0} \left( | \overline{\boldsymbol{c}} (d) - \hat{\boldsymbol{c}}_ t (d) | \geq \frac{\epsilon_0}{ \sqrt{D} \Vert \Delta_{i} \Vert_2 } \right)
> $$
> $$
> \qquad \qquad  \leq
>  2 \exp \left( - \frac{ \epsilon_0^2}{ 2 D \Vert \Delta_{i} \Vert_2^2 \sigma_{c}^2 + \frac{2}{3} \delta \epsilon_0 \sqrt{D} \Vert \Delta_{i} \Vert_2  } t \right),
> $$
> where the last inequality holds by Bernstein’s inequality (Lemma 11). Plugging back to Eq. (30) we have
>
> $$
> \mathbb{E}_ {\epsilon} \left[ N_{i,T}^{\tilde{c}} \right]
> = \mathbb{E}_ {\epsilon} \left[ \sum_{t=1}^{T} \boldsymbol{1}_ {\\{a_t = i \neq a^*, \hat{c}_ {t}^{T} \mu_{a^{* }} \leq \hat{c}_ {t}^{T} \mu_ {i} + \eta_i - \epsilon_0 \\}} \right]
> $$
> $$
> \qquad \qquad= \sum_{t=1}^{T} \mathbb{P}_ {\epsilon} \left( a_t = i \neq a^*, \hat{c}_ {t}^{T} \mu_{a^{*}} \leq \hat{c}_ {t}^{T} \mu_{i} + \eta_i - \epsilon_0 \right)
> $$
> $$
> \qquad \qquad  \leq
> \sum_{t=1}^{T} 2 D \exp \left( - \frac{ \epsilon_0^2}{ 2 D \Vert \Delta_{i} \Vert_2^2 \sigma_{c}^2 + \frac{2}{3} \delta \epsilon_0 \sqrt{D} \Vert \Delta_{i} \Vert_2  } t \right)
> $$
> $$
> \qquad \qquad \underset{(a)}{\leq}
> \frac{2 D}{ \exp \left(\frac{ \epsilon_0^2}{ 2 D \Vert \Delta_{i} \Vert_2^2 \sigma_{c}^2 + \frac{2}{3} \delta \epsilon_0 \sqrt{D} \Vert \Delta_{i} \Vert_2  } \right) -1 }
> $$
> $$
> \qquad \qquad  \leq
> \frac{4 D^2 \Vert \Delta_{i} \Vert_2^2 \sigma_{c}^2 + \frac{4}{3} \delta \epsilon_0 D^{\frac{3}{2}} \Vert \Delta_{i} \Vert_2}{ \epsilon_0^2 },
> \quad (\text{by the fact that } e^x \geq x+1, \forall x \geq 0 )
> $$
>
> where (a) holds since for any $a>0$, we have
> $$
> \sum_{t=1}^{T} \left( e^{-a} \right)^{t} =
> \sum_{t=0}^{T-1} e^{-a} \cdot \left( e^{-a} \right)^{t}
> $$
> $$ \qquad \qquad\leq
> \sum_{t=0}^{\infty} e^{-a} \cdot \left( e^{-a} \right)^{t}
> $$
> $$ \qquad \qquad= \frac{e^{-a}}{1 - e^{-a}} \quad \text{(by closed form of the geometric series)}
> $$
> $$\qquad \qquad =
> \frac{1}{e^a-1}.
> $$
>
> Combining above result with Eq. (27) in Appendix D.1.2, we have
>
> $$
> \mathbb{E}_ {\epsilon} [N_{i,T}] = \mathbb{E}_ {\epsilon} \left[ N_{i,T}^{\widetilde{\boldsymbol{r}}} \right] + \mathbb{E}_ {\epsilon} \left[ N_{i,T}^{\widetilde{\boldsymbol{c}}} \right]
> \leq
> \frac{4 \delta^2 \log (T/\alpha)}{(\eta_{i} - \epsilon_0)^2}
> +
> \frac{D \pi^2 \alpha^2}{3}
> +
> \frac{4 D^2 \Vert \Delta_{i} \Vert_2^2 \sigma_{c}^2 }{ \epsilon_0^2 }
> +
> \frac{4\delta D^{\frac{3}{2}} \Vert \Delta_{i} \Vert_2}{ 3\epsilon_0 }.
> $$
>
> By choosing $\epsilon_0 = \frac{1}{2}\eta_i$, and suming over $K$ arms, we have
> $$
> R(T) = \sum_{i \in [K]} \mathbb{E}_ {\epsilon} [N_{i,T}] \cdot \eta_i
> \leq \sum_{i \in [K]} \eta_i\left(\frac{16 \delta^2 \log (T/\alpha)}{\eta_{i}^2}
> +
> \frac{D \pi^2 \alpha^2}{3}
> +
> \frac{16 D^2 \Vert \Delta_{i} \Vert_2^2 \sigma_{c}^2 }{ \eta_i^2 }
> +
> \frac{8\delta D^{\frac{3}{2}} \Vert \Delta_{i} \Vert_2}{ 3\eta_i } \right)
> $$
> $$
> \qquad = O\left(\sum_{i \in [K]} (\frac{\log T}{\eta_i} + \eta_i + \frac{1}{\eta_i}) \right).
> $$

---

> ### Author Response · Authors · 2024-11-26
> **Your feedback is important to us**
>
> Dear Reviewer dpuc,
>
> We would greatly appreciate it if you could provide feedback on our latest response. We are more than happy to address any further questions or concerns you might have. If our response has resolved your concerns, we kindly ask you to consider raising the rating of our work. Thank you very much for your time and efforts!

---

> > ### Comment · Reviewer_dpuc · 2024-11-26
> >
> > Thank you for your feedback. I have decided to maintain my current evaluation, as this paper is not yet ready for publication.
> > 1. Algorithm 1 can be removed, and potentially Algorithm 3 as well.
> > 2. The proof of Proposition 1 can be simplified as discussed above.
> > 3. The introduction of the technical novelty requires refinement to better emphasize its contribution. As previously discussed, the current version makes it challenging for reviewers to clearly identify the innovative aspects of your technique.

---

### Official Review · Reviewer_sjsn · 2024-11-03

**Soundness:** 3
**Presentation:** 3
**Contribution:** 3
**Rating:** 6
**Confidence:** 3

**Summary:**

The paper extends the MO-MAB setting (Pareto bandits) to the setting where a fixed arm from the pareto front is the best action conditioned on user preferences. These user preferences are at the core of the novelty and the work is the first to introduce this additional structure on top of MO-MAB. The inclusion of user preference is motivated by the fact that different users weigh different metrics of a recommendation differently. For example, some users may care more about price, and others may care more about quality.

Different progressively more complex scenarios are considered in the work for user preference. First the case where user preferences can be observed directly are considered. Then the case where user preferences are stochastic but stationary and observed after taking an action are considered. Next the case where preferences are non-stationary but punctuated by breakpoints are considered. Finally the case of hidden preference is considered.

On top of the novel structure, the contributions of the work are a UCB-style algorithm for each kind of user-preference setting. They also analyze all the UCB variants and provide an upper bound on regret for each of these settings.

**Strengths:**

The work is well motivated by the limitations of the pareto bandit framework. They make a good case for their augmentations of the MO-MAB problem and provides several real-world examples to motivate their proposed structure. The presentation of the various variants of this structure is concise, well articulated, and backed by good visual representations. A principled algorithm is presented to solve the problem for each of the cases. Further these UCB-style algorithms are analyzed and order wise logarithmic instance dependent upper bounds on regret have been shown for them.

**Weaknesses:**

1. Some experiments should be included in the narrative of the main paper along with a discussion that contextualizes the baselines. Please consider adding an experiment comparing prior MAB-approaches listed in the literature (S-UCB, S-MOSS, Pareto-UCB, Pareto-TS) into the main paper. What I would be most interested to see is how and why these approaches that are agnostic to user preferences are configured for comparison. The how is already answered in Appendix A.1.1 however the why should also be addressed from a practitioner's point of view. While tests on synthetic data would also be informative, a test on a real dataset would be especially compelling since this would reveal the practical advantages or practical limitations of the PRUCB approach.

Currently there are no experiments in the main paper and there is no discussion of when and why a practitioner should use the UCB approach of this work as opposed to some other approach for solving their problem. Adding these experiments will address this limitation.

2. The authors might consider the generality of the assumptions in this work a strength but they seem unnecessary for some kinds of applications in the paper. See question 2.

**Questions:**

1. Please clarify the key differences between your setup in Section 5 and contextual bandits, or discuss the implications if the setup is indeed equivalent to contextual bandits in this case. I also request a more detailed explanation of how the approach differs from or extends contextual bandits in this specific scenario. This is in context to Remark 3.1 as well.

2. Perhaps modelling user preferences as quality constraints might be a better solution from a sample efficiency perspective? I am thinking of the MAB with cost subsidy paper by Sinha et al. (2021) for example which could with less structure address the problem example at the end of page 3 which reads: "For example, a user may prefer a 10 dollar meal with good taste over a 9.5 dollar meal with poor taste, even though cost is a priority." by imposing a "quality constraint" on taste.

---

> ### Author Response · Authors · 2024-11-21
> **Response to W1, Q1**
>
> > **W1:** Some experiments should be included in the narrative of the main paper along with a discussion that contextualizes the baselines. Please consider adding an experiment comparing prior MAB-approaches listed in the literature (S-UCB, S-MOSS, Pareto-UCB, Pareto-TS) into the main paper. What I would be most interested to see is how and why these approaches that are agnostic to user preferences are configured for comparison. The how is already answered in Appendix A.1.1 however the why should also be addressed from a practitioner's point of view. While tests on synthetic data would also be informative, a test on a real dataset would be especially compelling since this would reveal the practical advantages or practical limitations of the PRUCB approach.
> Currently there are no experiments in the main paper and there is no discussion of when and why a practitioner should use the UCB approach of this work as opposed to some other approach for solving their problem. Adding these experiments will address this limitation.
>
> **A1:** We appreciate this constructive suggestion and have addressed the reviewer’s helpful comments by modifying the paper accordingly. In the revised version, we added a section for experimental results and discussion, highlighted with blue-colored text in the main body. Due to the very limited space, we chose to present a subset of the experimental results under the stationary preference case.
>
> > **Q1:**  Please clarify the key differences between your setup in Section 5 and contextual bandits, or discuss the implications if the setup is indeed equivalent to contextual bandits in this case. I also request a more detailed explanation of how the approach differs from or extends contextual bandits in this specific scenario. This is in context to Remark 3.1 as well.
>
> Thank you for this comment. In Section 5, we study the PMO-MAB problem under known preferences, which differs fundamentally from contextual bandits in terms of problem formulation, methodology, and regret analysis. Below, we outline the key differences in each aspect.
>
> **Problem Formulation**
> - Contextual bandits: In linear-relational contextual bandits, the reward is defined as $r_ {a_t,t} = x_t^T \theta_ {a_t}$, where $x_t \in \mathbb{R}^D$ is the input context feature known in advance, $\theta_ {a_t}\in \mathbb{R}^D$ is an unknown parameter for arm $a_t$. In each step, the scalar reward $r_ {a_t}$ can be observed after pulling arm $a_t$. Importantly, the feedback in contextual bandits is **implicit**, meaning the unknown parameter $\theta_ {a_t}$ must be inferred from the scalar reward output and input features.
> - PAMO-MAB: In known preference PAMO-MAB, the overall reward is defined as $g_ {a_t,t} = c_t^T r_ {a_t}$, where $c_t \in \mathbb{R}^D$ is the known preference, $r_ {a_t}\in \mathbb{R}^D$ is the unknown reward of arm $a_t$. While this setup shares a similar linear relationship with contextual bandits, the key distinction lies in the **feedback mechanism**: in our problem, **the explicit feedback for $r_ {a_t}$ is observale after pulling arm $a_t$**. This allows us to directly estimate the reward $\hat{r}_ t$ using explicit historical data, bypassing the need for implicit feedback modeling as in contextual bandits.
>
> **Parameter Estimation**
> - Contextual bandits: Given the input features $\theta_ {a_t}$ and scalar outputs $r_ {a_t,t}$, the commonly used approach for parameter estimation in contextual bandits is linear regression to infer the unknown parameter $\theta_ {a_t}$.
> - PAMO-MAB: Due to the explicit feedback, the unknown reward mean vector $\mu_ {a_t}$ can be directly estimated using the empirical average from historical data (see Eq. (4) in main body).
>
> **Regret**
> - Contextual bandits: For linear contextual bandits using linear regression for parameter estimation (e.g., LinUCB), the regret bound is typically $O(\sqrt{T \log T}) [1]$.
> - PAMO-MAB: Due to the explicit feedback provided, we can have a smaller estimation error, with the regret bound of $O( \log T)$ (Theorem 2 in main body).
>
> [1] Chu et al. Contextual bandits with linear payoff functions. AISTATS, 2011.

---

> ### Author Response · Authors · 2024-11-21
> **Response to Q2**
>
> > **Q2:** Perhaps modeling user preferences as quality constraints might be a better solution from a sample efficiency perspective? I am thinking of the MAB with cost subsidy paper by Sinha et al. (2021) for example which could with less structure address the problem example at the end of page 3 which reads: "For example, a user may prefer a 10 meal with good taste over a 9.5 meal with poor taste, even though cost is a priority." by imposing a "quality constraint" on taste.
>
> **A3:** We sincerely thank the reviewer for suggesting this interesting perspective and for providing the related work [1]. Below, we discuss how modeling user preferences as quality constraints compares to our approach and provide an analysis of sample efficiency.
>
> 1.  **Explicit Preference Modeling vs. Quality Constraints**:
> Our framework explicitly models user preferences as an unknown vector $c_t$​, which interacts with arm rewards $r_{a_t}$​​ to determine the overall reward $g_{a_t} = c_t^\top r_{a_t}$. This approach allows for dynamically learning user preferences and optimizing arms based on trade-offs between multiple objectives.
> In contrast, the Cost-Subsidy approach introduces **quality constraints** to enforce fixed thresholds for secondary objectives (e.g., requiring taste to exceed a set level). While this method can improve sample efficiency by narrowing exploration to feasible arms, it imposes **hard boundaries** that may oversimplify user preferences and ignore trade-offs. For instance:
> 	- If cost is the primary objective and a taste constraint is applied, the algorithm will only explore arms where taste exceeds the threshold, ignoring those just below it.
> 	- However, a user might still prefer a slightly lower taste if the cost is sufficiently reduced.
>
> 	Our preference-aware approach inherently captures such trade-offs, offering greater flexibility by adapting to user-specific priorities dynamically, rather than being restricted by rigid constraints.
>
> 2. **Sample Efficiency**:
>
>     We respectfully argue that the Cost-Subsidy approach does not necessarily achieve higher sample efficiency compared to our proposed PRUCB. Below, we address both exploration and exploitation phases:
>
>     *a. Sample Efficiency in Exploration:*
> 	- The Cost-Subsidy approach begins with a **pure exploration phase**, requiring uniform sampling of all arms to estimate rewards and feasibility. This explicit exploration phase incurs a substantial cost, requiring $O(T^\frac{2}{3})$ samples to achieve sufficient exploration [1].
> 	- In contrast, exploration in PRUCB is **embedded into the decision-making process**, leveraging dynamically shrinking confidence intervals as more samples are collected. This adaptive exploration directly focuses on identifying the best arm while avoiding unnecessary sampling of suboptimal arms. Our theoretical analysis demonstrates that with $O(\log T)$ samples for exploration, the number of suboptimal pulls can be controlled by a constant, providing superior sample efficiency during this phase.
>
> 	*b. Sample Efficiency in Exploitation:*
> 	- After exploration, the Cost-Subsidy approach [1] applies **hard constraints** to prune infeasible arms, optimizing solely within the feasible set. This pruning can improve sample efficiency in the exploitation (Commit) stage by focusing on a smaller set of arms. However, the feasibility of each step is still determined using UCB-based bounds. As a result, high-variance arms with inflated confidence intervals are not pruned and will still be actively explored instead. This persistent exploration of high-variance arms aligns closely with PRUCB and may lead to comparable sample efficiency in the exploitation phase.
>
> [1] Sinha, Deeksha, et al. Multi-armed bandits with cost subsidy. International Conference on Artificial Intelligence and Statistics, 2021.

---

> > ### Comment · Reviewer_sjsn · 2024-11-24
> > **Response to authors**
> >
> > The reviewer thanks the authors for providing clarifications which are well received.
> > New details I have understood from the response don't cause a change in my evaluation.

---

### Official Review · Reviewer_FbNt · 2024-11-04

**Soundness:** 3
**Presentation:** 3
**Contribution:** 2
**Rating:** 5
**Confidence:** 3

**Summary:**

This paper studies the multi-objective multi-armed bandit (MO-MAB) problem. Different from the existing works that focus on the Pareto optimality, this work studies a preference-aware MO-MAB problem. To tackle this problem, the authors model the overall-reward as the inner product of the arm reward and the user preference. For this model, the authors give a comprehensive study under three preference structures, with corresponding algorithms that achieve provable sublinear regret regret. Finally, the theoretical results are validated by experiments.

**Strengths:**

1. The authors propose a natural preference-aware MO-MAB problem and conduct a systematic study with several different settings, considering both unknown rewards and/or preferences as well as the corruption and non-stationary settings.
2. The presentation is very clear and easy to follow. Almost all settings are illustrated with intuitive examples to motivate each setting.

**Weaknesses:**

Although the authors give a comprehensive study over a wide range of settings,  each setting can be easily reformulated or mapped as existing MAB settings, where leveraging existing approaches such as UCB/LinUCB with slight modification yields the result of the current paper. The core is to use the scalarization function to weight each objective and the scalarization function is modeled as a (unknown) preference vector. As such, the current study seems to be a wide combinatorial of separate pieces but lacks theoretical depth, which delivers few interesting and insightful intellectual merits.

**Questions:**

One interesting question is to study the case where only the overall reward can be observed or the case where the independence assumption does not hold. With discussions on these new settings, this paper can make more interesting and novel contributions to the community.

---

> ### Author Response · Authors · 2024-11-21
> **Response to W1 and Q1**
>
> We thank the reviewer for these constructive feedbacks. We hope our point-to-point response can address the reviewer's concerns and clarify our contribution.
>
> >  **W1**: Although the authors give a comprehensive study over a wide range of settings, each setting can be easily reformulated or mapped as existing MAB settings, where leveraging existing approaches such as UCB/LinUCB with slight modification yields the result of the current paper. The core is to use the scalarization function to weight each objective and the scalarization function is modeled as a (unknown) preference vector. As such, the current study seems to be a wide combinatorial of separate pieces but lacks theoretical depth, which delivers few interesting and insightful intellectual merits.
>
> **A1**:
>
> We respectfully disagree with the statement that our work is a trivial combination of previous works with little theoretical depth. The main theoretical novelties of our work are summarized in the following (also see our global response block for a more detailed illustration of our contributions including problem formulation, theoretical analyses and broader insights).
>
> For the unknown preference case (Alg. 2), we propose a new analytical method to address the joint impact from the preference and reward estimate errors, by introducing a tunable parameter $\epsilon$ to decompose the suboptimal actions into two disjoint sets. Then we characterize the regret for each disjoint component separately. Both these steps and our new developments therein differ from the standard approach.
>
> For unknown, non-stationary preference cases (Alg. 3), the dynamic environments lead to a potential mismatch between the dynamic episodes where an arm $i$ acting suboptimally and the episodes where arm $i$ is deemed to be suboptimal under preference estimates, making it challenging to analyze the decomposed regret terms based on preference estimates. To address this, we theoretically characterized  the learner’s behavior under UCB-based policy built on preference estimates and prove that it serves as an upper bound for the regret under true dynamic preferences.
>
> For the hidden preference cases (Alg. 4), we address the new challenge in preference estimation due to random mappings from reward to overall-reward by rigorously exploring the use of regression for preference estimation in this setting. In the regret analysis, the randomness of input rewards makes it challenging to directly bound the expected error term $\sum_{t=1}^{T}|(c_t - \hat{c}_ t)^T \mu_ {a_t}|$ using the empirical confidence region $\Vert \Upsilon_t^{\frac{1}{2}} (c - \hat{c}_t) \Vert_2 \leq \sqrt{\beta_t}$. To address this, we derive a novel upper bound based on the confidence region normed by the expected Gram matrix $\mathcal{E}[\Upsilon_t]$, enabling regret analysis involving $\mu$ with new theoretical insights.
>
>
> > **Q1** One interesting question is to study the case where only the overall reward can be observed or the case where the independence assumption does not hold. With discussions on these new settings, this paper can make more interesting and novel contributions to the community.
>
> **A2**: We thank the reviewer for suggesting these intriguing directions.
>
> - In the case where only the overall reward is observed without additional information, it essentially reduces to the standard scalar MAB setting, where scalar rewards are observed for each arm, and optimization is performed accordingly.
> - For the case where rewards and preferences are dependent, we agree this is a very interesting problem. Addressing this dependency would be challenging for algorithm design, as it would require joint estimation of rewards and preferences rather than treating them independently. Moreover, it would complicate the theoretical analysis, introducing an additional layer of dependence that would need to be rigorously modeled and accounted for in the regret bounds.
>
> While we are excited to explore these problems in future work, we believe they are beyond the scope of this study. We emphasize that, even without addressing these aspects, our work provides significant contributions to the community, particularly in aligning bandit outputs with human preferences. For further details, please refer to our global response.

---

> > ### Comment · Reviewer_FbNt · 2024-11-24
> >
> > The reviewer thanks the authors for their comprehensive feedback. I recognize the multiple contributions the authors have made to adapt the algorithm for various specific settings. However, when considered collectively, these contributions, from my perspective, do not seem to represent a significant in-depth advancement to the multi-objective/contextual MAB literature. As such, I will maintain my current evaluation.

---

### Official Review · Reviewer_zTNp · 2024-11-07

**Soundness:** 2
**Presentation:** 3
**Contribution:** 3
**Rating:** 6
**Confidence:** 3

**Summary:**

This paper introduces multi-objective multi-armed bandit (MO-MAB) problems with user-specific preferences. Unlike traditional approaches focusing solely on achieving Pareto optimality, this Preference-Aware MO-MAB  framework emphasizes optimizing within the Pareto front based on individual user preferences. Three different user preference structures are explored: known, unknown, and hidden preferences. Tailored algorithms are proposed for each scenario, achieving provably efficient sub-linear regret bounds.

**Strengths:**

* The paper proposes a new Preference-Aware MO-MAB  framework and provides very illustrative examples to help the reader understand it.

* Multiple different settings for the preference feedback are considered. This paper designed a tailored algorithm for each particular setting and provides comprehensive theoretical analyses.

* Sublinear regrets are guaranteed under all considered settings.

**Weaknesses:**

While this paper appears interesting and promising, several concerns need to be raised, including issues related to the presentation, the novelty of the algorithms, and the solidity of the theoretical results. Please refer to **Questions** below.

**Questions:**

* The main concern from this reviewer's point of view is the solidity of Theorem 5. When the preference is arbitrarily changed over time, how is it possible to find the best arm $a_t^*$ for each time before you know the preference?  Then the regret definition in Eq. (2) does not make sense for this setting. The regret $R(T) = O(\delta \log(T) + D^{1/3}T^{1/3})$ is highly skeptical as in this case, regret can be linearly increasing as $T$. Similar to the adversarial bandit setting where the rewards of each arm can be arbitrarily changed over time, the goal is never to find the best arm for each time, which is impossible.

* My second concern is that Algorithms 1-3 differ only in the estimation of the preference feedback, and are quite similar to conventional basic bandit settings.  In terms of algorithm design, there is really no novelty in this paper, although it is acceptable if the methods demonstrate strong performance. The concern is that the reviewer cannot intuitively explain why this more challenging multi-objective optimization with two unknown vectors in this paper can achieve nearly the same regret (e.g., Theorems 2, 3, 4 ) as the basic bandit setting with one scaler optimization, given the fact that the algorithms are largely similar? I do not see any additional technique to address such challenges. Can you clarify this?

* Could you explain the rationale behind using Eqs. (5), (8), and (9) for estimating preferences? What is the underlying reasoning for adopting this approach to preference estimation?  What is $c_{t-1}$ in Eq. (5)?

* It is unclear whether Proposition 1 defines the regret as in Eq. (2), where preferences are considered, even though the algorithms in those references are preference-free.

* Is that fair to compare with preference-free algorithms in terms of the regret considering preference? See lines 277-291 for a reference.

* Though the main paper contains a substantial amount of theoretical results and no space for numerical results, it is recommended to have some key results in the main paper as well.

*It is also recommended to include a table or a dedicated section to discuss the differences and similarities among all the algorithms and their respective regret orders. The prefactors are currently vague and difficult to understand.

---

> ### Author Response · Authors · 2024-11-21
> **Response to Q1**
>
> We thank the reviewer for this helpful feedback. We hope our point-to-point response can address the reviewer's concerns and clarify our contribution. We have revised the paper accordingly to address the comments.
>
> > **Q1:** The main concern from this reviewer's point of view is the solidity of Theorem 5. When the preference is arbitrarily changed over time, how is it possible to find the best arm  $a_t^*$  for each time before you know the preference? Then the regret definition in Eq. (2) does not make sense for this setting. The regret  $R(T)=O(\delta \log⁡(T)+D^{1/3}T^{1/3})$  is highly skeptical as in this case, regret can be linearly increasing as  T. Similar to the adversarial bandit setting where the rewards of each arm can be arbitrarily changed over time, the goal is never to find the best arm for each time, which is impossible.
>
> **A1:** Thank you for this comment. Theorem 5 characterizes the regret under **abruptly changing preference environments**, which differs from arbitrarily changing (adversarial) environments as you noted (see our first bullet below for details). Furthermore, we assert that the formal regret bound in Theorem 5 can also effectively capture the regret under adversarial settings, treating them as a special case of abruptly changing environments (see second bullet below for details).
>
> We give the detailed explanation as follows:
> - Specifically, in abruptly changing environments, we assume that the preference distribution $c_t$ remains fixed over intervals but changes at unknown time points, referred to as breakpoints (see Lines 386–387 in our manuscript). The number of breakpoints, $\psi_T$ can be chosen between $[0, T]$. Within each interval between two breakpoints, the problem can be treated as stationary, allowing us to model preferences and identify the best arm effectively.
>
> - Interestingly, our formal regret bound also encapsulates adversarial environments as a special case and characterizes the $O(T)$ regret under adversarial environments. Please note that regret $R(T) = O(\delta \log(T) + D^{\frac{1}{3}} T^{\frac{1}{3}})$ represents the best-case scenario when the number of breakpoints $\psi_T = O(1)$. A more formal regret bound is: $R(T) = O(\delta \log(T) + D^{\frac{1}{3}} \psi_T^{\frac{2}{3}} T^{\frac{1}{3}})$ (see Lines 4158–416 in our manuscript), which accounts for $\psi_T$, the number of breakpoints.
> Specifically, if every time step is a breakpoint, i.e., $\psi_T = O(T)$, the problem reduces to an adversarial setting. In such cases, the regret becomes: $R(T) = O(\delta \log(T) + D^{\frac{1}{3}} T^{\frac{2}{3}} T^{\frac{1}{3}}) = O(T)$, which aligns with the fact that in adversarial environments, it is impossible to identify the best arm at each step due to the arbitrary changes in preferences.

---

> ### Author Response · Authors · 2024-11-21
> **Response to Q2**
>
> > **Q2:** My second concern is that Algorithms 1-3 differ only in the estimation of the preference feedback, and are quite similar to conventional basic bandit settings. In terms of algorithm design, there is really no novelty in this paper, although it is acceptable if the methods demonstrate strong performance. The concern is that the reviewer cannot intuitively explain why this more challenging multi-objective optimization with two unknown vectors in this paper can achieve nearly the same regret (e.g., Theorems 2, 3, 4 ) as the basic bandit setting with one scalar optimization, given the fact that the algorithms are largely similar? I do not see any additional technique to address such challenges. Can you clarify this?
>
> **A2:**
> We thank the reviewer for raising this concern and apologize for omitting some detailed explanations due to space limitations in the main body. The reason the methods for PAMO-MAB achieve comparable regret to simpler scalar MAB can be thoroughly explained by our regret analysis.
>
> To address this concern, we first illustrate the connections between **known preference** PAMO-MAB and traditional scalar MAB, which form the theoretical foundation for addressing the **preference unknown** PAMO-MAB case (see the Known Preference Case part below for the detailed explanation). Next we extend to the preference unknown problem, where our regret analysis shows that the case dominating the regret can be transformed to a preference known case with a narrowed overall-reward gap (see the Unknown Preference Case part below for the detailed explanation). It bridges the preference unknown PAMO-MAB to the **known preference** PAMO-MAB, which has close connection with standard MAB, providing a unified theoretical explanation for the comparable regret .
>
> A more detailed explanation is given below:

---

> ### Author Response · Authors · 2024-11-21
> **Response to Q2 (Continued)**
>
> - **Known Preference Case**
>
>     To address this concern, we first provide two key remarks and clarifications to illustrate the connections between **known preference** PAMO-MAB and traditional scalar MAB, which form the theoretical foundation for addressing the **preference unknown** PAMO-MAB case.
>     - **Remark 1.** If the distribution of $c_t$ is stationary with known $\overline{c}$, each arm can be viewed as having a stationary reward distributed with mean of $\overline{c}^T \mu_i \in \mathbb{R}$, and the goal is to maximizing accumulative reward. This reduces the problem to a standard MAB framework. By treating $\eta_{i}^{\uparrow} = \eta_{i}^{\downarrow} = \overline{c}^T \Delta_i$ as the reward gap between arm $i$ and the best arm $a^*$, and $\delta$ as the upper-bound of reward $c_t^T r_{a_t,t}$ in each round $t$, our result in Theorem 2 matches the typical UCB bounds [1].
>     - **Remark 2.** Interestingly, the standard stochastic MAB can also be seen as a special case of PAMO-MAB with known preferences. Specifically, a $K$-armed stochastic bandit with reward means $x_1, \dots, x_K$ is equivalent to the PAMO-MAB case where $\exists j \in [D]$ s.t., $\boldsymbol{\mu}_ i(j) = x_i$, $\forall i \in [K]$ and $\overline{\boldsymbol{c}}_ t = \boldsymbol{e}_ j$ (the $j$-th standard basis vector). In this case, obviously the best arm $a_t^* = argmax_{i \in [K]} x_i$. Note $|\mathcal{\overline{C}}^{+}_ {T}| = 1$, $\eta_{i}^{\downarrow} = \eta_{i}^{\uparrow} = \Delta_i(j) = x_{a_{t}^*} - x_{i}$, the result in Theorem 2 can be rewrite as: $R(T) \leq \sum_{i=1}^{K}  \frac{4}{x_{a_{t}^*} - x_{i}} \log{(\frac{T}{\alpha})} + O(1)$, which recovers the bound in standard MAB [1].
>     - These remarks illustrate that under **stationary and known preference** environments, by introducing the preference-aware optimization, PAMO-MAB can be related to a standard MAB and is solvable using conventional techniques. We have also added this discussion in Appendix C.2 in our revised version and highlighted it with blue-colored text. For **non-stationary but known preferences**​, we develop an upper bound (Proposition 8, Appendix C.1), demonstrating that with sufficient sampling ($O(\log T)$), the number of suboptimal pulls can be bounded by a constant even under non-stationary preference. The above analysis demonstrates the feasibility of extending standard MAB methods to PAMO-MAB under known preferences. Next we extend to the unknown preference case.
>
> - **Unknown Preference Case**
>
>     For cases where preferences are unknown, we propose an innovative analytical framework that employs a tunable measure $\epsilon$ to decouple suboptimal pulls into two disjoint components (Step-1, Eq. (24) in Appendix D.1.2):
>     1. **Suboptimal pulls under precise preference estimation**: In this case, we show that the pseudo episode set $\mathcal{M}_i$ where the suboptimal arm $i$ is considered suboptimal under the preference estimate align with its true suboptimal episode set $\mathcal{T}_i$ (Step-2, Eq. (25) in Appendix D.1.2), and the best arm is consistently identified as better than arm $i$. Using this insight, we demonstrate that the scenario with precise preference estimation can be transformed to a known preference instance, but with a narrower overall-reward gap w.r.t $\epsilon$  (Step-2, Eq. (26) in Appendix D.1.2). This demonstrates that the regret remains asymptotically comparable to the known preference case.
>     2. **Suboptimal pulls under imprecise preference estimation**: The regret from this case is shown to be negligible compared to that from reward estimation errors. This is due to the much faster convergence rate of the preference estimation mechanism ($O(1/\sqrt{t})$) compared to the reward estimation rate ($O(1/\sqrt{\log t})$).
>
> The analysis above highlights the feasibility of extending techniques for known preferences to the unknown case, provided a well-designed preference estimation mechanism is employed. Since the dominant suboptimal case can be transformed to a known PAMO-MAB setting, and given the close performance of preference known PAMO-MAB with standard MAB, our analysis explains the reason behind the comparable performance of unknown PAMO-MAB and standard MAB.
>
> [1] Auer et al. Finite-time analysis of the multiarmed bandit problem. Machine learning, 2002.

---

> ### Author Response · Authors · 2024-11-21
> **Response to Q3**
>
> > **Q3:** Could you explain the rationale behind using Eqs. (5), (8), and (9) for estimating preferences? What is the underlying reasoning for adopting this approach to preference estimation? What is  ct−1  in Eq. (5)?
>
> **A3:** The Eqs. (5), (8) are the empirical estimation of the preference based on the hisitorical data, which is expressed in an incremental way for the ease of updating.  Eq. (9) is the closed-form solution for the $\ell_2$-regularized least-squares estimate of $\overline{c}$. Specifically, we provided the detailed explanations for each equation in the following:
>
> **Eq (5):**
>
> For Eq (5) we have $\hat{c}_ {t} = \frac{(t-2) \hat{c}_ {t-1} + c_{t-1}}{t-1} = \frac{(t-3) \hat{c}_ {t-2} + c_{t-2} +  c_{t-1}}{t-1} = \frac{0 \cdot \hat{c}_ {1} + c_{1} + \cdots + c_{t-1}}{t-1} = \frac{1}{t-1} \sum_{i=1}^{t-1} c_{i}$, where $c_{i}$ denotes the observed preference feedback at $i$-th step. This formulation indicates that the preference estimation $\hat{\boldsymbol{c}}_{t}$ used in step $t$ is the empirical average of the observed preference feedback over the past $t-1$ steps. Since in stationary case, the distribution of preference remains fixed, the sample mean​ is an unbiased estimator of the true preference expectation $\overline{c}$​, and by the law of large numbers, it converges to $\overline{c}$ as $t \to \infty$, we employ this approach for preference estimation.
>
> **Eq (8):**
>
> For Eq. (8), we adopt a similar approach to empirical averaging for preference estimation. However, since the preference distribution is no longer stationary, using an empirical average over the entire historical data may lead to biased estimates, as the current preference could follow a distribution different from that of past data. To address this issue, we employ a sliding-window-based estimation approach, which considers only the most recent $\tau$ plays for averaging.
>
> In an abruptly changing environment, where the preference distribution changes periodically but remains constant within intervals, this method ensures that the samples used for averaging are predominantly drawn from a distribution consistent with the current preference. By focusing on recent data and discarding outdated samples, the sliding-window approach effectively reduces estimation bias.
>
> Eq. (8) formulates this sliding-window average, where preference estimation is based on at most $\tau$ historical observations. Specifically:
> - When $t \leq \tau+1$, all previous $t-1$ samples are used for estimation: $\hat{c}_ {t} = \frac{1}{t-1} \sum_{\ell = 1 }^{t-1} c_{\ell}$
> - When $t > \tau+1$, only the most recent $\tau$ samples are used for estimation:  $\hat{c}_ {t} = \frac{1}{\tau} \sum_{\ell = t - \tau }^{t-1} c_{\ell}$
>
> By combining these two cases, Eq. (8) provides a unified formulation for preference estimation in dynamic environments.
>
> **Eq (9):**
>
> For the hidden preference cases, we can only observe the reward feedback $r_{a_t}$ and overall reward feedback $g_{a_t}$. By problem formulation, we have $g_{a_t,t} = c_{t}^T r_{a_t,t}$, where the distribution of $c_{t}$ is stationary. In this setting, the estimation of preferences can be framed as a regression problem, where the input data samples $\\{r_{a_i}\\}_ {i=1}^{t-1}$ and output data samples $\\{g_{a_i}\\}_ {i=1}^{t-1}$ are used as the training data to learn the predictor coefficients $\overline{c}$. Despite the dependent residual error introduced, we show that the residual error $\zeta_t^\top r_{a_t, t}$​ vanishes in expectation without biasing the predictor, thereby establishing the feasibility of using standard linear regression for estimating $\overline{c}$ in PAMO-MAB. Specifically, Eq (9) provides the closed-form solution for the $\ell_2$-regularized least-squares estimate of $\overline{c}$ based on the previous $t-1$ samples. This approach ensures effective preference estimation while maintaining computational efficiency.

---

> ### Author Response · Authors · 2024-11-21
> **Response to Q4-Q7**
>
> > **Q4:** It is unclear whether Proposition 1 defines the regret as in Eq. (2), where preferences are considered, even though the algorithms in those references are preference-free.
>
> **A4:** Proposition 1 aims to establish a theoretical baseline by evaluate **preference-aware regret** (as defined in Eq. (2)) using **preference-free algorithms**. While preference-free algorithms do not incorporate preferences in their decision-making, the comparison is fair because Eq. (2) reflects the natural regret metric in environments influenced by preferences. The purpose of Proposition 1 is to emphasize that preference-free algorithms suffer linear regret ($O(T)$) in such settings, underscoring the necessity and advantages of preference-aware methods.
>
> > **Q5:** Is that fair to compare with preference-free algorithms in terms of the regret considering preference? See lines 277-291 for a reference.
>
> **A5:** Please see **A4** for a detailed reply.
>
> > **Q6:** Though the main paper contains a substantial amount of theoretical results and no space for numerical results, it is recommended to have some key results in the main paper as well.
>
> **A6:** We appreciate this constructive suggestion and have addressed it by modifying the paper accordingly. In the revised version, we added a section for experimental results and discussion, highlighted with blue-colored text in the main body. Due to limited space, we chose to present a subset of the experimental results under the stationary preference case in the main body.
>
> > **Q7:** It is also recommended to include a table or a dedicated section to discuss the differences and similarities among all the algorithms and their respective regret orders. The prefactors are currently vague and difficult to understand

---

### Author Response · Authors · 2024-11-21
**Global Response**

Dear Area Chairs and Reviewers,

We sincerely appreciate the valuable feedback and constructive suggestions from the reviewers. Below, we summarize the **main contributions of our work to the community**, with detailed responses provided in the respective comments:

1. **New Problem Setting**: In our study, we introduce a new setting that incorporates user preferences into the multi-objective bandit structure, aiming to address the practical and valuable problem of aligning bandit outputs with explicit human preferences, which is typically not considered in existing multi-objective bandit research.

2. **Theoretical Contributions**:

    - For the unknown preference cases, we propose a new analytical method to address the joint impact from the preference and reward estimate errors, by introducing a tunable parameter $\epsilon$ to decompose the suboptimal actions into two disjoint sets. Then we characterize the regret for each disjoint component separately. Both these steps and our new developments therein differ from the standard approach.

    - For non-stationary preference cases (Alg. 3), the dynamic environments lead to a potential mismatch between the dynamic episodes where an arm $i$ acting suboptimally and the episodes where arm $i$ is deemed to be suboptimal under preference estimates, making it challenging to analyze the decomposed regret terms based on preference estimates. We address this challenge by theoretically characterizing  the learner’s behavior under UCB-based policy built on preference estimates and proving that it serves as an upper bound for the regret under true dynamic preferences (see A4 for details).

    - For the hidden preference cases, we address the new challenge in preference estimation due to random mappings from reward to overall-reward by rigorously exploring the use of regression for preference estimation in this setting. In the regret analysis, the randomness of input rewards makes it challenging to directly bound the expected error term $\sum_{t=1}^{T}|(c_t - \hat{c}_ t)^T \mu_ {a_t}|$ using the empirical confidence region $\Vert \Upsilon_t^{\frac{1}{2}} (c - \hat{c}_t) \Vert_2 \leq \sqrt{\beta_t}$. To address this, we derive a novel upper bound based on the confidence region normed by the expected Gram matrix $\mathbb{E}[\Upsilon_t]$, enabling regret analysis involving $\mu$ with new theoretical insights.

    Our analytical approaches go beyond traditional methods, providing deeper insights into regret specific to PAMO-MAB settings and establishing a solid analytical foundation for PAMO-MAB.

3. **Broader Insights**: Our analysis provides broader insights into the fundamental limitations and trade-offs in multi-objective bandits, such as the inherent conflicts in preference-free algorithms (Proposition 1) and the sub-linear regret achievable with preference-aware methods.

We believe these contributions provide meaningful theoretical depth and practical relevance to the field, advancing beyond combinatorial modifications of existing approaches.

---

### Meta-Review · Area_Chair_EJaR · 2024-12-17

**Metareview:**

This paper proposes several multi-objective bandit algorithms. In all, the reward of the arm is the dot product between the multi-objective rewards of the arm and user preferences. The preferences are either known, observed i.i.d., or estimated. The authors propose several algorithms, analyze them, and evaluate them empirically. The scores of this paper are 2x 6, 5, and 3; and did not change during the discussion. The reviewers had several concerns:

* **Technical novelty:** The settings of Algorithms 1-3 present no major challenge because the user preferences are either known or revealed i.i.d. independently of the pulled arm.

* **Technical correctness:** Algorithm 4 is wrong. The reason is that the least-squares estimator is a wrong estimator when the feature vectors are random. I encourage the authors to read on [Deming regression](https://en.wikipedia.org/wiki/Deming_regression) and [total least squares](https://en.wikipedia.org/wiki/Total_least_squares). Reviewer dpuc also showed that proofs in this paper can be simplified.

* **Writing:** The writing needs to be simplified and put in context. All proposed methods are based on linear scalarization, where the scalarization vector represents user preferences. This is the simplest form of multi-objective optimization. Linear scalarization has been used in bandits since [Designing multi-objective multi-armed bandits algorithms: A study](https://ieeexplore.ieee.org/document/6707036).

These concerns require a major revision and therefore the paper cannot be accepted at this time.

**Additional Comments On Reviewer Discussion:**

See the meta-review for details.

---

### Decision · Program_Chairs · 2025-01-22

Reject